# A seamless ensemble-based reconstruction of surface ocean $p\mathrm{CO_2}$ and air–sea $\mathrm{CO_2}$ fluxes over the global coastal and open oceans

Thi Tuyet Trang Chau, Marion Gehlen, and Frédéric Chevallier

Laboratoire des Sciences du Climat et de l'Environnement, LSCE/IPSL, CEA-CNRS-UVSQ, Université Paris-Saclay, F-91191 Gif-sur-Yvette, France

**Correspondence:** Thi Tuyet Trang Chau (trang.chau@lsce.ipsl.fr)

**Abstract.** We have estimated global air–sea $\mathrm{CO_2}$ fluxes ($fg\mathrm{CO_2}$) from the open ocean to coastal seas. Fluxes and associated uncertainty are computed from an ensemble-based reconstruction of $\mathrm{CO_2}$ sea surface partial pressure ($p\mathrm{CO_2}$) maps trained with gridded data from the Surface Ocean $\mathrm{CO_2}$ Atlas v2020 database. The ensemble mean (which is the best estimate provided by the approach) fits independent data well and a broad agreement between the spatial distribution of model–data differences and the ensemble standard deviation (which is our model uncertainty estimate) is seen. Ensemble-based uncertainty estimates are denoted by $\pm 1\sigma$. The space-time varying uncertainty fields identify oceanic regions where improvements in data reconstruction and extensions of the observational network are needed. Poor reconstructions of $p\mathrm{CO_2}$ are primarily found over the coasts and/or in regions with sparse observations, while $fg\mathrm{CO_2}$ estimates with largest uncertainty are observed over the open Southern Ocean ($44°S$ southward), the subpolar regions, the Indian gyre, and upwelling systems.

Our estimate of the global net sink for the period 1985–2019 is $1.643 \pm 0.125\ \mathrm{PgC\,yr^{-1}}$ including $0.150 \pm 0.010\ \mathrm{PgC\,yr^{-1}}$ for the coastal net sink. Among the ocean basins, the subtropical Pacific (18°N–49°N) and the subpolar Atlantic (49°N–76°N) appear respectively to be the strongest $\mathrm{CO_2}$ sinks for the open ocean and the coastal ocean. Based on mean flux density per unit area, the most intense $\mathrm{CO_2}$ drawdown is, however, observed over the Arctic (76°N poleward) followed by the Subpolar Atlantic and Subtropical Pacific for both open ocean and coastal sectors. Reconstruction results also show significant changes in the global annual integral of all open- and coastal-ocean $\mathrm{CO_2}$ fluxes with a growth rate of $+0.062 \pm 0.006\ \mathrm{PgC\,yr^{-2}}$ and a temporal standard deviation of $0.526 \pm 0.022\ \mathrm{PgC\,yr^{-1}}$ over the 35-year period. The link between its large interannual to multi-year variations and the El Niño-Southern Oscillation climate variability is reconfirmed.

# 1  Introduction

Since the onset of the Industrial Era, humankind has profoundly modified the global carbon (C) cycle. The use of fossil fuels, cement production, and land use change has added $700 \pm 75$ PgC (best estimate $\pm 1\sigma$) to the atmosphere between 1750 and 2019 (Friedlingstein et al., 2020). An estimated $285 \pm 5$ PgC of this excess C stayed there, the remainder was taken up by the ocean ($170 \pm 20$ PgC) and the land biosphere ($230 \pm 60$ PgC). While the fraction of total $CO_2$ emissions sequestered by the ocean remained rather stable ($22 - 25\%$) over the past six decades (Friedlingstein et al., 2020), the global ocean sink has varied significantly at interannual time scales (Rödenbeck et al., 2015). Global ocean biogeochemical models (GOBMs) are used within the framework of the annual assessment of the global carbon budget (Friedlingstein et al., 2020) to annually re-estimate the means and variations of $CO_2$ sinks and sources over the global ocean and major basins. However, these recent model-based estimates need to be benchmarked against observation-based estimates in order to better understand the global carbon budget as well as its yearly re-distribution in the biosphere (Hauck et al., 2020).

In situ measurements of sea surface fugacity of $CO_2$ collected by an international coordinated effort of the ocean observation community and combined into the Surface Ocean $CO_2$ Atlas (SOCAT, https://www.socat.info/, Bakker et al., 2016) provide an observational constraint on the assessment of the surface ocean partial pressure of $CO_2$ ($pCO_2$) and the ocean C sinks and sources. Despite an increasing number of observations since the 1990s, data density remains uneven in space and time. While, for instance, data coverage is sparse over the Southern basins of the Atlantic and Pacific oceans, observations are seasonally biased towards the summers at high latitudes (Landschützer et al., 2014; Denvil-Sommer et al., 2019; Gregor et al., 2019).

Various data-based approaches have been proposed to infer gridded maps of surface ocean $pCO_2$ from the sparse set of observation–based data. They have been successful in obtaining similarly low misfits between the reconstructed and evaluation data and reasonable estimates of air-sea $CO_2$ fluxes (see in Rödenbeck et al., 2015; Gregor et al., 2019; Friedlingstein et al., 2020) although model design and implementation are quite different (e.g., proportion of SOCAT data used in model fitting and evaluation). Aside from data reconstruction built on a single model mapping $pCO_2$ data with machine learning, classical regression, or mixed layer schemes (e.g., Rödenbeck et al., 2013; Landschützer et al., 2016; Iida et al., 2021), ensemble-based approaches have recently emerged but with their own concepts and objectives. For example, Denvil-Sommer et al. (2019) designed a two-step reconstruction of $pCO_2$ climatologies and anomalies based on five neural network models and selected the one that reproduced the $pCO_2$ field with the smallest model–data misfit. Gregor et al. (2019) and Gregor and Gruber (2021) introduced machine-learning ensembles with six to sixteen different two-step clustering-regression models mapping surface $pCO_2$ and suggest that the use of their ensemble mean is better than each member estimate. In a broader context, Rödenbeck et al. (2015) presented an intercomparison of fourteen mapping methods targeting the identification of common or distinguishable features of different products in long-term mean, regional and temporal variations. Hauck et al. (2020) and Friedlingstein et al. (2020) also synthesized $pCO_2$ mapping products and took an ensemble of their observation–based estimates of air-sea $CO_2$ fluxes as a benchmark to compare with the one derived from ocean biogeochemical models.

Despite positive conclusions overall, statistical data reconstructions are still subject to further improvements. In Rödenbeck et al. (2015), Hauck et al. (2020), Bushinsky et al. (2019), and Denvil-Sommer et al. (2021), the authors explain that substantial

extensions of surface ocean observational network systems are essential to better determine $p\mathrm{CO}_2$ and fluxes at finer scales and reduce mapping uncertainties. So far mapping uncertainties have been estimated by using misfits between the model outputs and SOCAT data (e.g., the root-mean-square deviation, RMSD). By construction, such uncertainty estimates are restricted to oceanic regions and periods when observations are available (Rödenbeck et al., 2015; Lebehot et al., 2019; Gregor et al., 2019) and the uncertainty quantification of an averaged $p\mathrm{CO}_2$ or an integrated flux over space and time of interest is under low confidence due to sparse data density. Also, most of the aforementioned mapping methods target $p\mathrm{CO}_2$ data and estimate air-sea fluxes solely over the open ocean, with the coastal data excluded or not fully qualified. In Laruelle et al. (2014), the authors present spatial distributions of air-sea flux density and estimates of the total coastal C sink inferred from spatial integration methods on coastal SOCAT data. Laruelle et al. (2017) adapted the two-step neural network approach described in Landschützer et al. (2016) to the coastal ocean $p\mathrm{CO}_2$. The coastal and open ocean products were combined into a single reconstruction to yield a global monthly climatology of $p\mathrm{CO}_2$ presented in Landschützer et al. (2020). Notwithstanding these advances, a global reconstruction and its uncertainty assessment of monthly varying coastal surface ocean $p\mathrm{CO}_2$ and air-sea fluxes are still missing.

In this work, we propose a new inference strategy for reconstructing the monthly $p\mathrm{CO}_2$ fields and the contemporary air–sea fluxes over the period 1985–2019 with a spatial resolution of $1° \times 1°$. It is based on a Monte Carlo approach, an ensemble of 100 neural network models mapping sub-samples drawn from the monthly gridded SOCATv2020 data and available data of predictors. This ensemble approach was developed at the Laboratoire des Sciences du Climat et de l'Environnement (LSCE) as both an extension and an improvement of the first version (LSCE-FFNN-v1, Denvil-Sommer et al., 2019). In the following sections, we first present the ensemble of neural networks designed with the aim of leaving aside the issue of discrete boundaries in the existing two-step clustering-regressions (see further discussion in Gregor and Gruber, 2021) and reducing the mapping uncertainties induced by the two-step reconstruction of the $p\mathrm{CO}_2$ fields (Denvil-Sommer et al., 2019) or by an ensemble-based reconstruction with a small ensemble size. In addition, each FFNN model follows a leave-$p$-out cross-validation approach, i.e., the exclusion of $p$ gridded SOCAT data of the reconstructed month itself in model training and validation. This allows to reduce model over-fitting and to leave much more independent data for model evaluation than the previous studies. Mean and standard deviation computed from the ensemble of 100 model outputs are defined as estimates of the mean state and uncertainty of the carbon fields. As one of the novel key findings of this study compared to the existing ones, we compute and analyze the estimates of $p\mathrm{CO}_2$ and air–sea fluxes, model errors, and model uncertainties for different time scales (e.g., monthly, yearly, and multi-decadal) and spatial scales (e.g., grid cells, sub-basins, and the global ocean). We then suggest the use of an indicator map built on the space-time varying uncertainty fields instead of model–data misfits for identifying regions that should be prioritized in future observational programs and model development in order to improve data reconstruction. Last but not least, the model best estimates and uncertainty of $p\mathrm{CO}_2$ and air–sea fluxes are analysed seamlessly over the open ocean to the coastal zone. Potential drivers of the spatio-temporal distribution and the magnitude of open ocean and coastal $\mathrm{CO}_2$ fluxes are discussed with the aim to better identify underlying processes and to detect potential focus regions for further studies on the evolution of oceanic $\mathrm{CO}_2$ sources and sinks.

## 2 Methods

### 2.1 General formulation

The air–sea flux density $(\mathrm{molC\,m^{-2}\,yr^{-1}})$ is calculated here by the standard bulk equation

$$fg\mathrm{CO_2} = kL\left(1 - f_{\mathrm{ice}}\right)\Delta p\mathrm{CO_2}$$
$$= kL\left(1 - f_{\mathrm{ice}}\right)\left(p\mathrm{CO_2^{atm}} - p\mathrm{CO_2}\right), \tag{1}$$

where $k$ is the gas transfer velocity computed as a function of the 10-meter ERA5 wind speed (Hersbach et al., 2020) following Wanninkhof (2014) and its coefficient is scaled to match a global mean transfer velocity of $16.5\ \mathrm{cm\,h^{-1}}$ (Naegler, 2009). $L$ is the temperature-dependent solubility of $\mathrm{CO_2}$ (Weiss, 1974), $f_{\mathrm{ice}}$ and $p\mathrm{CO_2^{atm}}$ are, respectively, the sea ice fraction and the atmospheric $\mathrm{CO_2}$ partial pressure. In Eq. (1), a positive (negative) flux indicates oceanic $\mathrm{CO_2}$ uptake (release). Details and references for the source of these variables are given in Table S1, except for $p\mathrm{CO_2}$ that is described in the following section.

### 2.2 An ensemble-based approach for the reconstruction of sea surface $p\mathrm{CO_2}$ and air–sea $\mathrm{CO_2}$ fluxes

The sea surface partial pressure of $\mathrm{CO_2}$ in Eq. (1) is estimated monthly over each point of the global ocean by analysing sparse in situ measurements of $\mathrm{CO_2}$ fugacity, gathered and gridded at monthly and 1-degree resolution in the 2020 release of the Surface Ocean $\mathrm{CO_2}$ Atlas (SOCATv2020, https://www.socat.info/). SOCATv2020 covers the period 1985–2019. First, monthly gridded $p\mathrm{CO_2}$ data were converted from SOCATv2020 $\mathrm{CO_2}$ fugacity (Körtzinger, 1999). We have then regressed these $p\mathrm{CO_2}$ values against a set of predictors with non-linear functions, i.e., feed-forward neural network models (FFNNs). As illustrated in Fig. 1, our predictors are biological, chemical, and physical variables commonly associated with the variations of $p\mathrm{CO_2}$ (e.g., Landschützer et al., 2013; Denvil-Sommer et al., 2019; Gregor et al., 2019): sea surface height (SSH), sea surface temperature (SST), sea surface salinity (SSS), mixed layer depth (MLD), chlorophyll-a (CHL), atmospheric $\mathrm{CO_2}$ mole fraction ($x\mathrm{CO_2}$). A $p\mathrm{CO_2}$ climatology (Takahashi et al., 2009) and the geographical coordinates (latitude and longitude) were also added to the predictors. Table S1 details the data source. All data were reprocessed and co-located at the same SOCAT resolution following Landschützer et al. (2016) and Denvil-Sommer et al. (2019). For instance, CHL was set approximately to $0\ \mathrm{mg\,m^{-3}}$ over the Arctic and the Southern Ocean winter when no data is available. In case of data unavailable before 1998, climatologies based on all available data were used as predictors. Exceptionally, predictors for SSH before 1993 were climatologies plus a linear trend in order to retain the overall response to the global warming. MLD before 1992 was taken as the average MLD between 1992 and 1997.

An ensemble of 100 FFNNs was used to reconstruct monthly $p\mathrm{CO_2}$ fields with a $1° \times 1°$ resolution over the global surface ocean during years 1985–2019. This ensemble approach was developed at the Laboratoire des Sciences du Climat et de l'Environnement (LSCE) as both an extension and an improvement of the first version (LSCE-FFNN-v1, Denvil-Sommer et al., 2019). Our model outputs are part of the Copernicus Marine Environment Monitoring Service (CMEMS). Throughout the paper, it is hence referred to as CMEMS-LSCE-FFNN.

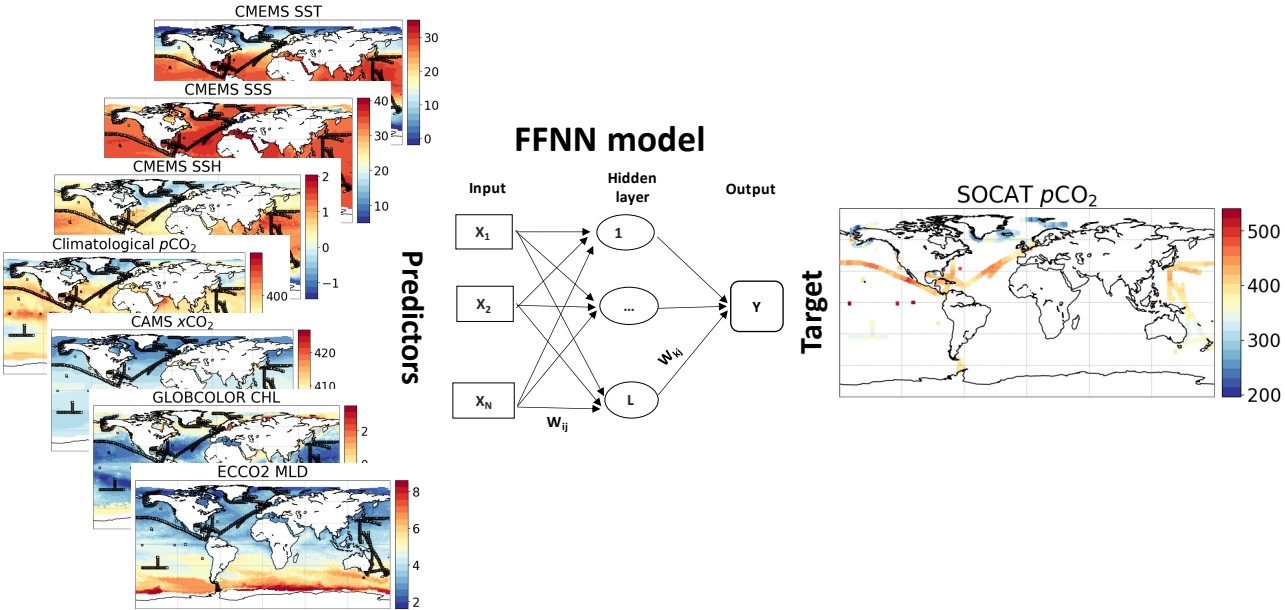

**Figure 1.** Illustration of a feed-forward neural network (FFNN) model mapping monthly gridded SOCAT data and feature variables (Table S1) co-located at a spatial resolution of $1° \times 1°$.

To reconstruct the $pCO_2$ fields over the global ocean for each target month over the 1985–2019 period, all the available SOCAT data and the co-located predictors have been collected for the month before and the month after the target month. We randomly extracted two thirds of each one of these datasets to make training datasets for the FFNNs, leaving the remaining third to be corresponding test datasets. The FFNNs were then trained for each target month. Moreover, the exclusion of the reconstructed month itself in the training and test datasets follows a leave-$p$-out cross-validation approach, where $p$ is the number of gridded SOCAT data in the target month. This approach allows to reduce model over-fitting, as well as to assess the quality of the reconstruction against SOCAT data that are fully independent from the training phase.

The random extraction and the FFNN training were repeated 100 times so that 100 versions of the monthly FFNNs have been obtained. Note that our ensemble approach belongs to the classes of bootstraping and Monte Carlo methods in statistics. Theoretically, the number of samples or the ensemble size must be substantially large to get a convergence. However, it was demonstrated in the literature (e.g., Goodhue et al., 2012; Efron et al., 2015) that with the ensemble size of 50 the model estimation is likely stable and with the ensemble size over 100 the improvement in standard errors between model outputs and evaluation data is negligible. Fig. S2 shows an illustration of the reconstruction skill with respect to the ensemble size $S$. For each ensemble of $S$ model outputs of $pCO_2$ ($S \in \{5, 10, 20, 50, 75, 100\}$), the root-mean-square deviation (RMSD) is computed between the ensemble mean (our best model estimate) and SOCAT data over the period 1985-2019. As seen in this figure, the reconstruction starts to stabilize with $S = 50$. In this study, we have exploited a large but realistic amount of

computing resources to run an ensemble of $S = 100$ neural network models. Equation (1) was then applied to the ensembles of FFNN outputs of $p\mathrm{CO}_2$ in order to obtain ensembles of monthly global $fg\mathrm{CO}_2$ fields.

## 2.3 Coastal and regional division

The reconstructed $p\mathrm{CO}_2$ fields and air–sea $\mathrm{CO}_2$ fluxes are analysed over the global ocean, at particular locations, and in 11 oceanic sub-basins used by the Regional Carbon Cycle Assessment Project Tier 1 (RECCAP1, Canadell et al., 2011) and previous studies (Schuster et al., 2013; Sarma et al., 2013; Ishii et al., 2014; Lenton et al., 2013; Wanninkhof et al., 2013; Landschützer et al., 2014). In order to distinguish the coastal from the open ocean, we use the coastal mask from the MARgins and CATchments Segmentation (MARCATS, Laruelle et al., 2013) interpolated on the $1° \times 1°$ SOCAT grid. Details of the regional (open and coastal) division are given in Table 1 and Fig. 2.

**Table 1.** Indication of 11 RECCAP1 regions (Fig. 2). Only the total area with respect to the maximum coverage of the reconstructed data is accounted for each region.

| Index | Region | Latitude | Area ($10^6\mathrm{km}^2$) | |
|---|---|---|---|---|
| | | | Open ocean | Coast |
| | Globe (G) | $90°\mathrm{S} - 90°\mathrm{N}$ | 330.42 | 22.35 |
| 1 | Arctic (Ar) | $76°\mathrm{N} - 90°\mathrm{N}$ | 1.07 | 0.99 |
| 2 | Subpolar Atlantic (SpA) | $49°\mathrm{N} - 76°\mathrm{N}$ | 8.88 | 4.15 |
| 3 | Subpolar Pacific (SpP) | $49°\mathrm{N} - 76°\mathrm{N}$ | 6.16 | 3.65 |
| 4 | Subtropical Atlantic (StA) | $18°\mathrm{N} - 49°\mathrm{N}$ | 23.22 | 1.83 |
| 5 | Subtropical Pacific (StP) | $18°\mathrm{N} - 49°\mathrm{N}$ | 36.37 | 1.65 |
| 6 | Equatorial Atlantic (EA) | $18°\mathrm{S} - 18°\mathrm{N}$ | 23.15 | 1.05 |
| 7 | Equatorial Pacific (EP) | $18°\mathrm{S} - 18°\mathrm{N}$ | 66.50 | 3.22 |
| 8 | South Atlantic (SA) | $44°\mathrm{S} - 18°\mathrm{S}$ | 17.79 | 0.83 |
| 9 | South Pacific (SP) | $44°\mathrm{S} - 18°\mathrm{S}$ | 37.15 | 0.50 |
| 10 | Indian Ocean (IO) | $44°\mathrm{S} - 30°\mathrm{N}$ | 52.80 | 2.71 |
| 11 | Southern Ocean (SO) | $90°\mathrm{S} - 44°\mathrm{S}$ | 59.47 | 3.12 |

With the above definitions, the coastal regions encompass $6.33\%$ of a total maximum ocean area of $352.77 \times 10^6$ km$^2$. The computation of these numbers was based on the maximum data coverage of the CMEMS-LSCE-FFNN reconstruction taking into account the variable monthly sea–ice fraction. The number of monthly gridded SOCATv2020 data used in the reconstruction of $p\mathrm{CO}_2$ is reported in Table S2 for each region, with 301,449 in total and $10.36\%$ of the data available over the predefined coastal regions.

## 2.4 Statistics

The mean ($\mu$) and standard deviation ($\sigma$) of the 100-member ensembles of $p\mathrm{CO}_2$ and $fg\mathrm{CO}_2$ are respectively chosen as their best estimate and the associated uncertainty. Unless stated otherwise, a model best estimate and its uncertainty computed at

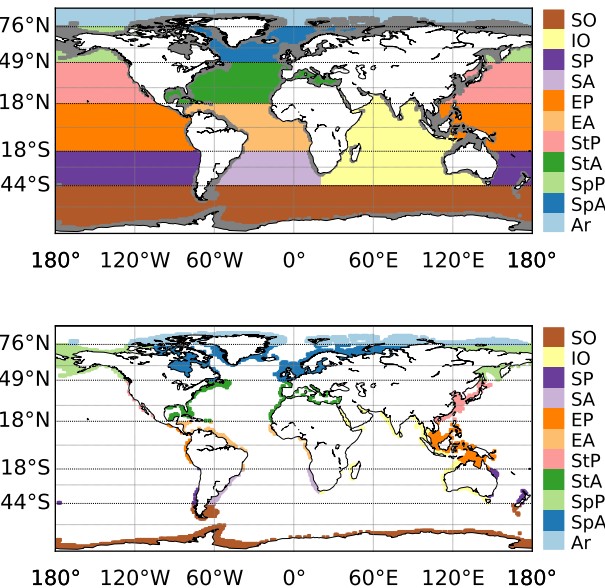

**Figure 2.** Map of RECCAP1 regions (Regional Carbon Cycle Assessment and Processes, Canadell et al., 2011) and MARCATS coastal mask (MARgins and CATchments Segmentation, Laruelle et al., 2013) co-located on the $1° \times 1°$ SOCAT grid.

each desired space-time resolution are denoted by $\mu_{\text{ensemble}} \pm \sigma_{\text{ensemble}}$, where

$$\mu_{\text{ensemble}} = \frac{\sum_{i=1}^{i=100} pCO_2^{\text{Reconstruction}(i)}}{100}, \quad \sigma_{\text{ensemble}} = \sqrt{\frac{\sum_{i=1}^{i=100} \left( pCO_2^{\text{Reconstruction}(i)} - \mu_{\text{ensemble}} \right)^2}{100}}, \quad (2)$$

and $pCO_2^{\text{Reconstruction}(i)}$ is one of the 100 members of the reconstructed $pCO_2$ fields. Similar definitions are applied for $fgCO_2$. The units of air-sea flux estimates is $\text{molC}\,\text{m}^{-2}\text{yr}^{-1}$ for a flux density and converted to $\text{PgC}\,\text{yr}^{-1}$ for an integral over a region or the global ocean.

Model robustness of the reconstructed $pCO_2$ fields is evaluated on the gridded SOCAT data and in situ observations (Sutton et al., 2019). The evaluation data is denoted as $pCO_2^{\text{Observation}}$ in the following formulas. Standard statistics include the coefficient of determination $(r^2)$, misfit mean (model bias) and misfit standard deviation,

$$\mu_{\text{misfit}} = \frac{\sum_{j=1}^{j=N} dpCO_2^j}{N}, \quad \sigma_{\text{misfit}} = \sqrt{\frac{\sum_{j=1}^{j=N} \left( dpCO_2^j - \mu_{\text{misfit}} \right)^2}{N}}, \quad (3)$$

and the root-mean-square deviation (RMSD)

$$\text{RMSD} = \sqrt{\frac{\sum_{j=1}^{j=N} \left( dpCO_2^j \right)^2}{N}}, \quad (4)$$

where $dpCO_2^j = pCO_2^{\text{Reconstruction}}[j] - pCO_2^{\text{Observation}}[j]$, and $N$ is a number of evaluation data. All these scores are computed for different coastal and open regions from the scale of grid cells to the global scale.

Generally, RMSD measures the reconstruction skill in terms of mean distance between model estimates and evaluation data while $r^2$ measures the proportion of data variation predicted by the model. Compared to other metrics such as mean absolute bias and $r^2$, RMSD takes another role, an outlier detector, which gives larger weights to high model–data misfits. Note that $r^2$, $\mu_{\mathrm{misfit}}$, $\sigma_{\mathrm{misfit}}$, and RMSD reflect the model performance with respect to evaluation data, while $\sigma_{\mathrm{ensemble}}$ measures the stability of the model best estimate $\mu_{\mathrm{ensemble}}$. Nevertheless, these different statistics should consistently reflect the skill of the model reconstruction, e.g., depending on the density and distribution of data sampling.

In the next section, both the temporal and spatial distributions of gridded SOCAT data and in situ observations, model–data errors, model best estimates and uncertainties are shown. An intensive analysis is presented for both the open ocean and the coastal zones. We then interpret key factors leading to a good or poor reconstruction of surface $p\mathrm{CO_2}$ and $fg\mathrm{CO_2}$, e.g., SOCAT data density and distribution, model design and resolution, regional to local characteristics of $p\mathrm{CO_2}$ and $fg\mathrm{CO_2}$, and their potential driving mechanisms.

## 3   Results

### 3.1   Evaluation

To verify the robustness of the mapping method, we first evaluate the goodness of fit of reconstructed $p\mathrm{CO_2}$ against the independent SOCAT data from the leave-$p$-out cross-validation set (see Sect. 2.2).

Empirical Cumulative Distribution Functions (CDFs) and frequency histograms drawn from these data are compared in Figs. 3a and 3b. While a frequency histogram in Fig. 3a shows the number of gridded SOCAT $p\mathrm{CO_2}$ data distributed for each bin, the one in Fig. 3b (grey) reflects how the $p\mathrm{CO_2}$ values in grid boxes with observations are distributed within their bounds. The probability–probability (P–P) plot of Fig. 3b (blue curve) measures the fit in the distributions of the reconstruction and SOCAT data. The same presentation is used in Figs. 3c and 3d for the misfit standard deviation $\sigma_{\mathrm{misfit}}$ and the ensemble standard deviation $\sigma_{\mathrm{ensemble}}$ (see their definitions in Eqs. 2 and 3 and their values in Figs. S3c and S3g).

The reconstructed $p\mathrm{CO_2}$ field matches SOCAT data well: both are normally distributed with the same mean of $361.3~\mu\mathrm{atm}$ (Fig. 3a) and a high agreement for all percentiles (Fig. 3b) is seen. The slight under- or overestimation at high and low percentiles implies that the model is slightly biased towards the mean value, as is expected when predictor variables do not fully explain predictand variables in the training dataset. This reduced variability is also reflected in the difference between the data standard deviation based on SOCAT $p\mathrm{CO_2}$ ($41.79~\mu\mathrm{atm}$) and the one based on CMEMS-LSCE-FFNN ($36.30~\mu\mathrm{atm}$).

Displayed on Fig. 3c, both misfit standard deviation ($\sigma_{\mathrm{misfit}}$) and model uncertainty ($\sigma_{\mathrm{ensemble}}$) empirically follow the exponential distribution. $\sigma_{\mathrm{misfit}}$ is much higher than $\sigma_{\mathrm{ensemble}}$ as the CDF and frequency histogram of the former (blue) show heavier tails than those of the latter (orange), which brings the P–P curve below the bisector in Fig. 3d. When dividing the misfit standard deviation values shown in Fig. S3c by 2, $\sigma_{\mathrm{misfit}}$ (green) shares a similar distribution as $\sigma_{\mathrm{ensemble}}$ (orange). A natural explanation for this twofold tuning factor would point to a simple lack of spread of the ensemble, either because the FFNN ensemble would be too small or because the uncertainty in the predictors (not accounted for here in the ensemble) would be significant. The SOCAT $\mathrm{CO_2}$ fugacity data are sampled at uneven space-time resolution (e.g., the sampling frequency varies

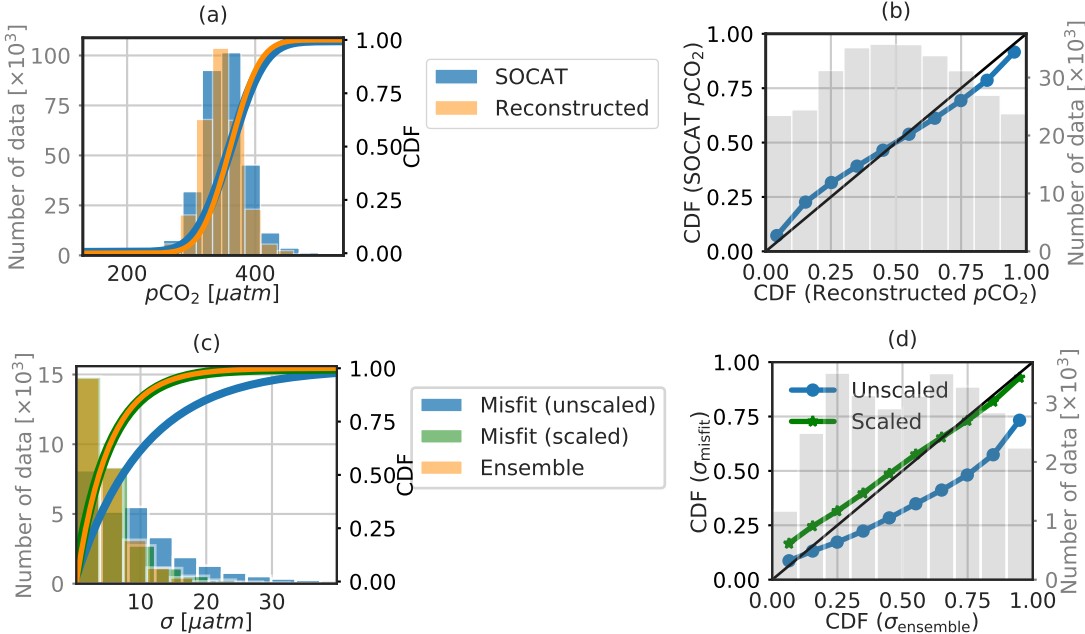

**Figure 3.** Comparison between empirical Cumulative Distribution Functions (CDFs) of (a,b) SOCATv2020 data and the reconstructed $pCO_2$ field and (c,d) model–data misfit standard deviation ($\sigma_{\mathrm{misfit}}$) and model uncertainty ($\sigma_{\mathrm{ensemble}}$), as seen in Fig. S3. In (c,d), the distribution of $\sigma_{\mathrm{misfit}}$ values scaled with a factor of 2 is plotted. A histogram with the axis in grey of the four subplots displays the number of gridded data distributed in each bin, the bins with less than 200 data for (a) and 20 data for (c) have been excluded. In (b,d), the bisector is shown in black.

between one read per minute to one per hour). Gridded data correspond to the average of measurements collected within a $1° \times 1°$ box and in a month over the entire cell area. Variability in the sampling time and location of cruises and instruments induces temporal sampling bias (e.g., towards some days in a month and/or the summer months at high latitudes) and latitude and longitude offsets from the cell center (e.g., with an average of $0.34° \pm 0.14°$ as reported in Sabine et al., 2013) which are

200 not taken into account.

Assume that

(1) Such practical imperfection presents a systematic error in each measurement from the true data with an overall standard deviation of $\sigma_{\mathrm{observation}}$.

(2) Systematic errors between SOCAT data and the reconstructed data equal those between the true data and the recon-
205 structed data.

As observation errors are independent from the random errors induced by the ensemble approach in each grid cell (further to the implementation of the leave-$p$-out cross-validation in model training; see Sect. 2.2), $\sigma_{\mathrm{misfit}}$ in Eq. (3) can be interpreted as

$$\sigma_{\mathrm{misfit}}^2 = \sigma_{\mathrm{ensemble}}^2 + \sigma_{\mathrm{observation}}^2, \tag{4}$$

where $\sigma_{\mathrm{observation}}^2$ varies in space and time and is larger near shelves (see the observation variability in Figs. S1b and S1c).

The interpretation of the magnitude of mismatch is therefore not straightforward, but we note that the spatial distribution of model errors and uncertainty estimates over the global ocean (Fig. 5) consistently identifies the spatial distribution of the model skill. This asset is prioritized in our preliminary study and further analysed in the next sections. The twofold factor used for the illustration in Fig. 3 has not been kept for the following results.

### 3.1.1 Global ocean

At global scale, the model fits the data with a mean bias close to zero, an RMSD of $20.48$ $\mu$atm, and a coefficient of determination ($r^2$) of 0.76. The temporal fluctuation of the spatial mean of the model–data mean difference over the global ocean is displayed on Fig. 4a along with the number of available gridded data. The time series of the yearly bias (black curve) starts with a large positive value ($7.47 \pm 1.60$ $\mu$atm) in year 1985 ($\sim 740$ gridded data). The bias drops during the following years and fluctuates around zero from 1994 onward (the number of grid boxes containing SOCAT observations per year is generally

larger than $5000$). In general, the magnitudes of the yearly model bias and model spread are correlated with the number of observation-based data which increased greatly since the 1990s. The importance of sustained data coverage is emphasized by Fig. S4. It illustrates the fact that large model–data mismatches are frequently associated with the interruption of Voluntary Observing Ship (VOS) lines and thus with the tracking of $CO_2$ fugacity over large regions. The larger bias computed prior to the 1990s (Fig. 4a) might intuitively lead to the conclusion that model outputs are less reliable than those in the later periods. How-

ever, this global mean score is influenced by the amount and distribution of data, and consequently the increased data density does not fully explain the reconstruction skill. For instance, even with a higher number of observation-based data than that in the pre-1990s, years 2001 and 2007 stand out with strong negative biases ($-5.44 \pm 1.26$ and $-3.12 \pm 0.92$ $\mu$atm, respectively). While such a comparison between the global bias and the number of data highlights the lack of a simple relationship between the number of data and the skill of the mapping method, the ensemble spread (dark grey area) of model errors, representing the

spread of the annual mean of $pCO_2$ estimates at SOCAT grid with observations, is reduced with an exponential decay constant of $0.46 \pm 0.06$ per 1000 gridded data (Fig. 4b).

   The model scores for the open ocean over the period 1985 to 2019 are $17.87$ $\mu$atm for RMSD and 0.78 for $r^2$. The skill of this novel method, which uses only two thirds of SOCAT data for fitting each of 100 FFNN models ranks similar to those from alternative statistical reconstruction approaches (Rödenbeck et al., 2013; Landschützer et al., 2014; Gregor et al., 2019) which

have been used to complement model-based estimates of the ocean carbon sink (Friedlingstein et al., 2019, 2020).

   The CMEMS-LSCE-FFNN reconstruction over the coastal regions for the full period is roughly twice less effective than over the open ocean in terms of RMSD ($35.86$ $\mu$atm) while it shows a rather good fit with $r^2 = 0.70$. The high RMSD reflects local high model errors along the continental shelves (Fig. S3). For the 1998–2015 period, the CMEMS-LSCE-FFNN

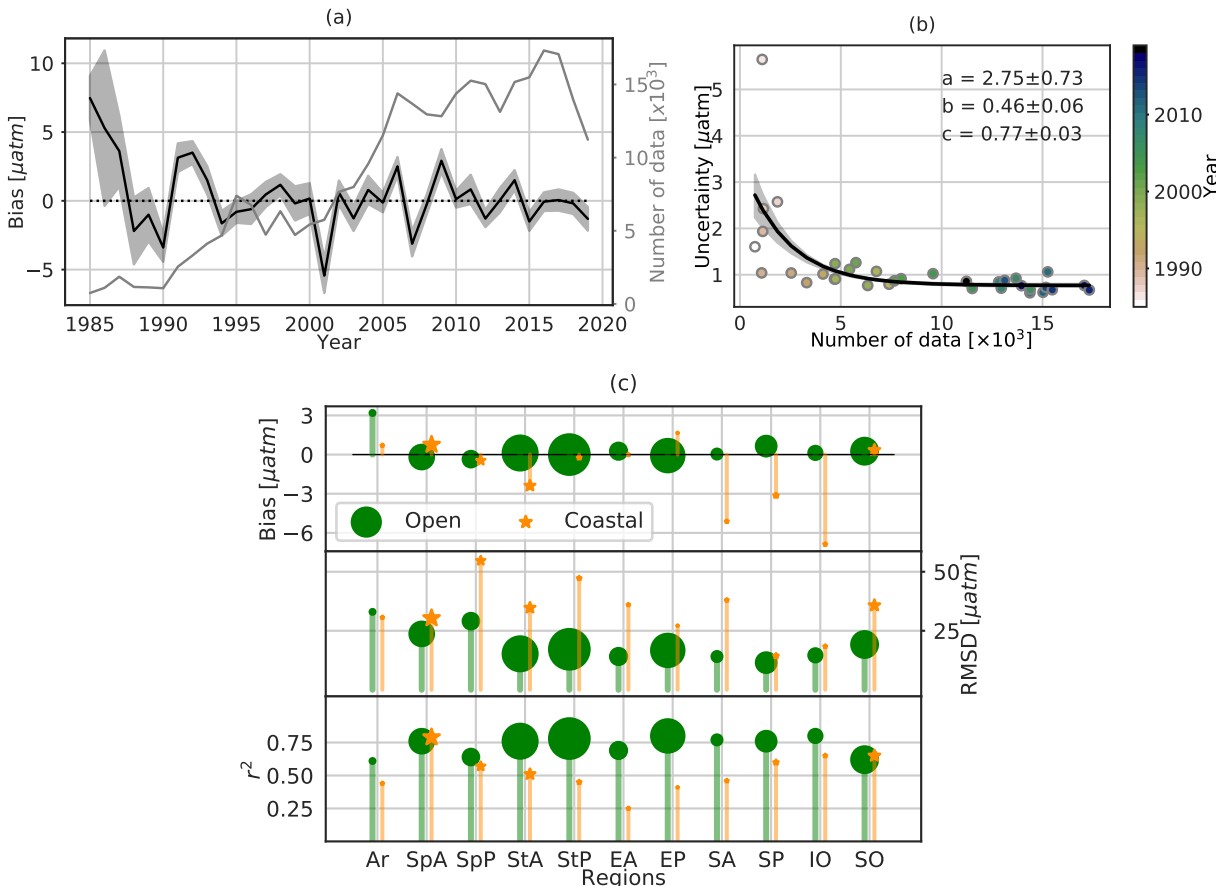

**Figure 4.** (a) Time series of the yearly mean model bias, i.e., the reconstructed $pCO_2$ data minus SOCATv2020 data, over the global ocean. The black curve and dark grey area represent the mean estimate and $1\sigma$-envelop of errors of the 100-member ensemble, the light grey curve represents the total number of gridded SOCAT data used in the FFNN model construction. (b) Exponential fits of the model uncertainty (the magnitude of the $1\sigma$-envelop in Fig. 4a) against the number of gridded data per year. The exponential function is $y = a \exp^{-bx} + c$. The black curve is derived from the best fit and the grey shaded area corresponds to the spread derived from standard errors of parameter estimates. (c) Statistical scores for 11 oceanic regions with the size of each scattered object proportional to the number of regional data (Table S2).

approach scored an RMSD of 35.84 $\mu$atm while a recent coastal reconstruction by Landschützer et al. (2020) obtained an error of 26.8 $\mu$atm (see their Table 1). The latter presents a global ocean $pCO_2$ climatology product by unifying data over the same period from two conceptually equivalent reconstruction models: one covering the open ocean at a $1° \times 1°$ resolution (Landschützer et al., 2016) and one targeting the coastal ocean at a $0.25° \times 0.25°$ resolution (Laruelle et al., 2017). These heretofore reconstructions cover the coastal region with a broader definition (400 km distance from the sea shore) than the MARCATS mask used in this study leading to the differences in characteristics and numbers of evaluation data of $pCO_2$. In addition, the CMEMS-LSCE-FFNN model was designed with the leave-$p$-out cross-validation approach excluding much more

independent data from monthly model fitting for model evaluation than in the previous models. Overall model errors remain high despite the increase in spatial resolution and in the number of observations. Coastal and shelf seas are characterized by complex physical and biological dynamics leading to high variability at small scales. For instance, $pCO_2$ levels over the Californian shelf can exceed $850$ $\mu$atm and with a spatial gradient of $pCO_2$ as large as $470$ $\mu$atm over a distance less than 0.5 km (Chavez et al., 2018; Feely et al., 2008). Clearly, further model improvement is needed in order to capture such high spatial and temporal variability of surface ocean $pCO_2$ present in observations (see also in Bakker et al., 2016; Laruelle et al., 2017, and references therein).

In the following subsections, we present and discuss the reconstruction skills for different ocean regions, as well as for open ocean and coastal domains (Fig. 4c). Complete results including the numbers of gridded data, RMSDs, and $r^2$ for each region are summarized in Table S2.

### 3.1.2 Ocean basins

#### 3.1.2.1 Arctic

Data coverage is particularly sparse over the Arctic ocean (Ar) with 50 to 220 grid boxes with observations per year since 2007 and an interruption in 2010 (Fig. S4). While continental shelves account for $50\%$ of the region's area, only one third of the observation-based data are from coastal regions. Moreover, observations are seasonally biased towards ice-free summer months (Bakker et al., 2016). Though reconstruction standard errors are similar for open basins and coastal regions (RMSDs of 33.01 and 30.65 $\mu$atm respectively), the coefficient of determination is higher over the open ocean ($r^2 = 0.61$) compared to coastal seas ($r^2 = 0.44$), suggesting a higher model skill over open basins. The close-to-zero bias of the coastal reconstruction shown in Fig. 4c results from the compensation between highly positive and negative values over the continental shelves of Alaska, the Canadian Archipelagos, the Barents and Kara Seas (see Fig. S3), the yearly bias fluctuates within $[-50, 30]$ $\mu$atm (Fig. S4). Of all open ocean regions, the Arctic reconstruction has the highest bias (3.19 $\mu$atm). Cold Arctic waters are characterized by low levels of surface ocean $pCO_2$ due to the temperature effect on $CO_2$ solubility and the seasonal draw-down of dissolved inorganic carbon (DIC) during summer months by intense biological production (Feely et al., 2001; Takahashi et al., 2009; Arrigo et al., 2010). Assuming that the Arctic predictors remain within the range of global relationships, the overestimation of $pCO_2$ by CMEMS-LSCE-FFNN, as seen in Fig. 4c, suggests a possible underestimation of biological productivity. While the preceding remains conjectural, we acknowledge a large uncertainty on the contribution of biological activity (net primary production, NPP) on surface ocean $pCO_2$, as it is "proxied" by chlorophyll-a derived from remote sensing (Maritorena et al., 2010; Babin et al., 2015b). Overall, these scores point to the Arctic as a relatively poorly reconstructed region.

#### 3.1.2.2 Atlantic

The North Atlantic stands out as a region with high data coverage (Fig. S1a) and a rapidly increasing number of data since 2000 (Fig. S4). A sustained sampling effort adds between 2000 to 4000 data each year to the database over the Subtropical (StA) and Subpolar Atlantic (SpA) regions (including between $10 - 40\%$ of coastal data). The data density

over the North Atlantic stands in strong contrast to the often less than $1000$ gridded data per year collected over the Equatorial (EA) and South Atlantic (SA) and their strong year-to-year variability.

The comparison between the reconstructed open ocean $pCO_2$ and evaluation data over the four sub-regions of the open Atlantic (Fig. 4c and Table S2) reveals small mean model–data differences, which together with the two other scores, identify the Atlantic as the basin with the highest reconstruction skill. RMSDs corresponding to the StA, the EA, and the SA are below $15.50$ $\mu$atm and $r^2$ values are in the range of $[0.69, 0.77]$. While a larger RMSD is obtained over the SpA ($23.68$ $\mu$atm), the $r^2$ of $0.76$ falls close to the upper end of the range determined for the three other regions. As discussed in Schuster et al. (2013), large temporal and spatial gradients of $pCO_2$ as well as its variability driven by a diversity of physical and biological processes (e.g., surface ocean temperature gradients, biological production, vertical mixing, and horizontal advection of water masses) keep the analysis of $pCO_2$ over the SpA challenging.

Despite accounting for over $59\%$ of the total of coastal data, skillful data reconstruction over the coastal Atlantic regions remains difficult. RMSDs are in general above $30$ $\mu$atm and, with the exception of the coastal SpA ($r^2 = 0.79$), below $51\%$ of the observed variance is predicted by the model over the other regions (StA: 0.51, EA: 0.25, SA: 0.46). The large model–data mismatch along the Atlantic continental shelves (Fig. S3) reflects the poor reconstruction of $pCO_2$ over regions under the influence of upwelling systems (e.g., Moroccan coast, Benguela), large river discharges (e.g., Amazon, Congo, Florida, Mississippi), and the bottle necks of gulfs or bays (e.g., Bahamas, English Channel).

### 3.1.2.3  Pacific

With the exception of the Subpolar Pacific (SpP), the number of observations has increased regularly over the Pacific basin. In the recent years, there are from $1000$ to $3500$ grid boxes with observations recorded over the Subtropical Pacific (StP), the Equatorial Pacific (EP), and the South Pacific (SP) (Fig. S4). Forty percent of total open ocean data belong to the StP and the EP in the years 1985–2019. Corresponding RMSDs are $17.15$ and $16.68$ $\mu$atm, with $r^2$ above $0.78$. Despite a data coverage below one third of that reported for the two previous regions, the model proved skillful in reconstructing $pCO_2$ over the SP (Fig. 4c) with RMSD $= 11.50$ $\mu$atm and $r^2 = 0.76$.

The overall good performance of the FFNN over these three Pacific sub-regions contrasts with its lack of skill over the open SpP. The data density is poor and highly variable. Before 1994, less than $250$ gridded data per year are available to constrain the reconstruction, followed by several years of intense effort and a maximum of about $1250$ data in 2000, before decreasing again to the pre-1994 values. At first order, skill scores fluctuate in line with data density. During the first period (up to 1994), the bias varies within $[-25, 25]$ $\mu$atm (Fig. S4), it decreases close to $[-2, 4]$ $\mu$atm between 1997 and 2000, and increases again along with decreasing data density. Much like the SpA, the SpP is a region characterized by a strong spatial and temporal variability in $pCO_2$ (Ishii et al., 2014), challenging any reconstruction method. The difficulty is further aggravated by the paucity of data in this region compared to the SpA. Skill scores are modest over the SpP with an RMSD of $29.08$ $\mu$atm and $r^2$ of $0.64$ (Fig. 4c and Table S2).

The ratio between coastal and open ocean observation-based data is $1 : 24$. The paucity of data for the coastal domain is reflected by lower skill scores compared to the open ocean. Over the coastal SpP, for example, the RMSD amounts to

54.69 $\mu$atm, while it is 29.08 $\mu$atm for the corresponding open ocean region. Comparable to the SpP, data reconstruction over the coastal regions of the StP (e.g., North American coast, Sea of Japan), as well as over the western EP (e.g., Peruvian upwelling) and the SP (e.g., offshore Chile) remains difficult (Fig. S3). Similar results have been found by Landschützer et al. (2020).

The EP is characterized by strong equatorial upwelling making it one of the major outgassing regions of $CO_2$ (Feely et al., 2001). Surface ocean $pCO_2$ shows a strong interannual variability predominantly in response to the El Niño Southern Oscillation (ENSO), the dominant regional climate mode (Rödenbeck et al., 2015; Landschützer et al., 2016; Denvil-Sommer et al., 2019). Before the 2000s, negative [positive] peaks of bias (Fig. S4) coincide with La Niña years; e.g., 1988–1990, 1995–1996, 1999–2001 [El Niño; e.g., 1986–1987, 1991–1992, 1997–1998] (see the ENSO events highlighted in Fig. 9). A strong negative bias is again computed in 2010–2012 which could reflect the lack of data during that cooling phase. On the contrary, the reconstruction seems less sensitive to the strong warm anomalies associated with the 2015–2016 El Niño. The model appears to be more efficient at reconstructing surface ocean $pCO_2$ during the hot climate mode (El Niño) than during the cool one (La Niña) when enhanced upwelling drives surface ocean $pCO_2$ up and towards unusual large values. This allows us to anticipate the effect of a general decrease in data collection and processing since 2020 in response to the Coronavirus disease 2019 (COVID-19) pandemic on the estimation of the ocean carbon sink. We expect a high negative bias in model estimates of $pCO_2$ and the consequent underestimation of $CO_2$ outgassing due to the combined impact of data decreasing and La Niña conditions governing since August/September 2020 (https://public.wmo.int/en/media/press-release/la-nina-has-developed). It is worthwhile to also note that monthly gridded SOCAT data in the eastern EP have declined in the last five years compared to the other years in the 2010s.

### 3.1.2.4 Indian Ocean

The Indian Ocean (IO) is the third largest oceanic regions by area but also the one with the lowest data density. With the exception of the year 1995 (approximately 1900 grid boxes including observations), as few as 500 gridded data have been provided per year (Fig. S4), yielding a total number of data often below 10 per grid cell for the entire reconstruction period (Fig. S1a). There have been even less than 75 grid boxes with observations per year over the continental shelf. However, the reconstruction over the coastal region is comparable to the open IO with a low RMSD ($< 19$ $\mu$atm) and a high correlation to the observation-based data ($r^2 = 0.65$). The overall negative bias shown in Fig. 4c for the coastal IO points to the model underestimating coastal $pCO_2$ levels. Large errors are distributed along the western Arabian Sea, the western Madagascar, and the tropical eastern IO (Fig. S3). These regions are under the influence of the southwest monsoon giving rise to a seasonal upwelling regime (see Feely et al., 2001; Sabine et al., 2002; Sarma et al., 2013, and references therein). Strong seasonal upwelling results in a marked seasonal cycle of surface ocean $pCO_2$ with high levels during the upwelling season. The paucity of data is likely to limit the skill of the model reconstruction of the seasonal cycle over large parts of the IO with consequences for the annual mean analyzed here.

### 3.1.2.5 Southern Ocean

Up to recently, data coverage over the Southern Ocean (SO) has been sparse (Fig. S1a), irregular at grid cell scale, and biased towards Austral summer months (e.g. Bushinsky et al., 2019; Gregor et al., 2019). A strong sampling effort allowed a recent increase in observations to reach up to 2000 gridded data per year (Fig. S4). Model scores for the open, respectively the coastal ocean are: RMSDs of $19.18$ $\mu$atm and $35.73$ $\mu$atm, as well as $r^2$ of $0.62$ and $0.65$. The reconstruction lacks skill over the continental shelves of South America and Antarctica (see Fig. S3).

In general, the $p\text{CO}_2$ reconstruction over the SO has less skill compared to the Atlantic or the Pacific due to the paucity in observation-based data compared to its large area. Rödenbeck et al. (2015) reported inconsistent reconstructed inter-annual variability of $p\text{CO}_2$ between different data-based methods. The interannual variability is large due to the natural variability of the coupled ocean-atmosphere system characterized by one of the globe's strongest ocean current, strong winds, vertical mixing and upwelling of DIC rich deep waters (Gregor et al., 2018; Gruber et al., 2019). Efforts to improve $p\text{CO}_2$ reconstruction are ongoing and include model development (e.g., Gregor et al., 2017), as well as the increase in data coverage by the addition of data from different sampling platforms (e.g., profiling floats, Bushinsky et al., 2019). For the time being, CMEMS-LSCE-FFNN stands out as one of the skillful models with respect to observation-based data in the SO (Friedlingstein et al., 2020; Hauck et al., 2020).

### 3.1.3 Time series stations

CMEMS-FFNN-LSCE estimates of $p\text{CO}_2$ are now compared with moored $p\text{CO}_2$ time series provided by Sutton et al. (2019). This data product comprises $p\text{CO}_2$ measurements collected from a wide range of oceanic regions since 2004 (Figs. S5–S8). Most of the stations were established in the North Atlantic and the North and Equatorial Pacific, one site is in the IO, and another in the SO. Approximately one third of Sutton et al. (2019) sites belong to the coastal seas and shelves (Fig. S8). Table S3 details the information of the moored $p\text{CO}_2$ time series.

Observation-based data used for model–data comparison (black points in Figs S6–S8) are monthly averages of $p\text{CO}_2$ measurements at each site. This interpolation results in monthly time series with a number of data $N$ between 9 (NH10) and 98 (WHOTS). The ensemble mean $\mu_{\text{ensemble}}$ and ensemble spread $\sigma_{\text{ensemble}}$ (Eq. 2) are computed from the CMEMS-LSCE-FFNN ensemble of model outputs at the four nearest model grid boxes of each location. Results confirm a reasonably good reconstruction of the proposed approach. The model best estimates (coloured thick lines) characterise $p\text{CO}_2$ trends and variations of in situ data well and the model ensembles almost catch the observation-based data in their $99\%$ confidence interval (light shaded envelop). Over $90\%$ of the time series stations, the model estimation obtains a moderate to high coefficient of determination $r^2$ with a linear model–data correlation $r$ larger than $0.5$ (e.g., BTM: 0.98, CRESCENTREEF: 0.92, HOGREEF: 0.84, SOFS: 0.79, TAO110W: 0.75, WHOTS: 0.73). Mean bias $\mu_{\text{misfit}}$ (Eq. 3) and RMSD (Eq. 4) are relatively low compared to mean $p\text{CO}_2$ values of the time series stations.

Half of the open-ocean reconstructions have model errors less than 20 $\mu$atm and even less than 10 $\mu$atm at KEO, PAPA, SOLS, STRATUS, and WHOTS (Figs S6 and S7). Despite less skill than the open-ocean reconstructions, the coastal-ocean reconstructions are quite compatible with the in situ data (Fig. S8). Most of RMSDs remain lower than $20\%$ of the mean $p\text{CO}_2$

values of coastal time series (e.g., CCE2: 36.53 $\mu$atm, ICELAND: 12.26 $\mu$atm, M2: 36.58 $\mu$atm). For some other stations in the US west coast and the oceanic regimes of coral reef, the estimates differ from the observation-based data in terms of magnitude of $pCO_2$ (e.g., CRIMP2, LA PARGUERA) and/or of its seasonal cycle (e.g., CHABA, CHEECAROCKS, SEAK).

The reconstructed time series cover the full period 1985-2019 while observation-based data are still sparse and almost distributed over the past two decades (Figs. S6-S8). The CMEMS-LSCE-FFNN time series would be useful for estimating and assessing long-term means, trends, and variations of $CO_2$ surface partial pressure and the corresponding air-sea fluxes.

## 3.2 Long term mean and uncertainty estimates

Fig. 5 shows temporal mean estimates, their associated uncertainty, and RMSDs of the monthly air–sea $pCO_2$ gradient ($\Delta pCO_2$) and $CO_2$ fluxes ($fgCO_2$) over the full period (see also Fig. S9 for the coastal regions only). In the top maps, the regions in blue are dominant $CO_2$ uptake regions (influxes) and the regions in red are dominant source regions of $CO_2$ to the atmosphere (effluxes). The uncertainty of $\Delta pCO_2$ is merely computed from the ensemble of the reconstructed sea surface $pCO_2$ since the randomness in the atmospheric $pCO_2$ field is assumed to be negligible. Due to impacts of wind stress, solubility of $CO_2$, and seasonal sea–ice coverage on the gas transfer coefficient, spatial distributions of mean estimates, their uncertainty, and RMSDs of $\Delta pCO_2$ (Figs. 5a, 5c, 5e) and $fgCO_2$ (Figs. 5b, 5d, 5f) differ from low to high values. The means of air–sea fluxes integrated/averaged over different RECCAP1 regions (Table 1) are shown in Fig. 6. The distribution of uncertainty estimates and numbers of gridded SOCAT data for these regions is also displayed on Fig. 7, wherein only values smaller than 90%-quantile of uncertainty estimates shown in Figs. 5c and 5d are plotted to reduce the effects of outliers on data visualization. The seasonal average computed over the full reconstruction period of air–sea $CO_2$ fluxes over the global ocean is shown in Fig. 8.

### 3.2.1 Arctic

The Arctic ocean stands out as the region with the strongest $CO_2$ uptake per unit area with $2.336 \pm 0.104$ molC m$^{-2}$yr$^{-1}$ for the open sea and $1.522 \pm 0.108$ molC m$^{-2}$yr$^{-1}$ for the continental shelf margins (Figs. 5b and 6b). At the scale of grid cells, air–sea gradients of $pCO_2$ are large but the downward fluxes are relatively modest over the shelves of the eastern Greenland, the Barents and Kara Seas, and the Siberia Seas (Fig. 5 or S9). During the sea–ice covered seasons, these coastal regions are neutral while the open ocean Arctic sectors (e.g., the Norwegian Sea, the Barents Sea, the Kara Seas) are $CO_2$ sinks with moderate influx densities (Fig. 8). The open ocean influx density exceeds 3 molC m$^{-2}$yr$^{-1}$ in the Arctic summer. This substantial amount of $CO_2$ uptake is driven by low surface ocean temperature, seasonal changes in sea–ice cover, and intense biological production. Increasing light availability and input of nutrients through melt waters and river discharges sustain high levels of primary production and $CO_2$ drawdown (Bates and Mathis, 2009; Arrigo et al., 2010; Yasunaka et al., 2016, 2018). Notwithstanding, the Arctic ocean represents roughly $0.58\%$ of the total surface ocean area (Table 1) and the yearly mean $CO_2$ uptake integrated over the Arctic for the full period amounts to only $1.64\%$ of the global ocean sink (Table 2 and Fig. 6a).

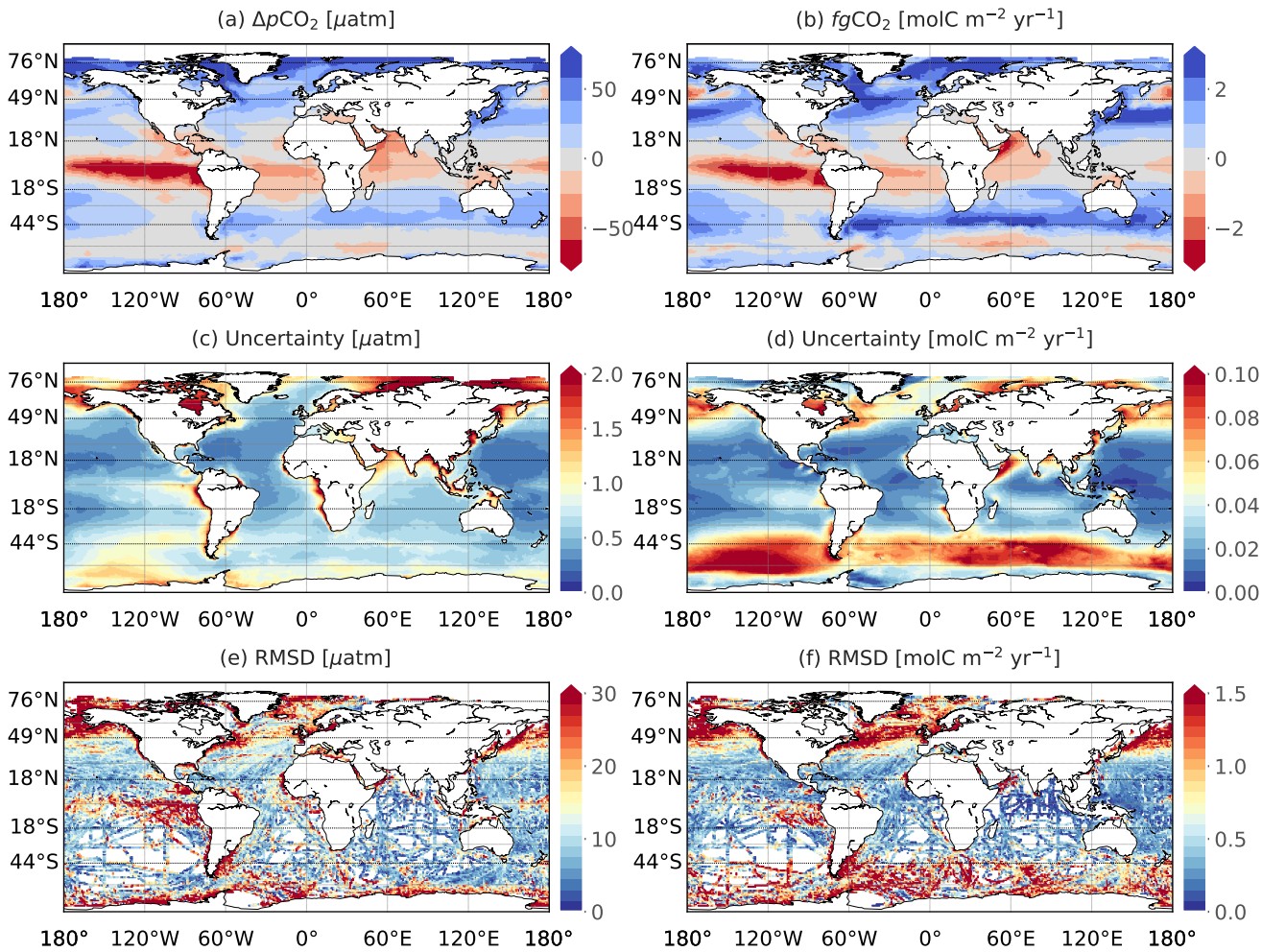

**Figure 5.** Climatological mean (top) and uncertainty (middle) of air–sea $pCO_2$ difference (a, c) and of $CO_2$ fluxes (b, d) over 1985–2019. Uncertainty (Eq. 2) is computed as the standard deviation of the 100-member CMEMS-LSCE-FFNN model outputs of sea surface $pCO_2$ and air–sea $CO_2$ fluxes. The bottom plots (e, f) show RMSDs (Eq. 4) between the SOCAT data (or data-based estimates of fluxes for (f)) and the mean CMEMS-LSCE-FFNN model outputs.

### 3.2.2 Atlantic

The open ocean Subpolar Atlantic (SpA) sink contributes approximately $78\%$ to the total SpA annual C uptake ($0.259 \pm 0.011$ $\mathrm{PgC\,yr^{-1}}$), as well as with $12.29\%$ to the total ocean sink ($1.643 \pm 0.125$ $\mathrm{PgC\,yr^{-1}}$, Table 2). Per unit area, the open ocean influx amounts to $2.012 \pm 0.092$ $\mathrm{molC\,m^{-2}yr^{-1}}$, the coastal ocean influx is $30.51\%$ less than its open ocean counterpart and slightly lower than the coastal Arctic sink (Fig. 6b). However, when integrated over the region, the yearly uptake of $0.057 \pm 0.004$ $\mathrm{PgC\,yr^{-1}}$ makes the coastal SpA the strongest sink among the 11 coastal regions (Fig. 6a). The interplay between

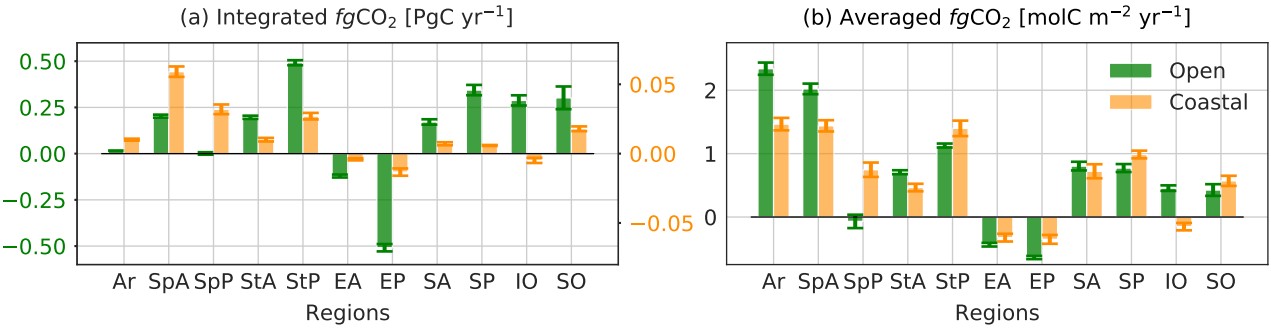

**Figure 6.** Distribution of contemporary fluxes (positive into the ocean) over 11 regions (see in Fig. 2) for the full period 1985–2019. Uncertainties of the mean estimates of air–sea fluxes integrated (a) or averaged (b) over each region are shown with error bars.

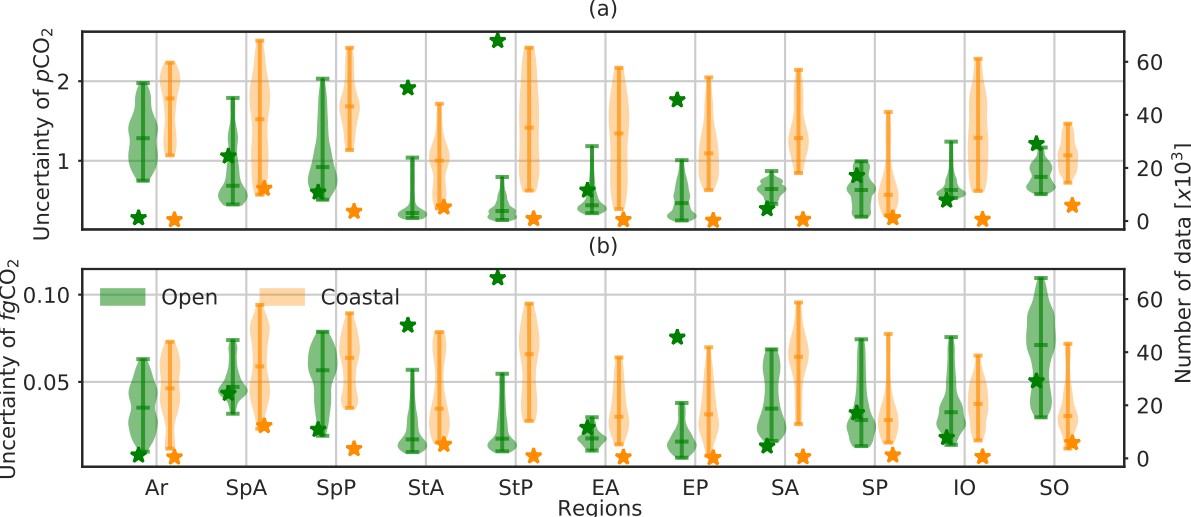

**Figure 7.** Distribution (violin) of all uncertainty estimates (Figs. 5c and 5d) and the total number (star) of gridded SOCAT data (Fig. S1a) split for 11 RECCAP1 regions. A violin plot shows the range, median, and density of uncertainty estimates for $pCO_2$ ($\mu$atm) and $fgCO_2$ ($\mathrm{molC\,m^{-2}\,yr^{-1}}$).

temperature- and biology driven effects results in changes in the seasonal and spatial distributions of surface ocean $pCO_2$ and ultimately air–sea $CO_2$ fluxes. During boreal winter/spring, high wind speeds enhance gas transfer velocities, contribute to a strong cooling and an increase of $CO_2$ solubility (Takahashi et al., 2009; Feely et al., 2001), both enhancing uptake of $CO_2$ over the Labrador Sea, the North Atlantic and Norwegian Currents, the Barents and Kara Seas (Fig. 8). High wind speeds also strengthen vertical mixing, a process supplying dissolved inorganic carbon (DIC) and nutrients to the surface ocean. During the spring and summer months, a vigorous biological activity (Sigman and Hain, 2012) counteracts the warming induced decrease in $CO_2$ solubility and increase in $pCO_2$ by drawing down DIC (Feely et al., 2001). Along the coast of the Barents and Kara

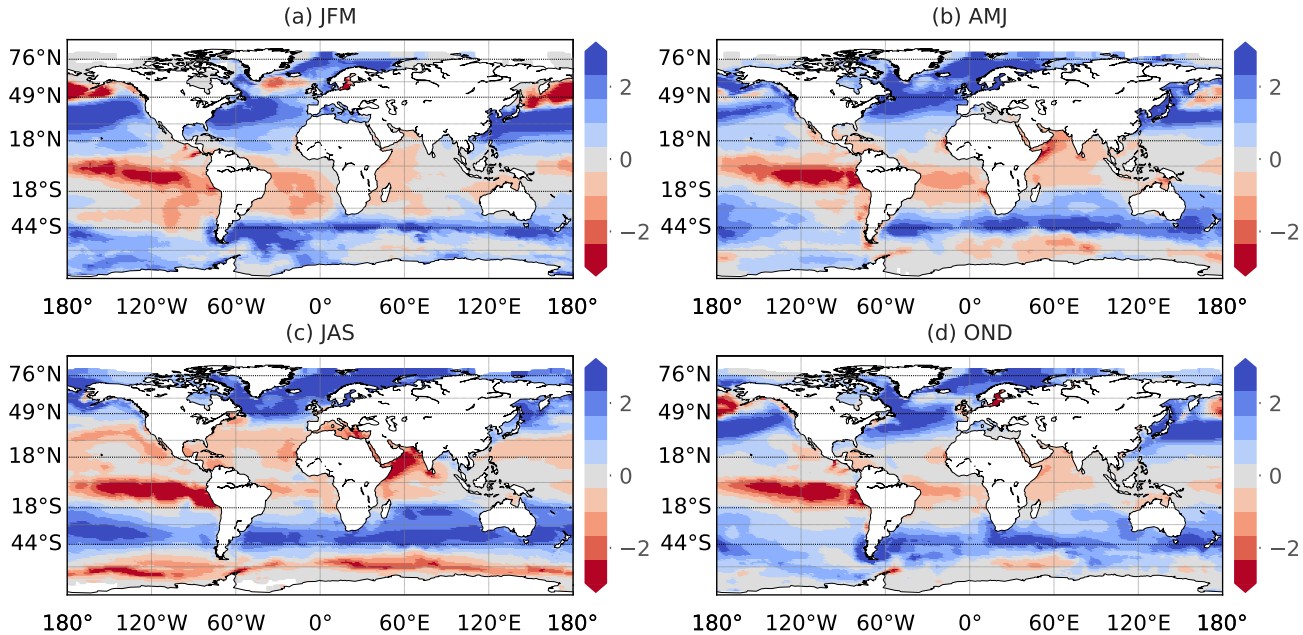

**Figure 8.** Seasonality of downward $CO_2$ fluxes $[\mathrm{molC\,m^{-2}yr^{-1}}]$ in 1985–2019. Temporal means of the reconstructed $fgCO_2$ field for January to March (JFM), April to June (AMJ), July to September (JAS) and, October to December (OND) are shown.

Seas, inputs of fresh water (decrease in salinity and increase in $CO_2$ solubility) and nutrients (biological activity and DIC drawdown) combine to strengthen $CO_2$ uptake (Arrigo et al., 2010; Yasunaka et al., 2016, 2018; Olafsson et al., 2021). This contrasts with other coastal regions (e.g., southern North Sea and Baltic Sea) where the respiration of terrestrial particulate organic carbon from river run-off contributes to making these areas a strong seasonal source of $CO_2$ (Borgesa and Gypensb, 2010; Becker et al., 2021).

The Subtropical Atlantic (StA) is characterized by weak to moderate mean flux densities per unit area (open: $0.733\pm 0.036\,\mathrm{molC\,m^{-2}yr^{-1}}$, coastal: $0.457\pm0.064\,\mathrm{molC\,m^{-2}yr^{-1}}$). The total integrated C uptake amounts to $0.214\pm0.011\,\mathrm{PgC\,yr^{-1}}$, with $0.204\pm0.010\,\mathrm{PgC\,yr^{-1}}$ contributed by the open ocean. As for the SpA, the net uptake reflects the combined effect of cooling, mixing, and biological activity. Figures 5 and S9 show the regional distribution of sources and sinks. Regions of intense $CO_2$ uptake are associated with the warm Gulf Stream and its northeastward extension, the North Atlantic Drift. Strong uptake is also found over the western continental shelf where strong river discharges sustain high levels of biological productivity in particular during spring (Jamet et al., 2007; Kealoha et al., 2020). Weaker sinks or sources of $CO_2$ in the southwestern StA and the eastern subtropical gyre are primarily driven by high surface temperature and enhanced stratification (Schuster et al., 2013). The latter restricts the vertical supply of nutrients and limits biological production. Finally, a relatively strong source of $CO_2$ is found over the Canary upwelling system in summer (Fig. 8).

The Equatorial Atlanic (EA) stands out as the second strongest source region of $CO_2$ after the Equatorial Pacific (EP) with a yearly outgassing of $-0.117 \pm 0.009$ PgC yr$^{-1}$ (Fig. 6a). Most of $CO_2$ is released from the open ocean with an average efflux of $-0.407 \pm 0.031$ molC m$^{-2}$yr$^{-1}$ (Figs. 5b and 6b). This intense source of $CO_2$ stems from upwelling of cool and $CO_2$-rich waters in the eastern EA. A westward increase in outgassing is observed along with the advection of $CO_2$-rich waters (Schuster et al., 2013). The coastal EA regions release an average of $-0.288 \pm 0.064$ molC m$^{-2}$yr$^{-1}$ of $CO_2$. Over large areas, the opposing effects of primary production and high surface temperature combine to weaken the coastal sink or seasonally switch it from a weak to a moderate source (e.g., the north east EA, Caribbean Sea, Venezuelan and Guiana basins, Gulf of Guinea) (Fig. 8). The Amazon river is a notable exception. Its large discharges of fresh water, nutrients, as well as of dissolved and particulate carbon turn the coastal and adjacent shelf seas into a net sink of $CO_2$ (Medeiros et al., 2015; Ibánhez et al., 2015).

The South Atlantic (SA) uptake amounts to $0.192 \pm 0.016$ PgC yr$^{-1}$. Regions north of $30°S$ act as weak sources or are neutral with respect to air–sea exchanges of $CO_2$, as opposed to regions to the South which are significant sinks of $CO_2$ (Fig. 5b). For the full period, densities over the open and coastal regions are, respectively, $0.862 \pm 0.072$ molC m$^{-2}$yr$^{-1}$ and $0.776 \pm 0.125$ molC m$^{-2}$yr$^{-1}$. Coastal regions are changing from moderate sources to sinks with increasing latitude (Fig. S9). The SA has similar seasonal dynamics as the StA with $CO_2$ uptake in winter and outgassing in summer (Takahashi et al., 2009; Schuster et al., 2013). During the austral winter, deep mixed layers result in cold surface waters which absorb $CO_2$ from the atmosphere. By contrast, warming during the summer reduces the solubility of $CO_2$ leading to a weak sink or even a source (Fig. 8). As explained before, biological production counteracts the effect of warming and the vigorous spring bloom contributes to the uptake south of $30°S$ (Sigman and Hain, 2012; Carvalho et al., 2020).

### 3.2.3 Pacific

The Subpolar Pacific (SpP) is the second smallest region by area (2.78% of the total surface ocean area) and with $0.040 \pm 0.010$ PgC yr$^{-1}$ (net coastal and open ocean sinks) provides the smallest contribution to the total yearly ocean C uptake (Table 2 and Fig. 6a). The coastal ocean contributes about $0.032 \pm 0.004$ PgC yr$^{-1}$ to the total yearly C uptake, making the SpP the only region for which coastal fluxes exceed open ocean fluxes. The strength of its coastal C sink ranks second among all coastal regions (Fig. 6a). Seasonal features of $CO_2$ fluxes are shown in Fig. 8. The SpP is ice-covered during the winter months which results in close to zero air–sea fluxes per unit area north of $60°N$ (e.g., Beaufort, Siberia, and Chukchi Seas). Besides, vertical convection during winter brings up DIC-rich old waters leading to $CO_2$ outgassing exceeding $-3$ molC m$^{-2}$yr$^{-1}$ in the South of the region (Bates and Mathis, 2009; Arrigo et al., 2010; Ishii et al., 2014; Yasunaka et al., 2016). An intense biological production during the boreal summer drives an intense uptake of $CO_2$ over the entire SpP (Feely et al., 2001; Sigman and Hain, 2012; Ishii et al., 2014). The interplay of these two seasonal mechanisms and their opposing effects make the open SpP a weak yearly net sink (Fig. 6). The average flux density per unit area is $0.044 \pm 0.123$ molC m$^{-2}$yr$^{-1}$ over the open ocean, much smaller than the value determined for the coastal ocean of $0.775 \pm 0.127$ molC m$^{-2}$yr$^{-1}$ (Fig. 6b). As shown in Bates (2006), Arrigo et al. (2010), and Ishii et al. (2014), surface DIC concentration is higher over the open, deep basins than the shallow coastal seas of the SpP, particularly induced by deep mixing during winter/spring. Over the same period, seasonal

sea-ice also restricts gas exchanging, the coastal sector thus acts as a neutral region of $CO_2$ fluxes (Fig. 8). During spring and summer, a substantial amount of $CO_2$ is also absorbed in the coastal shelf seas influenced by high biological production in large ice-free areas (e.g., Chukchi and Gulf of Alaska), and/or by dilution of sea waters from river freshwater with low salinity and DIC concentration (e.g., Beaufort, Laptev, and East Siberia Seas) (Arrigo et al., 2010; Yasunaka et al., 2016, 2018).

A total mean uptake of $0.523 \pm 0.016$ $PgC\,yr^{-1}$ makes the Subtropical Pacific (StP) the largest sink region. The open ocean contribution dominates the regional sink with $0.495 \pm 0.015$ $PgC\,yr^{-1}$ (Table 2 and Fig. 6a). The corresponding mean flux density per unit area is $1.136 \pm 0.036$ $molC\,m^{-2}yr^{-1}$ (Fig. 6b) and makes the StP rank third after the open ocean Arctic and SpA regions. As discussed for the StA, during winter months cooling and high wind intensities along the Kuroshio and North Pacific Currents enhance the uptake of $CO_2$ (Takahashi et al., 2009; Ishii et al., 2014). By contrast, summer warming drives the StP towards close to neutral conditions, respectively a weak source (Fig. 8). With a yearly mean uptake of $0.028 \pm 0.003$ $PgC\,yr^{-1}$, the coastal StP sink becomes third in terms of intensity among the coastal sinks (Fig. 6a). The influx density is $1.444 \pm 0.130$ $molC\,m^{-2}yr^{-1}$. Western coastal systems and shelf seas are under the influence of the delivery of freshwater and nutrients by large river systems (Liu et al., 2014). The resulting intense biological production contributes to influx densities per unit area that are higher over the western continental shelf and seas (e.g., East China Sea, Sea of Japan) than over the California upwelling system (Figs. 5b, S9b, and 8).

The Equatorial Pacific (EP) is the strongest source region of $CO_2$ to the atmosphere with a yearly average efflux of $-0.490 \pm 0.021$ $PgC\,yr^{-1}$ from the open ocean, respectively $-0.013 \pm 0.003$ $PgC\,yr^{-1}$ from the continental shelves. On average per unit area, the open sea emits $-0.616 \pm 0.027$ $molC\,m^{-2}yr^{-1}$ of $CO_2$. This high rate of outgassing is a distinct feature of the EP (e.g., Feely et al., 2001; Takahashi et al., 2009; Rödenbeck et al., 2015; Landschützer et al., 2016; Denvil-Sommer et al., 2019; Landschützer et al., 2019) and is primarily due to the upwelling of DIC rich deep waters. The magnitude of $CO_2$ release decreases westward - from Eastern boundary upwelling (e.g., Peru, Panama) to the International Date line - in line with decreasing upwelling intensity, warmer sea surface temperature, and lower salinity (Ishii et al., 2014). Compared to the open EP, the efflux density of the coastal regions ($-0.334 \pm 0.071$ $molC\,m^{-2}yr^{-1}$) is roughly half that of the open ocean.

The South Pacific (SP) ranks second as a sink region for $CO_2$ with a yearly net flux of $0.358 \pm 0.029$ $PgC\,yr^{-1}$, mostly contributed by the open ocean (Fig. 6a). Uptake rates per unit area are very similar to those obtained for the SA (Fig. 6b). A detailed assessment reveals the open ocean influx density to be slightly lower ($0.791 \pm 0.066$ $molC\,m^{-2}yr^{-1}$), respectively the coastal one to be slightly higher ($0.987 \pm 0.063$ $molC\,m^{-2}yr^{-1}$) over the SP compared to the SA. Due to the larger area of the SP (Table 1), its integrated sink is approximately twice that of the SA. Similar to the processes discussed above for the SA, vertical mixing drives the uptake of $CO_2$ during austral winter (Takahashi et al., 2009; Ishii et al., 2014) and the effect of warming on $CO_2$ solubility during spring and summer is off-set by biological production. The latter leads to moderate to high uptake of $CO_2$ over the coasts and the southwest open sea (e.g., Eastern Australian Currents, Southern Australia, New Zealand) (Fig. 8). The influx density decreases eastward under the influence of the strong upwelling of DIC driven by the Peru Current.

### 3.2.4 Indian Ocean

The total integrated Indian Ocean (IO) sink is evaluated to $0.300 \pm 0.033\,\mathrm{PgC\,yr^{-1}}$, with $0.305 \pm 0.033\,\mathrm{PgC\,yr^{-1}}$ contributed by the open ocean and a weak coastal source of $-0.004 \pm 0.002\,\mathrm{PgC\,yr^{-1}}$. The spatial distribution of flux densities (Fig. 5b) reveals the northwestern IO to be a net source of $CO_2$ to the atmosphere, while the northeastern IO is close to neutral and latitudes south of $18°S$ act as a strong sink. This regional compensation leads to a small open ocean influx density per unit area of $0.482 \pm 0.052\,\mathrm{molC\,m^{-2}yr^{-1}}$ and a small coastal efflux per unit area of $-0.131 \pm 0.061\,\mathrm{molC\,m^{-2}yr^{-1}}$ (Fig. 6b). The northern IO is a strong source of $CO_2$ sustained by the monsoon-driven seasonal upwelling along the Arabian and Somalian coasts (Behrenfeld et al., 2006; Sarma et al., 2013). The northeastern IO regions including the Bay of Bengal and its continental shelves receive fresh waters discharged from the Ganges river and lateral inputs from Indonesian outflows (see Sarma et al., 2013, and references therein), and switch between mild sources and sinks (Fig. 8). The Subtropical Front ($40°S$) divides the region south of $18°S$ into a weak sink to the North and over the oligotrophic gyre and a band of vigorous uptake to its South over the Subantarctic zone (SAZ) between $40°S$ and $44°S$ (Fig. 5b). Similar to the SA and SP, this entire region is identified as a significant net sink of $CO_2$ in winter (Fig. 8) possibly driven by enhanced solubility in response to cooling and mixing. While biological production maintains the sink over the SAZ during austral spring and summer months, warming reduces $CO_2$ uptake over the oligotrophic gyre.

### 3.2.5 Southern Ocean

The total Southern Ocean (SO) sink amounts to $0.349 \pm 0.070\,\mathrm{PgC\,yr^{-1}}$, including a coastal uptake of $0.018 \pm 0.002\,\mathrm{PgC\,yr^{-1}}$. The mean influx per unit area over the open SO is $0.468 \pm 0.104\,\mathrm{molC\,m^{-2}yr^{-1}}$ and close to the one obtained for the open IO (Fig. 6b). The area-averaged $CO_2$ drawdown over the coastal SO is $0.599 \pm 0.089\,\mathrm{molC\,m^{-2}yr^{-1}}$ with strong coastal sinks distributed over the South American and Antarctic shelves ($60°W$ westward as seen in Fig. 5b or S4b). During the austral spring and summer, intense phytoplankton blooms enhance the consumption of $CO_2$ over the Subantarctic and the Polar Frontal Zones between $44°$S and $58°$S (Sigman and Hain, 2012; Lenton et al., 2013), leading to a large sink with a flux density exceeding $1.667\,\mathrm{molC\,m^{-2}yr^{-1}}$ (Fig. 8). South of $58°$S, sea–ice retreat and vertical stratification contribute to a mild sink over the Antarctic Zone. During winter, vertical mixing brings DIC rich deep waters to the surface triggering a strong outgassing of $CO_2$ along the Antarctic Circumpolar Current.

**Table 2.** Yearly mean of contemporary air–sea $CO_2$ fluxes ($PgC\,yr^{-1}$) integrated over the global ocean and 11 RECCAP1 regions. Mean estimate and uncertainty ($\mu_{ensemble} \pm \sigma_{ensemble}$) of the CMEMS-LSCE-FFNN approach is shown for the coast (C), the open ocean (O), and the total area (T). For a comparison, estimates derived from RECCAP1 (Canadell et al., 2011; Schuster et al., 2013; Ishii et al., 2014; Sarma et al., 2013; Lenton et al., 2013; Wanninkhof et al., 2013) are provided. In column 'RECCAP1', values in parentheses are the 'best' estimates proposed by RECCAP1 studies which were derived from averages or medians of estimates based on the $pCO_2$ climatology or $pCO_2$ diagnostic model, and/or the atmospheric and ocean inversions, and GOBM models. The 'RECCAP1' values out of parentheses are the estimates derived from different methods mapping observation-based data of $pCO_2$. With an exception for the global estimate* (Wanninkhof et al., 2013), those of the RECCAP1 sub-basins are available only for the open ocean.

| Approach | | CMEMS-LSCE-FFNN | | RECCAP1 |
|---|---|---|---|---|
| Regions | | 1985–2019 | | 1990–2009 |
| Globe | (T) | $1.643 \pm 0.125$ | $1.486 \pm 0.114$ | |
| | (O) | $1.493 \pm 0.122$ | $1.344 \pm 0.111$ | $1.18^*$ |
| | (C) | $0.150 \pm 0.010$ | $0.141 \pm 0.009$ | $0.18^*$ |
| Arctic (Ar) | (T) | $0.027 \pm 0.001$ | $0.024 \pm 0.001$ | |
| | (O) | $0.016 \pm 0.001$ | $0.015 \pm 0.001$ | $(0.12 \pm 0.06)$ |
| | (C) | $0.011 \pm 0.001$ | $0.010 \pm 0.001$ | |
| Subpolar Atlantic (SpA) | (T) | $0.259 \pm 0.011$ | $0.255 \pm 0.010$ | $0.07 \pm 0.04, 0.30 \pm 0.13$ |
| | (O) | $0.202 \pm 0.009$ | $0.197 \pm 0.008$ | $(0.21 \pm 0.06)$ |
| | (C) | $0.057 \pm 0.004$ | $0.058 \pm 0.004$ | |
| Subtropical Atlantic (StA) | (T) | $0.214 \pm 0.011$ | $0.202 \pm 0.009$ | $0.18 \pm 0.09, 0.24 \pm 0.16$ |
| | (O) | $0.204 \pm 0.010$ | $0.192 \pm 0.009$ | $(0.26 \pm 0.06)$ |
| | (C) | $0.010 \pm 0.001$ | $0.010 \pm 0.001$ | |
| Equatorial Atlantic (EA) | (T) | $-0.117 \pm 0.009$ | $-0.128 \pm 0.008$ | $-0.10 \pm 0.05, -0.12 \pm 0.14$ |
| | (O) | $-0.113 \pm 0.009$ | $-0.123 \pm 0.008$ | $(-0.12 \pm 0.04)$ |
| | (C) | $-0.004 \pm 0.001$ | $-0.004 \pm 0.001$ | |
| South Atlantic (SA) | (T) | $0.192 \pm 0.016$ | $0.174 \pm 0.015$ | $0.25 \pm 0.12, 0.21 \pm 0.23$ |
| | (O) | $0.184 \pm 0.015$ | $0.167 \pm 0.015$ | $(0.14 \pm 0.04)$ |
| | (C) | $0.008 \pm 0.001$ | $0.007 \pm 0.001$ | |
| Subpolar Pacific (SpP) | (T) | $0.040 \pm 0.010$ | $0.029 \pm 0.009$ | |
| | (O) | $0.008 \pm 0.008$ | $-0.002 \pm 0.007$ | |
| | (C) | $0.032 \pm 0.004$ | $0.031 \pm 0.003$ | $0.44 \pm 0.21, 0.37$ |
| Subtropical Pacific (StP) | (T) | $0.523 \pm 0.016$ | $0.512 \pm 0.014$ | $(0.47 \pm 0.13)$ |
| | (O) | $0.495 \pm 0.015$ | $0.485 \pm 0.014$ | |
| | (C) | $0.028 \pm 0.003$ | $0.027 \pm 0.002$ | |
| Equatorial Pacific (EP) | (T) | $-0.503 \pm 0.022$ | $-0.514 \pm 0.020$ | $-0.51 \pm 0.24, -0.27$ |
| | (O) | $-0.490 \pm 0.021$ | $-0.500 \pm 0.020$ | $(-0.44 \pm 0.14)$ |
| | (C) | $-0.013 \pm 0.003$ | $-0.013 \pm 0.003$ | |
| South Pacific (SP) | (T) | $0.358 \pm 0.029$ | $0.343 \pm 0.029$ | $0.29 \pm 0.14, 0.24$ |
| | (O) | $0.352 \pm 0.029$ | $0.337 \pm 0.028$ | $(0.37 \pm 0.08)$ |
| | (C) | $0.006 \pm 0.0004$ | $0.006 \pm 0.0004$ | |
| Indian Ocean (IO) | (T) | $0.300 \pm 0.033$ | $0.281 \pm 0.027$ | $0.24 \pm 0.12$ |
| | (O) | $0.305 \pm 0.033$ | $0.286 \pm 0.027$ | $(0.37 \pm 0.06)$ |
| | (C) | $-0.004 \pm 0.002$ | $-0.005 \pm 0.002$ | |
| Southern Ocean (SO) | (T) | $0.349 \pm 0.070$ | $0.307 \pm 0.061$ | $0.27 \pm 0.13$ |
| | (O) | $0.330 \pm 0.069$ | $0.290 \pm 0.061$ | $(0.42 \pm 0.07)$ |
| | (C) | $0.018 \pm 0.002$ | $0.017 \pm 0.002$ | |

## 4   Discussion

### 4.1   Contemporary air–sea $CO_2$ flux estimates

Our estimates of contemporary net fluxes of $CO_2$ for the global ocean and 11 open ocean regions are compared to estimates from RECCAP1 in Table 2 after adjusting them to the same period (1990–2009). RECCAP1 best estimates were derived from averages or medians of estimates based on the $pCO_2$ climatology or $pCO_2$ diagnostic model, and/or the atmospheric and ocean inversions and GOBM models (see Schuster et al., 2013; Ishii et al., 2014; Sarma et al., 2013; Lenton et al., 2013, and references therein). The observation-based estimates of regional net fluxes reported in these studies were computed from the reconstruction of SOCAT $pCO_2$ data (only used in Schuster et al., 2013), LDEO data (https://www.ldeo.columbia.edu/res/pi/CO2/), and its climatology (Takahashi et al., 2009). With the exception of the global ocean, coastal fluxes were not part of the earlier assessment. The global open ocean uptake obtained in this study of $1.344 \pm 0.111\ \mathrm{PgC\,yr^{-1}}$ lies between the observation-based net sink estimate by Wanninkhof et al. (2013) ($1.18\ \mathrm{PgC\,yr^{-1}}$) and the global sum of regional best estimates given in Table 2 ($1.8\ \mathrm{PgC\,yr^{-1}}$). Net regional fluxes computed from CMEMS-LSCE-FFNN are mostly within the range of fluxes derived from observation-based reconstructions and multi-approach best estimates. Our Southern Ocean open ocean sink ($0.290 \pm 0.061\ \mathrm{PgC\,yr^{-1}}$) compares well with previous observation-based estimates ($0.27 \pm 0.13\ \mathrm{PgC\,yr^{-1}}$), but is lower than multi-approach best estimates ($0.42 \pm 0.07\ \mathrm{PgC\,yr^{-1}}$). A significant discrepancy between the present and previous estimates is also found over the Arctic ocean for which the regional open ocean net $CO_2$ uptake is about 1 order of magnitude lower in CMEMS-LSCE-FFNN compared to the RECCAP1 best estimate (Schuster et al., 2013).

Based on the MARCATS mask (Fig. 2), the CMEMS-LSCE-FFNN estimate of the yearly net coastal sink over the full reconstruction period is $0.150 \pm 0.010\ \mathrm{PgC\,yr^{-1}}$. For 1990–2011, we estimate a yearly net coastal sink of $0.147 \pm 0.009\ \mathrm{PgC\,yr^{-1}}$ which is lower than the one based on SOCATv2 data by Laruelle et al. (2014) ($0.19 \pm 0.05\ \mathrm{PgC\,yr^{-1}}$). Despite the fact that the present estimate was obtained with a model at a lower spatial resolution, the flux density of coastal sources and sinks, as well as their spatial distribution (Fig. S9b) are, in general, consistent with Laruelle et al. (2014) (Fig. 2) with exceptions found in Northern polar and subpolar regions. For instance, Laruelle et al. (2014) suggested the Okhotsk shelf to be a strong source of $CO_2$ in excess of $-3\ \mathrm{molC\,m^{-2}yr^{-1}}$. To the contrary and in line with Otsuki et al. (2003), it is identified as a significant sink in this study taking up 1 to $2.333\ \mathrm{molC\,m^{-2}yr^{-1}}$ (Fig. 5).

Our estimates for the mean annual open and coastal ocean uptake over the Arctic ($> 76°N$) are $0.015 \pm 0.001\ \mathrm{PgC\,yr^{-1}}$ and $0.010 \pm 0.001\ \mathrm{PgC\,yr^{-1}}$ (Table 2) which are respectively less than the best estimate of $0.12 \pm 0.06\ \mathrm{PgC\,yr^{-1}}$ by Schuster et al. (2013) and that of $0.07\ \mathrm{PgC\,yr^{-1}}$ by Laruelle et al. (2014). The discrepancy is possibly due to an overestimation of Arctic $pCO_2$ by the CMEMS-LSCE-FFNN (see in Sect. 3.1.2) and to the lack of estimates over a large portion of the seasonally sea–ice covered regions (see in Figs. 5 and 8). Further improvements would include using additional products of sea surface height and input from river discharge and sea–ice melt available over the Arctic. Besides, in Eq. (1), the air–sea flux density is a linear function of the sea–ice fraction leading to $fgCO_2 = 0$ as $f_{ice} = 1$. Loose et al. (2009) suggest that the flux density in such regions is larger than evaluated by Eq. (1). A suggestion for a better assessment of air–sea fluxes over the Arctic and other

regions with sea–ice cover (i.e., Antarctic and partly subpolar regions) would be to impose a sea–ice concentration of $99\%$ for values exceeding $99\%$ (Bates et al., 2006).

## 4.2 Model errors and uncertainties

Our uncertainty evaluation for estimates of $pCO_2$ and air–sea $CO_2$ fluxes is based on a Monte Carlo approach. Statistics (i.e., ensemble standard deviation, Eq. 2) are based on ensembles of CMEMS-LSCE-FFNN model realizations. It allows producing spatially and temporally varying uncertainty fields of $pCO_2$ and $fgCO_2$ estimates covering the global ocean and the full period. This asset can be used for quantifying the uncertainty for different spatial and temporal resolutions (e.g., monthly/yearly integrated fluxes at regional/global scales).

As a complement to Fig. 3 (bottom plots) which generally evaluates the reliability of model uncertainty estimates compared to model–data misfit deviations, Fig. 5 shows some similarity between their spatial distributions for $pCO_2$ (Figs. 5c and 5e) as for $fgCO_2$ (Figs. 5d and 5f). For $pCO_2$, large model–data misfits and uncertainties are found over regions with sparse density or devoid of SOCAT data (see in Figs. S1a and S4), but also with high temporal and/or spatial $pCO_2$ variations (partly shown in Figs. S1b and S1c). High temporal/spatial gradients of $pCO_2$ are typically associated with upwelling systems (e.g., Eastern boundary upwelling systems, Arabian Sea upwelling), Western boundary currents (e.g., Gulf Stream, Kuroshio), intense biological production (e.g., spring bloom in temperate Northern/Southern latitudes), coastal and shelf dynamics including river runoff (e.g., Amazon, Congo, Mississippi, and great subpolar and Arctic rivers such as Ob, Yenisey, Lena, Mackenzie). Comparing between Figs. 5c and 5d (5e and 5f), the magnitude of the uncertainty estimates (model errors) of air–sea $CO_2$ flux estimates appears to be much less correlated to measurement density (Fig. S1) than the $pCO_2$ field (see also in Figs. 7a and 7b). The model uncertainty and errors of $fgCO_2$ estimates are highest over the open SO ($> 44°S$), the subpolar regions, the Indian gyre, and upwelling systems.

In this study, the uncertainty quantified for the reconstruction of $pCO_2$ and ultimately $fgCO_2$ is a result of randomly sampling training and validation datasets from predictors and SOCAT observation-based data for 100 FFNN model runs (see Sect. 2.2). This subsampling approach permits to take into account an assumption of uncertainties of predictors and SOCAT data, i.e., random errors exist through changes in the range between their sub-samples. For a better assessment of the reconstruction uncertainty, future studies would need to include realistic uncertainties of these data, and also of local (sub-)skin effects of temperature and salinity as suggested in Watson et al. (2020). Additional sources of uncertainty in the computation of air–sea fluxes are discussed by Wanninkhof (2014), Woolf et al. (2019), and Fay et al. (2021). These studies have demonstrated the strong impact of different wind field products and model parameterizations on the gas transfer velocity $k$ in Eq. (1) and the corresponding air–sea flux estimates. For instance, using the eight expressions for the parameterization of $k$ proposed in Woolf et al. (2019) and references therein would inflate the uncertainty of the global mean annual uptake from $5\%$ to $10\%$. However, it would not significantly impact the spatial distribution of uncertainty, but only its magnitude.

## 4.3 Quantification of the global ocean carbon sink

Table 3 presents the comparison of estimates between the CMEMS-LSCE-FFNN, an ensemble of data-based reconstruction approaches, and an ensemble of global ocean biogeochemical models (GOBMs) used in the Global Carbon Project (GCP, Friedlingstein et al., 2019, 2020; Hauck et al., 2020) for the reconstruction of air–sea $CO_2$ fluxes. The reconstructed CMEMS-LSCE-FFNN field covers approximately 88.9% of the total ocean area used by the GCP ($361.9 \times 10^6 \, \mathrm{km}^2$). The annual contemporary uptake over the global ocean and the full period 1985–2019 was $1.643 \pm 0.125 \, \mathrm{PgC \, yr^{-1}}$ with a starting net influx of $0.784 \pm 0.178 \, \mathrm{PgC \, yr^{-1}}$, a growth rate of $+0.062 \pm 0.006 \, \mathrm{PgC \, yr^{-2}}$, and an interannual variability (temporal standard deviation) of $0.526 \pm 0.022 \, \mathrm{PgC \, yr^{-1}}$ (Fig. 9). The contemporary sink amounted to $2.301 \pm 0.126 \, \mathrm{PgC \, yr^{-1}}$ for the last decade (2010s) and $2.877 \pm 0.154 \, \mathrm{PgC \, yr^{-1}}$ in the year 2019 (Table 3). The long term positive trend of the global ocean carbon sink estimates tracks the growth rate of atmospheric $CO_2$ concentration since the mid-1980s (Friedlingstein et al., 2019, 2020). The interanual to multi-annual variability of the global ocean carbon sink co-varies with cold and hot ENSO phases (Fig. 9) confirming ENSO as a leading mode of variability of the ocean carbon sink (Feely et al., 1999).

**Figure 9.** Yearly global integrated air–sea flux estimates derived from the CMEMS-LSCE-FFNN ensemble (mean ± uncertainty) for 1985–2019. Multivariate El Niño-Southern Oscillation Index (MEI; Wolter and Timlin, 1993, https://psl.noaa.gov/enso/mei/, last access: December 2020) is used to generally indicate a link between variations, e.g. Yearly uptake - Trend , in the CMEMS-LSCE-FFNN sink estimate and the ENSO climate mode (El Niño: MEI > 0.5, La Niña: MEI < -0.5, Neutral: otherwise).

Taking into account the total ocean area of $361.9 \times 10^6 \, \mathrm{km}^2$ and the outgassing of river carbon of $0.78 \, \mathrm{PgC \, yr^{-1}}$ (Resplandy et al., 2018) yields an anthropogenic sink estimate of $2.423 \pm 0.125 \, \mathrm{PgC \, yr^{-1}}$ for the years 1985–2019, respectively $3.141 \pm 0.129 \, \mathrm{PgC \, yr^{-1}}$ for the 2010s and $3.732 \pm 0.158 \, \mathrm{PgC \, yr^{-1}}$ for 2019. As shown in Table 3, the CMEMS-LSCE-FFNN estimates of the annual anthropogenic C uptake for different decades (1990s to 2010s) are in line with the data-based estimates but above

**Table 3.** Comparison of the global anthropogenic $CO_2$ uptake (mean $\pm$ uncertainty) between CMEMS-LSCE-FFNN, and data-based and model-based estimates used in the Global Carbon Project (Friedlingstein et al., 2019, 2020; Hauck et al., 2020). The CMEMS-LSCE-FFNN approach provides contemporary flux estimates. Anthropogenic flux estimates are derived from contemporary fluxes adjusted with the global ocean area of $361.9 \times 10^6 \, \text{km}^2$ and the riverine flux of 0.78 PgC yr$^{-1}$. The estimates in parentheses were provided in Hauck et al. (2020) as the ensemble mean and standard deviation ($\mu_{\text{ensemble}} \pm \sigma_{\text{ensemble}}$) of the model- or data-based estimates.

| | | Periods | | | | | |
|---|---|---|---|---|---|---|---|
| | Methods | 1985–1989 | 1990–1999 | 2000–2009 | 2009–2018 | 2010–2019 | 2019 |
| CMEMS | Contemporary | $0.952 \pm 0.162$ | $1.347 \pm 0.124$ | $1.624 \pm 0.103$ | $2.212 \pm 0.120$ | $2.301 \pm 0.126$ | $2.877 \pm 0.154$ |
| | Anthropogenic | $1.757 \pm 0.166$ | $2.162 \pm 0.127$ | $2.446 \pm 0.106$ | $3.049 \pm 0.123$ | $3.141 \pm 0.129$ | $3.732 \pm 0.158$ |
| GCP2019 | Data | | $(2.32 \pm 0.18)$ | $(2.44 \pm 0.14)$ | $(3.09 \pm 0.10)$ | | |
| | Model | | $2 \pm 0.6 \, (1.99 \pm 0.25)$ | $2.2 \pm 0.6 \, (2.17 \pm 0.26)$ | $2.5 \pm 0.6 \, (2.52 \pm 0.29)$ | | |
| GCP2020 | Model | | $2 \pm 0.5$ | $2.1 \pm 0.5$ | | $2.5 \pm 0.6$ | $2.6 \pm 0.6$ |

the model-based estimates in the GCP publications. Hauck et al. (2020) demonstrated that the spatial distribution of $CO_2$ sources and sinks, as well as decadal trends of the annual mean flux estimates derived from the data-based reconstruction methods and the GOBMs are consistent at the global and regional scales. However, the mismatch in magnitude of these estimates, seasonal cycles, and their interannual variability are still large and remain to be resolved. Note that the uncertainties computed in Hauck et al. (2020) (see estimates in parentheses in Table 3) are defined as the ensemble standard deviation of multiple data-based or model-based products and are lower than the uncertainties reported in the GCP (Friedlingstein et al., 2019, 2020). The latter published a total estimate of $\pm 0.6$ PgC yr$^{-1}$ which corresponds to the combination of the interannual variability derived from GOBMs-based estimates ($\pm 0.4$ PgC yr$^{-1}$) and the uncertainty of the ensemble mean ocean sink ($\pm [0.2 - 0.4]$ PgC yr$^{-1}$).

## 5 Summary and Conclusions

In this paper, we proposed an ensemble of 100 feed-forward neural network models for the reconstruction of air–sea fluxes of $CO_2$ ($fg CO_2$) over the global ocean for the period 1985–2019. This *CMEMS-LSCE-FFNN* model was first used to reproduce the $pCO_2$ fields and we have evaluated its skill. The corresponding monthly fields of $fg CO_2$ were then deduced by applying the air–sea $CO_2$ flux formulation (Eq. 1). Mean state estimates and uncertainty (Eq. 2) from the CMEMS-LSCE-FFNN ensemble-based estimates of air–sea $CO_2$ fluxes have been analysed for the global ocean and 11 RECCAP1 sub-basins (Fig. 2) from the open seas to the continental shelves.

Our estimate for the contemporary net global sink over the period 1985–2019 is $1.643 \pm 0.125$ PgC yr$^{-1}$ including $0.150 \pm 0.010$ PgC yr$^{-1}$ for the coastal sink. The model suggested a net flux of $0.784 \pm 0.178$ PgC yr$^{-1}$ in the year 1985 followed by an increase in the global ocean uptake with a growth rate of $+0.062 \pm 0.006$ PgC yr$^{-2}$. $CO_2$ absorption by the ocean showed little fluctuation in the 1990s followed by an anomalous reduction in the years 1999–2001 (Fig. 9). Thereafter, the ocean sink has strengthened leading to a global uptake rate of $2.301 \pm 0.126$ PgC yr$^{-1}$ in the 2010s. The large interannual to multi-year

variations of the global carbon sink with a temporal standard deviation of $0.526 \pm 0.022$ PgC yr$^{-1}$ are associated to the ENSO climate variability.

The global ocean sink and regional sources and sinks of $CO_2$ computed by CMEMS-LSCE-FFNN (Tables 2 and 3) were compared to the estimates by RECCAP1 (Canadell et al., 2011; Wanninkhof et al., 2013; Schuster et al., 2013; Ishii et al.,
2014; Sarma et al., 2013; Lenton et al., 2013) and GCP (Friedlingstein et al., 2019; Hauck et al., 2020; Friedlingstein et al., 2020). We showed that the magnitude, spatial distribution, and seasonal variations of CMEMS-LSCE-FFNN $CO_2$ fluxes are generally consistent with those suggested in the preceding studies (Feely et al., 2001; Takahashi et al., 2009; Laruelle et al., 2014, 2017) for both the open and coastal seas. Mechanisms shaping the regional distribution (Figs. 5b and 6) and seasonal variations (Fig. 8) of net sinks and sources of $CO_2$ were briefly discussed in Sect. 3.2. The results in Fig. 6 also suggest a
difference between the rank of 11 RECCAP1 sub-basins with respect to their total net sinks or sources and with respect to their mean flux densities per unit area:

- Ranking regional contributions to the global integration of air–sea fluxes: the EP is confirmed as the predominant ocean source region compensating approximately $25\%$ of the total sinks for both the open and coastal seas. The EA regions and the coastal IO are diagnosed as weak sources. Due to its large area, the open StP contributes with the largest regional
sink of $CO_2$ to the global ocean net flux (the StP sink is equivalent to the EP source), followed by the SO, the IO, and the SP. For the coastal regions, the largest sink is computed for the SpA (one third of the total coastal uptake), followed by the northern Pacific and the SO.

- Ranking mean regional flux densities per unit area: the EP remains the strongest source of $CO_2$ followed by the EA and the coastal IO. The $CO_2$ absorption is higher over the Northern hemisphere than over the Southern one with the
strongest uptake per unit area over the open Arctic and SpA. The coastal Arctic, SpA, and StP are identified as the dominant coastal sinks with similar flux densities.

Though statistics and relevant analyses throughout the paper have confirmed that the CMEMS-LSCE-FFNN estimates of sea surface $pCO_2$ and air–sea $CO_2$ fluxes are reasonably reliable, we believe that the model skill can be further improved. The spatial patterns of model–data misfit (RMSD between SOCAT data and the reconstructed fields, Eq. 4) and model uncertainty
(ensemble standard deviation, Eq. 2) computed by the proposed approach (Fig. 5) agree in pointing out where the model poorly recovers evaluation data and/or results large uncertainty estimates. We showed that the uncertainty fields (e.g., Figs. 5c and 5d) produced by the CMEMS-LSCE-FFNN approach are more informative than the standard error maps (e.g., Figs. 5e and 5f). Thus, the CMEMS-LSCE-FFNN uncertainty fields could be used to identify regions that should be prioritized in future extensions of the observational network and confirmed through dedicated observing system simulation experiments (Denvil-
Sommer et al., 2021).

*Data availability.*  The dataset of sea surface $pCO_2$ and air–sea $CO_2$ fluxes analysed in this study is under quality control of the European Copernicus Marine Environment Monitoring Service (CMEMS) and is available for use since 2019. Our dataset can be downloaded through

the CMEMS portal (https://resources.marine.copernicus.eu/?option=com_csw&view=details&product_id=MULTIOBS_GLO_BIO_CARBON_SURFACE_REP_015_008).

*Author contributions.* TTTC, MG, and FC developed the CMEMS-LSCE-FFNN model. TTTC conducted numerical experiments and analysed results with supports from MG and FC. All the authors contributed in preparing and completing the manuscript.

*Competing interests.* The authors declare that they have no conflict of interest.

*Acknowledgements.* The authors would like to thank the Editor, all the referees, and Nicolas Metzl for their constructive comments, suggestions, and supports to improve our manuscript. Many thanks send to everyone involved in data collection and processing of surface ocean $CO_2$ fugacity. The Surface Ocean $CO_2$ Atlas (SOCAT, www.socat.info) is an international effort, endorsed by the International Ocean Carbon Coordination Project (IOCCP), the Surface Ocean Lower Atmosphere Study (SOLAS) and the Integrated Marine Biogeochemistry and Ecosystem Research program (IMBER), to deliver a uniformly quality-controlled surface ocean $CO_2$ database. We acknowledge funding from the European Copernicus Marine Environment Monitoring Service (CMEMS) for the MOB-TAC project in 2018–2021 (83-CMEMS-TAC-MOB contract, https://marine.copernicus.eu/about/producers/mob-tac).

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
