# Peer review of "A seamless ensemble-based reconstruction of surface ocean $pCO_2$ and air-sea $CO_2$ fluxes over the global coastal and open oceans"

_Biogeosciences, 2021_

## Author Comment (AC1)

**Response to Referee's Comments**

**Referee 1: The authors use a neural network model to generate a $pCO_2$ product for the global ocean using the SOCAT data, and combine these $pCO_2$ estimates with a wind speed product to compute the $CO_2$ flux. The ensemble model results compare well overall to the observations, and the carbon flux estimates are in-line with the literature. My main comments concern how novel these results are compared to the extensive literature on the topic, and the interpretation of some of the model statistics.**

Authors: We would like to thank Referee 1 for constructive comments and suggestions on our study. Each point will be addressed in the following and the manuscript has been revised on this base. Throughout this document, the referee's comments are in bold.

**1  There is a lot of previous literature using spatially and temporally sparse observations of surface $pCO_2$ to generate global data products and provide estimates of ocean carbon uptake, some of which use very similar methods to those in this present manuscript. The authors cite this previous literature, but there's very little discussion of it. Consequently, I found it difficult to interpret how the present authors' methods and results are novel and differed from these previous studies. The motivation appears to be in lines 41-46, however, I don't follow how the previous literature did not incorporate "space-time varying uncertainty estimates"? It would also appear that the incorporation of the coasts is relatively new, though the authors then cite a few recent studies and declare that it's a closed gap? I suggest that the introduction needs to contain a much clearer description of how the methods used here compare to previous studies, and what is new about this analysis.**

This study is made up of our efforts to reproduce and intensively analyse the spatially and temporally varying surface $pCO_2$ fields, the air–sea $CO_2$ fluxes, and their reconstruction uncertainties over the global ocean. We acknowledge previous studies pursuing the same target and are aware that the existing observation-based mapping methods (for instance proposed by Rödenbeck et al., 2013; Landschützer et al., 2016; Denvil-Sommer et al., 2019; Gregor et al., 2019; Watson et al., 2020) succeeded in obtaining a relatively low misfit between the reconstructed and observational data (see in Rödenbeck et al., 2015; Gregor et al., 2019; Friedlingstein et al., 2020). Although similar interpolation and machine learning approaches (e.g., clustering, classical regression, neural networks) and/or similar sets of predictor variables for $pCO_2$ have been considered in the preceding literature, model design and implementation are still different (e.g., proportion of SOCAT data used in model fitting and evaluation). The present manuscript reflects our vision on the following key features.

i. A design of an ensemble of numerous feed forward neural network (FFNN) models:

It is based on a Monte Carlo approach wherein each model is trained and validated on sub-samples randomly drawn from the monthly gridded SOCATv2020 data and available data of predictors. The ensemble size of 100 is considered in this study. Our proposed ensemble approach was developed at the Laboratoire des Sciences du Climat et de l'Environnement (LSCE) as both an extension and an improvement of the first version (LSCE-FFNN-v1, Denvil-Sommer et al., 2019). Quality assessments comparing these two model versions are documented in Chau et al. (2020). Besides, the proposed approach inherits strengths of the existing statistical models and further aims at reducing mapping uncertainties induced by, for instance, discrete boundaries in the two-step clustering-regression by Landschützer et al. (2016); Gregor et al. (2019) or the two-step FFNN-based reconstruction of $pCO_2$ climatologies and anomalies by Denvil-Sommer et al. (2019). As described in the Method section (2.2) in the manuscript, each FFNN model follows a leave-$p$-out cross-validation approach, i.e., the exclusion of $p$ gridded SOCAT data of the reconstructed month itself in model training and validation. This allows to reduce model over-fitting. In addition, it leaves more independent data for evaluation than previous approaches, results obtained by the proposed reconstruction are in line with the others though (see e.g., Friedlingstein et al., 2020, and references therein).

ii. Quantification and evaluation of model best estimates (ensemble means) and uncertainties (ensemble spreads):

There exists other ensemble-based methods, their concepts and principle objectives are nevertheless different. For example, Gregor et al. (2019) and Gregor and Gruber (2021) introduce machine-learning ensembles with a small ensemble size of different two-step clustering-regression models mapping surface $pCO_2$ and propose the ensemble mean as their model best estimate. In a broader context, Rödenbeck et al. (2015) suggest an intercomparison of multiple mapping methods targeting the identification of common or distinguishable features of different mapping results. Hauck et al. (2020) and Friedlingstein et al. (2020) synthesize $pCO_2$ mapping products and refer to an ensemble of their observation-based estimates of air-sea $CO_2$ fluxes as a benchmark to compare with the one derived from ocean biogeochemical models.

Up to recently, most of these studies have used misfits between the reconstructed and observational data (e.g., the root-mean-square deviation, RMSD) to evaluate product quality and infer uncertainty estimates of the reconstructed $pCO_2$. Reconstruction errors of $pCO_2$ are then propagated to get uncertainty estimates of the reconstructed $CO_2$ fluxes (see in Landschützer et al., 2014, for instance). By construction, such uncertainty estimates are restricted to oceanic regions and periods when observations are available (Lebehot et al., 2019; Hauck et al., 2020), and the uncertainty quantification of an averaged $pCO_2$ or an integrated flux is under low confidence due to sparse data density. An advantage of our approach is that an ensemble of 100 model outputs of $pCO_2$ and $CO_2$ fluxes is available at each $1° \times 1°$ ocean grid cell of the globe for each month in the period 1985–2019. The ensemble asset facilitates the quantification of model uncertainty of $pCO_2$ and $CO_2$ fluxes averaged or integrated over space and time of interest (see for instance Figures 5 and 9 in the manuscript). This is expected to provide more robust estimates than the ones based on reconstruction errors.

iii. Seamless analysis of the reconstructed data and uncertainty estimates over the open ocean and coastal zones:
An in-depth analysis has been made and presented for both the open and coastal regions divided by latitude bands. Interpretations of good or poor reconstructions of surface $pCO_2$ and air-sea $CO_2$ fluxes (e.g., data density and distribution, regional to local characteristics of $pCO_2$ and its potential drivers, model design and resolution) and changes in spatial and seasonal variations of $CO_2$ fluxes are given. To strengthen our interpretation, we have shown both the temporal and spatial distribution of the reconstructed $pCO_2$ and $CO_2$ fluxes fields, model-data misfits, model uncertainty, and linked these materials with their driving mechanisms suggested in previous literature. More importantly, we have made an intercomparison of model reconstruction ability between regions, identified oceanic sectors where the model does not fit the data well, and suggested further improvements on the data reconstruction based on the proposed space-time varying uncertainty fields.

With these points involved in the manuscript, we believe that our study is novel and statistics and keys findings therein would be useful contributions for the marine science community. However, we agree with the referee that the first version of the manuscript missed part of discussions on the comparison among the existing methods, and thus the novelty of this study was not bold. We consider this referee's feedback important and it has been taken into account in our revision. Precisely, we have reworked on the last two paragraphs in the Introduction section (Lines 37-55 of the manuscript). The two new paragraphs are reproduced below.

[revised manuscript text omitted]

**2  I think the methods section is missing a few key details that will help support this manuscript.**

- **First, it would be helpful for the authors to explain how to interpret and compare the RMSD and $r^2$ values for each region. The reason being, that these values are listed for each ocean region, but it's a little unclear what differences in these values between regions is saying about the model estimate. For example, I was surprised by how low the RMSD value for the Southern Ocean is (slightly lower than the global mean), despite the somewhat limited observational data and well documented, substantial inter-annual variability. However, the Southern Ocean does have a lower $r^2$ value, which the authors seem to attach a greater weight to in their interpretation.**

We have revised the manuscript and will add in Section Methods details of the statistics used in this study to facilitate the readers' interpretation of our results. In general, RMSD measures the model skill in terms of mean distance between model estimates and evaluation data while $r^2$ measures the proportion of data variation predicted by the model. RMSDs between the model and SOCAT gridded data over the Southern Ocean (open: 19.18 $\mu$atm, coastal: 35.73 $\mu$atm) are slightly higher [lower] than the global errors (open: 17.87 $\mu$atm, coastal: 35.86 $\mu$atm) for the open ocean [coastal zone], but the regional $r^2$ values (open: 0.62, coastal: 0.65) are lower than the global ones (open: 0.78, coastal: 0.70). The global scores involve the ones of all the regions, where the poorest reconstruction were found over the Arctic, subpolar, and coastal regions (Figures 3c, 5e, S2, and Table S2). Compared to other metrics such as model bias and $r^2$, RMSD takes another role as an outlier detector of model-data misfits which gives larger weights to such high errors over these regions. Yet, data sampling is limited over the Southern Ocean but similar circumstances appear over polar/subpolar and coastal regions. We have also learned that the interannual variability of $pCO_2$ over the Southern Ocean is moderate compared to that over the Equatorial Pacific and polar/subpolar regions (see also in Rödenbeck et al., 2015; Denvil-Sommer et al., 2019). However, air-sea fluxes vary greatly over the Southern Ocean (SO), we also show that the SO RMSD between our fluxes and SOCAT-based estimates are larger than those of certain regions (Table S2).

The statistics (e.g., Bias, RMSD, $r^2$, and number of data grided from SOCAT observations) listed in Table S2 and scattered in Figure 3c for different open and coastal regions provide a general comparison of the reconstruction skill of the CMEMS-LSCE-FFNN model among the oceanic basins. Nevertheless, examining merely these numbers would not give us a robust assessment of the full story behind. As one of the contributions of this study compared to the existing publications, an intensive analysis has been made and presented in subsections 3.1.2-3.1.6 for both the open ocean and the coastal zones. Interpretations of key factors driving a good or poor reconstruction of surface $pCO_2$ (e.g., data density and distribution, regional to local characteristics of $pCO_2$ and its potential drivers, model design and resolution) are given. To strengthen our interpretations, we have shown both the temporal and spatial distribution of SOCAT data, model-data errors, model uncertainty (e.g., in Figures 3ab, 5e, S1-S3) and scattered them with their driving mechanisms suggested in the literature. Based on these materials, we have made an intercomparison of model reconstruction ability between regions, identified oceanic sectors where the model does not fit the data, and importantly we have suggested improvements on the data reconstruction.

- **Second, I'm a little confused by equation (2). Why is the equation for the mean squared deviation (MSD) shown when it's the root mean squared deviation (RMSD) which is calculated throughout the manuscript? Also, the text refers back to this equation for the definition of the $\sigma$ misfit, but this definition is itself within the MSD equation and is not clearly labeled on its own.**

  For Equation 2, we will put the square root over the formula of MSD. The precise definition of $\sigma_{\mathrm{misfit}}$ will be given in the revised manuscript.

- **Lastly, I think the description of the wind speed product used should be included in the main text rather than the supplementary, considering that this will have a large impact on the overall flux numbers (which the authors do highlight in the results).**

  The wind speed product will be included in the main text.

**3  I suggest re-working the 2nd paragraph of the abstract. This paragraph currently reads like a laundry list of different regions and where they fall in terms of largest total source/sink, largest flux density source/sink, along with coastal and open ocean qualifiers. This many iterations of "X is the greatest …" makes it difficult to follow-along and is not particularly interesting (e.g. the equatorial Pacific as the strongest source of carbon to the atmosphere is not a surprising result). Instead, highlight some of the other key findings, like the increase in ocean carbon uptake over the 1985-2019 timeframe (right now the mean is just listed, but the change is highlighted in the conclusion).**

The second paragraph of the abstract has been modified leading to changes in the full abstract as follows.

- Previous text: *We have estimated the air–sea $CO_2$ fluxes ($fgCO_2$) over the global ocean from the open sea to the continental shelves. Fluxes and associated uncertainty were computed from an ensemble-based reconstruction of $CO_2$ sea surface partial pressure ($pCO_2$) maps trained with observations from the Surface Ocean $CO_2$ Atlas v2020 database. The ensemble mean (which is the best estimate provided by the approach) fits independent data well and a broad agreement between the spatial distribution of model-data differences and the ensemble standard deviations (which are our model uncertainty estimate) is seen. The space-time varying uncertainty fields identify oceanic regions where improvements in data reconstruction and extensions of the observational network are needed. Poor reconstructions of $pCO_2$ are primarily found over the coasts and/or in regions with sparse observations, while $fgCO_2$ estimates with largest uncertainty are observed over the open Southern Ocean ($44°S$ southward), the subpolar regions, the Indian gyre, and upwelling systems.*

  *Our estimate of the global net sink for the period 1985–2019 is $1.643\pm0.125\,\mathrm{PgC\,yr^{-1}}$ including $0.150\pm0.010\,\mathrm{PgC\,yr^{-1}}$ for the coastal net sink. Results suggest that the open ocean Subtropical Pacific (between $18°N$–$49°N$) has the strongest $CO_2$ sink ($0.485\pm0.014\,\mathrm{PgC\,yr^{-1}}$) among the basins of the world, followed by the open ocean sub-basins in the Southern hemisphere. The coastal Subpolar Atlantic (between $49°N$–$76°N$) is the most significant coastal net sink, amounting to one third of the total coastal uptake; the northern Pacific continental shelves (north of $18°N$) are the next contributors. The Equatorial Pacific (between $18°S$–$18°N$) is the predominant source emitting $0.523\pm0.016\,\mathrm{PgC\,yr^{-1}}$ of $CO_2$ back to the atmosphere. Based on the mean flux density per unit area, the most intense $CO_2$ drawdown is, however, observed over the Arctic ($76°N$ poleward) followed by the Subpolar Atlantic and Subtropical Pacific for both open ocean and coastal sectors. The mean efflux density over the Equatorial Pacific remains the highest, but similar densities can also be found along other strong upwelling systems in the equatorial band.*

- Revised text: *We have estimated the air–sea $CO_2$ fluxes ($fgCO_2$) over the global ocean from the open sea to the coastal ocean. Fluxes and associated uncertainty are computed from an ensemble-based reconstruction of $CO_2$ sea surface partial pressure ($pCO_2$) maps trained with observations from the Surface Ocean $CO_2$ Atlas v2020 database. The ensemble mean (which is the best estimate provided by the approach) fits independent data well and a broad agreement between the spatial distribution of model-data differences and the ensemble standard deviations (which is our model*

*uncertainty estimate) is seen. The space-time varying uncertainty fields identify oceanic regions where improvements in data reconstruction and extensions of the observational network are needed. Poor reconstructions of $p$CO$_2$ are primarily found over the coasts and/or in regions with sparse observations, while $fg$CO$_2$ estimates with largest uncertainty are observed over the open Southern Ocean ($44°S$ southward), the subpolar regions, the Indian gyre, and upwelling systems.*

*Our estimate of the global net sink for the period 1985–2019 is $1.643\pm0.125\,\mathrm{PgC\,yr^{-1}}$ including $0.150\pm0.010\,\mathrm{PgC\,yr^{-1}}$ for the coastal net sink. Among oceanic basins, the open ocean Subtropical Pacific ($18°N$–$49°N$) and the coastal Subpolar Atlantic ($49°N$–$76°N$) are recognized as the strongest CO$_2$ sinks contributing respectively $0.485\pm0.014\,\mathrm{PgC\,yr^{-1}}$ to the global ocean sink and one third to the total coastal uptake. Reconstruction results show significant changes in the global integration of CO$_2$ fluxes exchanging through the air-sea interface. We compute a net flux of $0.784\pm0.178\,\mathrm{PgC\,yr^{-1}}$ in the year 1985 and an increase of the global ocean sink with a growth rate of $+0.062\pm0.006\,\mathrm{PgC\,yr^{-2}}$. CO$_2$ absorption by the ocean was rather stable in the 1990s followed by an anomalous reduction in the years 1999–2001 and a strengthening uptake of $2.301\pm0.126\,\mathrm{PgC\,yr^{-1}}$ in the 2010s. The temporal standard deviation of the annual ocean uptake is $0.526\pm0.022\,\mathrm{PgC\,yr^{-1}}$ for the full period. The link between its large interannual to multi-year variations and the El Niño-Southern Oscillation climate variability is also reconfirmed.*

**OTHER COMMENTS**

**4   Lines 37-40: Should all the manuscripts be separated with a comma rather than a semicolon? And why is the Rödenbeck et al. (2015) manuscript specifically highlighted as "other mapping methods"?**

The comma is now used to separate the references. Rödenbeck et al. (2015) is cited in the manuscript as one of the studies which made an intercomparison between different observation-based mapping methods reconstructing ocean surface $p$CO$_2$ and quantifying CO$_2$ fluxes. However, we have changed the text in the Introduction (see our reply to Referee's comment **1**).

**5   Line 139: How is the variability in "analytical equipment" accounted for here?**

Thank you. The word "analytical equipment" was not appropriate in the context of the sentence in lines 139-141. We have rewritten it as follows:

- Previous text: *Variability in the number of cruises and analytical equipment induces measurement latitude and longitude offsets from the cell center, e.g., with an average of $0.34°\pm0.14°$ as reported in Sabine et al. (2013) which are not taken into account.*

- Revised text: *Variability in the location of cruises and instruments induces measurement latitude and longitude offsets from the cell center, e.g., with an average of $0.34°\pm0.14°$ as reported in Sabine et al. (2013) which are not taken into account.*

**6   Figure 2: I suggest directly labeling each region in the figure with the abbreviated label (i.e. SpA for subpolar Atlantic) for clarity. Figure 5 and 8: The tick marks in the colorbar for these figures are relatively large and look like a negative sign, I'd suggest making them much smaller.**

Figures 2, 5, and 8 have been modified following the Referee's suggestion. The label of Figure 2 is now changed from the numbers to the abbreviated names of 11 regions. The size of tick marks in the colorbar of Figures 5 and 8 is also reduced.

[revised manuscript text omitted]

---

## Author Comment (AC2)

**Response to Referee's Comments**

**Referee 2: At present, there are many data products in marine physics, such as temperature and salinity products, but there are few data products in marine chemistry. I support the publication of more marine chemistry data products.**

Authors: We thank Referee 2 for his/her interest in marine chemistry data products and comments/suggestions on our study. We will reply to each comment in the following. Throughout this document, the referee's comments are in bold.

**1 The author reconstructed surface ocean $p$CO$_2$ based on FFNN with region divided by latitudes and similar predictors with previous researches was used, which is not novel.**

With the proposed ensemble-based mapping method, statistics, and keys findings presented in the manuscript, we believe that our study is novel and would be a valuable contribution for the marine science community. However, we admit that the first version of the manuscript missed part of discussions on the comparison among the existing methods, and thus the novelty of this study was not easy to interpret. We consider this Referee's feedback important and have revised the manuscript in such a way that our three main contributions are elaborated and highlighted.

First, we propose an ensemble of 100 feed forward neural network (FFNN) models to reproduce and intensively analyse spatially and temporally varying surface $p$CO$_2$ fields, air–sea CO$_2$ fluxes, and their associated uncertainties over the global ocean. At the first glance, this FFNN-based approach looks similar to the other mapping methods proposed in previous studies (e.g., Rödenbeck et al., 2013; Landschützer et al., 2016; Denvil-Sommer et al., 2019; Gregor et al., 2019) since all have used statistical approaches (e.g., interpolation, clustering, classical regression, neural networks) to pursue the same target. Nevertheless, our model design and implementation are different from the others'. The mapping method developed in this study is based on a Monte Carlo approach wherein each model is trained and validated on sub-samples randomly drawn from the monthly gridded SOCATv2020 data and available data of predictors. Besides, the exclusion of $p$ gridded SOCAT data of the reconstructed month itself in model training and validation allows to reduce model over-fitting and to leave much more independent data for model evaluation possibly than the previous studies. Despite leaving a large volume of evaluation data, results obtained are in line with the other reconstructions (see e.g., Friedlingstein et al., 2020, and references therein). The proposed approach is both an extension and an improvement of the first version developed by Denvil-Sommer et al. (2019) at the same laboratory (see quality assessments comparing these two model versions in Chau et al., 2020). Furthermore, the ensemble of FFNN models is framed such that strengths of the existing statistical models have been taken into account and mapping uncertainties can be reduced. That latter are systematically induced by, for instance, discrete boundaries in the two-step clustering-regression by Landschützer et al. (2016); Gregor et al. (2019) or the two-step FFNN-based reconstruction of $p$CO$_2$ climatologies and anomalies by Denvil-Sommer et al. (2019) .

Second, we quantify model best estimates (ensemble mean) and uncertainties (ensemble standard deviation) based on the ensembles of 100 model outputs of surface $p$CO$_2$ and air–sea CO$_2$ fluxes, and evaluate these estimates with independent SOCAT data and in situ observations. It is noteworthy that there exist other ensemble-based approaches (Gregor et al., 2019; Rödenbeck et al., 2015; Friedlingstein et al., 2020) but with a small ensemble size (less than 20) and with different concepts and objectives (see in the revised text of the Introduction below for further details). Additionally, in the preceding studies, misfits between the reconstructed and observational data (e.g., the root-mean-square deviation, RMSD) are used to evaluate product quality and infer uncertainty estimates of the reconstructed $p$CO$_2$. Reconstruction errors of $p$CO$_2$ are then propagated to get uncertainty estimates of the reconstructed CO$_2$ fluxes (see in Landschützer et al., 2014, for instance). By construction, such uncertainty estimates are restricted to oceanic regions and periods when observations are available (Lebehot et al., 2019; Hauck et al., 2020), and the uncertainty quantification of an averaged $p$CO$_2$ or an integrated flux is under low confidence due to sparse data density. As an advantage of our approach, an ensemble of 100 model outputs of $p$CO$_2$ and CO$_2$ fluxes is available at each $1° × 1°$ ocean grid cell of the globe for each month in the period 1985–2019. The ensemble asset facilitates the quantification of model uncertainty of $p$CO$_2$ and CO$_2$ fluxes averaged or integrated over space and time of interest (see for instance Figures 5 and 9 in the manuscript and Figures RP2.0, RP2.1, RP2.2, and RP2.3 in this document). This is expected to provide more robust estimates than the ones based on reconstruction errors.

Another key contribution of this study is a seamless analysis of the reconstructed data and uncertainty estimates over both the open ocean and coastal zones. Interpretations of good or poor reconstructions of surface $pCO_2$ and air-sea $CO_2$ fluxes (e.g., data density and distribution, regional to local characteristics of $pCO_2$ and its potential drivers, model design and resolution) and changes in spatial and seasonal variations of $CO_2$ fluxes are given. To strengthen our interpretation, we have shown both the temporal and spatial distribution of the reconstructed $pCO_2$ and $CO_2$ fluxes fields, model-data misfits, model uncertainty, and linked these materials with their driving mechanisms suggested in previous literature. More importantly, we have made an intercomparison of model reconstruction ability between regions, identified oceanic sectors where the model does not fit the data well, and suggested further improvements on the data reconstruction based on the proposed space-time varying uncertainty fields. Note that oceanic regions divided by latitude bands are only used for this analysis of our results, FFNN models themselves do not follow oceanic regions or biomes in clustering before training as proposed in Landschützer et al. (2016); Gregor et al. (2019). The Referee's comment **"The author reconstructed surface ocean $pCO_2$ based on FFNN with region divided by latitudes"** seems to be misleading.

These three main points are now elaborate in Section Introduction. Precisely, the two new paragraphs replacing the ones in Lines 37-55 of the manuscript are as follows.

Revised text:

[revised manuscript text omitted]

**2  The reconstruction of $p\mathrm{CO_2}$ and sea-air $\mathrm{CO_2}$ flux over global coastal oceans are interesting works but the author needs to do much more works on the validation of coastal results. Because a standard deviation of 41.79 $\mu$atm between $p\mathrm{CO_2}$ results and SOCAT observations possibly leads to opposite results in the estimate of coastal $\mathrm{CO_2}$ flux.**

We will add in the revised manuscript results comparing the reconstructed data and independent in-situ data of $p\mathrm{CO_2}$ (part in the coastal regions) proposed by Sutton et al. (2019) (see Figures RP2.0, RP2.1, RP2.2, and RP2.3 attached below). The results confirm a reasonably good reconstruction of the proposed approach that fits observational data well at these locations. Also shown in these figures, the reconstructed timeseries cover the full period 1985-2019 while observations are still sparse

and almost distributed in the last two decades. Thus, the reconstructed timeseries would be favored to provide robust estimates of long-term trends and variations of surface pressure of $CO_2$ and ultimately the corresponding air-sea fluxes.

Referee 2 explains that **a standard deviation of 41.79 $\mu$atm between $pCO_2$ results and SOCAT observations possibly leads to opposite results in the estimate of coastal $CO_2$ flux**. This seems to be misleading. Indeed, we have written in the manuscript (Lines 126-130):

*The reconstructed $pCO_2$ field matches SOCAT data well: both are normally distributed with the same mean of 361.3 $\mu$atm (Fig. 3a) and a high agreement for all percentiles (Fig. 3b) is seen. The slight under- or overestimation at high and low percentiles implies that the model is slightly biased towards the mean value, as is expected when predictor variables do not fully explain predictand variables in the training dataset. This reduced variability is also reflected in the difference between the data standard deviation based on SOCAT $pCO_2$ (41.79 $\mu$atm) and the one based on CMEMS-LSCE-FFNN (36.30 $\mu$atm).*

In our context, 41.79 $\mu$atm is the standard deviation of SOCAT data itself, it is not the standard deviation of differences between the reconstructed and SOCAT observational data.

[Figure]

**Figure RP2.0.** Location map of in situ measurements of ocean surface $pCO_2$ (Sutton et al., 2019).

**3   The CHL data used was only from 1992 to 2019, and was not available in the Arctic and the Southern Ocean in winter, the details about how the reconstruction was carried out when CHL was not available should be declared clearly in the method section.**

We will provide this information in the method section. To be precise, climatologies based on all available CHL data (1998-2019) were used as predictors for data unavailable before 1998. We set CHL approximately to 0 mg m$^{-3}$ as its data is missing in the Arctic and the Southern Ocean in winter. These suggestions can also be found in previous studies (e.g., Landschützer et al., 2016; Denvil-Sommer et al., 2019).

**4   The subskin temperature correction (Watson et al., 2020) should be considered in the estimate of sea-air $CO_2$ flux.**

Watson et al. (2020) proposed a double correction to the SOCAT data and to the computation of the $CO_2$ flux in order to remove some aliasing caused by the temperature vertical gradient within the marine boundary layer. However, the Watson et al. (2020) adjustment, if applied here, would add roughly 0.9 $PgCyr^{-1}$ to the global ocean sink estimate based on observations. The adjusted ocean sink estimate would thus surpass the land sink and result in a large carbon budget imbalance. More evidence of their genericity are needed to apply skin and subskin corrections (Friedlingstein et al., 2020).

[Figure]

**Figure RP2.1.** Time series of ocean surface $pCO_2$ at different stations- part 1 (see in Location map RP2.0). Measurements at each station (Sutton et al., 2019) are monthly averaged, the ensemble mean and ensemble spread are computed from reconstructed data at the four nearest neighbors of that location. Number of observations (N), Bias, RMSD, and model-data correlation (r2) have been computed on these monthly interpolated data. In each subplot, dots stand for observational data and the coloured line with shaded areas stand for the mean and uncertainty envelops computed from the CMEMS-LSCE- FFNN 100-memble ensemble (dark: 68% confidence interval, i.e. mean ± 1std; light: 99% confidence interval, t.e. mean ± 3std).

[Figure]

**Figure RP2.2.** Time series of ocean surface $pCO_2$ at different stations- part 2 (see in Location map RP2.0). Measurements at each station (Sutton et al., 2019) are monthly averaged, the ensemble mean and ensemble spread are computed from reconstructed data at the four nearest neighbors of that location. Number of observations (N), Bias, RMSD, and model-data correlation (r2) have been computed on these monthly interpolated data. In each subplot, dots stand for observational data and the coloured line with shaded areas stand for the mean and uncertainty envelops computed from the CMEMS-LSCE- FFNN 100-memble ensemble (dark: 68% confidence interval, i.e. mean $\pm$ 1std; light: 99% confidence interval, t.e. mean $\pm$ 3std).

[Figure]

**Figure RP2.3.** Time series of ocean surface $pCO_2$ at different stations- part 3 (see in Location map RP2.0). Measurements at each station (Sutton et al., 2019) are monthly averaged, the ensemble mean and ensemble spread are computed from reconstructed data at the four nearest neighbors of that location. Number of observations (N), Bias, RMSD, and model-data correlation (r2) have been computed on these monthly interpolated data. In each subplot, dots stand for observational data and the coloured line with shaded areas stand for the mean and uncertainty envelops computed from the CMEMS-LSCE- FFNN 100-memble ensemble (dark: 68% confidence interval, i.e. mean $\pm$ 1std; light: 99% confidence interval, t.e. mean $\pm$ 3std).

**5  The author should reconsider the topic of this manuscript. If the author want focus on the $CO_2$ flux of global open oceans, additional work was necessary rather than only discussing spatial distribution or interannual variability, because the reconstruction method in this manuscript and the results was not novel. If the author want focus on the $CO_2$ flux of global coastal oceans, which was still a research gap, much more works are needed to make the result convince.**

As mentioned in our response to Referee's comment **1**, the manuscript presents three main contributions:

i. Our data reconstruction is based on a new model design - an ensemble of 100 neural network models.

ii. We quantify and evaluate model best estimates and uncertainties based on the ensemble asset. For the first time, the space-time varying uncertainty estimates (see for instance Figures 5 and 9 in the manuscript and Figures RP2.0, RP2.1, RP2.2, and RP2.3 in this document) derived from the ensemble of model outputs are presented and analysed. We promote the use of the proposed uncertainty fields which would be more informative than the RMSD-based uncertainty fields (e.g., proposed in Landschützer et al., 2014) in identifying oceanic sectors where further improvements on the data reconstruction will be needed.

iii. More importantly, the manuscript presents seamless analysis of the reconstructed data and uncertainty estimates over the open ocean and coastal zones. Furthermore, the open ocean estimates are considered as references for the coastal data assessment.

We believe that the ensemble-based approach and analysis therein are novel and the Introduction section has been changed to better highlight these contributions. However, we agree with the referee that the reconstructed data need to be evaluated with independent in situ observations. In the revised version of the manuscript we propose to add Figures RP2.0, RP2.1, RP2.2, and RP2.3 and an interpretation of these results (see also in our reply to Referee's comment **2**).

**6  Line 76: "An ensemble of 100 FFNNs was used to reconstruct monthly $pCO_2$ fields......", How are these 100 models built? Why did you do that? How are the results of 100 models selected? Line 84-85: "The random extraction and the FFNN training were repeated 100 times so that 100 versions of the monthly FFNNs have been obtained", Why is it the "100 times"? How is the "100 times" determined? Does it converge after 100 iterations?**

The description of the construction of the ensemble approach is given in the manuscript (Lines 81-89) as follows (Figure 1 is used to illustrate a neural network model mapping the target $pCO_2$ and predictor variables).

*To reconstruct the $pCO_2$ fields over the global ocean for each target month over the 1985–2019 period, all the available SOCAT data and the co-located predictors have been collected for the month before and the month after the target month. We randomly extracted two thirds of each one of these datasets to make training datasets for the FFNNs, leaving the remaining third to be corresponding test datasets. The FFNNs were then trained for each target month.*

Our ensemble approach comprises multiple network models, each trained and validated on resampled data of SOCAT $pCO_2$ and predictors. In statistics, it belongs to the classes of bootstraping and Monte Carlo methods. Theoretically, the number of samples or the ensemble size must be substantially large to get a convergence. However, it was demonstrated in the literature (e.g., Goodhue et al., 2012; Efron et al., 2015) that with the ensemble size of 50 the model estimation is likely stable and with the ensemble size over 100 the improvement in standard errors between model outputs and evaluation data is negligible. It was also tested in the first phase of our model development. Figure RP2.4 shows an illustration of the reconstruction skill with respect to the ensemble size. For each ensemble of $N$ model outputs of $pCO_2$ ($N \in \{5, 10, 20, 50, 75, 100\}$), RMSD is computed between the ensemble mean (our best model estimate) and SOCAT data over the period 1985-2019. As seen in this figure, the reconstruction starts to stabilize with $N = 50$. In this study, we have exploited a large but realistic amount of computing resources to run an ensemble of 100 neural network models.

We will make this information appear in the Method section.

[Figure]

**Figure RP2.4.** RMSD between a best estimate (ensemble mean) and SOCAT data of ocean surface $pCO_2$ with respect to the ensemble size in $\{5,\ 10,\ 20,\ 50,\ 75,\ 100\}$.

**7    Table 2, What is the meaning of two numbers in the rightmost column of Table 2, for example: $0.07 \pm 0.04$, $0.30 \pm 0.13$**

The numbers without brackets (e.g., $0.07 \pm 0.04$, $0.30 \pm 0.13$) in the rightmost column of Table 2 in the manuscript refer to the estimates derived from observation-based methods. In the caption of Table 2, we wrote: *In column 'RECCAP1', values in parentheses are the 'best' estimates proposed by RECCAP1 studies, the others are the estimates computed with different methods using $pCO_2$ observations*. Further information can be found in Lines 438-443:

*RECCAP1 best estimates were derived from averages or medians of estimates based on the $pCO_2$ climatology or $pCO_2$ diagnostic model, and/or the atmospheric and ocean inversions and GOBM models (see Schuster et al., 2013; Ishii et al., 2014; Sarma et al., 2013; Lenton et al., 2013, and references therein). The observation-based estimates of regional net fluxes reported in these studies were computed from the reconstruction of SOCAT $pCO_2$ data (only used in Schuster et al., 2013), LDEO data (https://www.ldeo.columbia.edu/res/pi/CO2/), and its climatology (Takahashi et al., 2009).*

This information will be added to the caption of Table 2 to make it more visible to the readers.

**8    Line 444-446: "The global open ocean uptake obtained in this study of $1.344 \pm 0.111$ PgC yr$^{-1}$ lies between the observation based net sink estimate by Wanninkhof et al. (2013) ($1.18$ PgC yr$^{-1}$) and the global sum of regional best estimates given in Table 2 ($1.8$ PgC yr$^{-1}$)". In table 2, I can't find the value of $1.8$ PgC yr$^{-1}$**

It means that $1.8$ PgC yr$^{-1}$ is the sum of all the 'best' estimates (between brackets) given in Table 2.

**9    Line 462-463: The discrepancy is possibly due to an overestimation of Arctic $pCO_2$ by the CMEMS-LSCE-FFNN (see in Sect. 3.1.2) and to the lack of estimates over a large portion of the seasonally sea–ice covered regions. This sentence means that the data in the Arctic are not accurate at present. So the data in the Arctic is not suitable for use at present.**

Results shown in Sect. 3.1 in the manuscript confirm that the model reconstruction of $pCO_2$ over the Arctic does not fit SOCAT data well and is much more uncertain than for other oceanic regions. The factors behind the poor estimates of Arctic $pCO_2$

have been further discussed in the Discussion section (Lines 460-469). Despite the need for further improvements, our analysis fairly documents the current status and discusses the way forward.

**References**

Bushinsky, S. M., Landschützer, P., Rödenbeck, C., Gray, A. R., Baker, D., Mazloff, M. R., Resplandy, L., Johnson, K. S., and Sarmiento, J. L.: Reassessing Southern Ocean air-sea $CO_2$ flux estimates with the addition of biogeochemical float observations, Global biogeochemical cycles, 33, 1370–1388, 2019.

Chau, T. T. T., Gehlen, M., and Chevallier, F.: QUALITY INFORMATION DOCUMENT for Global Ocean Surface Carbon Product MULTIOBS_GLO_BIO_CARBON_SURFACE_REP_015_008, Research report, Le 
[revised manuscript text omitted]

---

## Author Comment (AC3)

**Response to Referee's Comments**

**Meike Becker (Referee 3): The authors present an estimate of global air-sea $CO_2$ fluxes based on interpolating gridded SOCAT $pCO_2$ data. They use an ensemble of 100 feed-forward neural network models (FFNN) and sea surface height, sea surface temperature, sea surface salinity, mixed layer depth, chlorophyll-a, atmospheric mole fractions, a $pCO_2$ climatology and position data as drivers. They present an uncertainty analysis based on their ensemble spread as a semi-independent parameter, which is better than many available air-sea flux products. However, there are a few things that should be improved.**

Authors: We would like to thank Meike Becker (Referee 3) for her positive feedback and suggestions. We respond to each point as follows. Throughout this document, the referee's comments are in bold.

1  **One point that needs improvement is the description of the driving data, where some important information is missing. The driving data that were used are not available for the full period for which the authors present flux maps. How did you deal with that? Did you use a climatology for CHL, SST, MLD etc. before the early/mid 90s? If so, what was this based on? This information is crucial for interpreting interannual variability prior to the mid-90s.**

The information of all data used in our reconstruction has been presented in Table S1 (Supplementary material). As shown in this table, monthly data of sea surface temperature (SST) and atmospheric mole fractions ($xCO_2$) which are possibly key drivers to trends and interannual variability of the $pCO_2$ field are available over the full period (1985-2019). It is also noted at the end of Table S1: **For some data unavailable before 1998, climatologies based on all available data were used as predictors. Exceptionally, predictors for SSH before 1993 are climatologies plus a linear trend in order to retain the overall response to the global warming. MLD before 1992 was taken as the average MLD between 1992 and 1997.** We, however, agree with the referee to make this information clearer to the readers. This information will be added in the main text.

2  **Another thing that I want to point out, is the inconsistent and partly misleading use of the terms 'observations', 'sample' and 'data'. The authors base their product on a gridded version of the SOCAT data set (monthly, 1x1). In order to avoid confusion, the term 'observations' should be reserved for data that has been retrieved from field work, in the case the original $pCO_2$ measurements in the SOCAT database. The gridded version contains monthly, 1x1 averages of these $pCO_2$ measurements. When the authors write about 'X observations' in a certain region, they actually mean 'grid boxes with observations'. Please make sure that this becomes clearer. In line 195 for example, the authors write 50 to 220 samples per year'. Here the authors should specify that they mean 'grid boxes with data' as the reader easily can assume that there were only 50-220 $pCO_2$ observations every year.**

Thank you for this suggestion. We will use precisely the terms 'observations', 'sample' and 'data' in the revised version of the manuscript.

3  **I also want to comment on Figure S1. Here the authors show the coverage of the gridded SOCAT product and its variability where they mention '$pCO_2$ individuals'. I don't understand if this means the original SOCAT $pCO_2$ observations (i.e. a measure of how well the gid box mean represents the actual conditions), or the $pCO_2$ of the gridded version (showing the variability within the gridded product).**

In the (b) and (c) subplots of Figure S1 we show maximal variability of $pCO_2$ individuals within a grid cell, i.e.,

$$\max_{t}\{pCO_{2,tij}^{\max} - pCO_{2,tij}^{\min}\}$$

where $t$, $ij$ indicates time and space indices. $pCO_{2,tij}^{\max}$ and $pCO_{2,tij}^{\min}$ were converted from the corresponding values of $CO_2$ fugacity observations which are available in the monthly gridded SOCATv2020 database. We think that the term '$pCO_2$

individuals' is correct in this sense. We will make consistent use of the terms of 'observations' and 'gridded data' throughout the main text, and reword the legend of Figure S1 to avoid any ambiguity.

**4   I understand that the authors used the subocean divisions from RECCAP 1. This of course increases the comparability to the results of RECCAP 1, but also this makes the results difficult to interpret. Using a biome scheme such as used in RECCAP 2 (e.g. after Fay and McKinley (2014)) would have led to a clearer separation of regions with similar characteristics, and thus increased the interpretability. I also miss a discussion of how this product performs in comparison to other global air-sea $CO_2$ flux products.**

We are on the same page with Referee 3 that using biomes proposed by Fay and McKinley (2014) would provide a better interpretation of small-scale characteristics of $pCO_2$ and air-sea $CO_2$ fluxes. We have also used the biome mask for further studies on their trends, seasonal cycles, and spatial and interannual variability. However, geometries of the biomes (e.g., their boundaries) would complicate the evaluation of the CMEMS-LSCE-FFNN model estimates and uncertainty, e.g., the comparison between the model outputs and sparse SOCAT data, and the analysis of results obtained for the open ocean and the coastal regions. Regarding the scope of this study, we have chosen to use the subocean divisions with latitude bands. Consequently, the CMEMS-LSCE-FFNN estimates of regional air-sea $CO_2$ fluxes have been compared to the ones presented in RECCAP1.

Each of the observation-based reconstruction methods for $pCO_2$ and $CO_2$ fluxes has both strengths and weaknesses, we have revised the Introduction (Lines 37-55), the new paragraphs (see italic text below) better interpret these terms and discuss the performance of the CMEMS-LSCE-FFNN approach compared to the others. In-depth intercomparisons amongst different model-based and/or observation-based products are presented in previous works including Rödenbeck et al. (2015); Denvil-Sommer et al. (2019); Hauck et al. (2020) are beyond the scope of this study.

Revised text:

*Various data-based approaches have been proposed to infer gridded maps of surface ocean $pCO_2$ from the sparse set of observational data. They have been successful in obtaining similarly low misfits between the reconstructed and evaluation data and reasonable estimates of air-sea $CO_2$ fluxes (see in Rödenbeck et al., 2015; Gregor et al., 2019; Friedlingstein et al., 2020) although model design and implementation are quite different (e.g., proportion of SOCAT data used in model fitting and evaluation). Aside from data reconstruction built on a single model mapping $pCO_2$ data with machine learning, classical regression, or mixed layer schemes (see Rödenbeck et al., 2013; Landschützer et al., 2016, for a few), ensemble-based approaches have recently emerged but with their own concepts and objectives. For example, Denvil-Sommer et al. (2019) designed a two-step reconstruction of $pCO_2$ climatologies and anomalies based on five neural network models and selected the one that reproduced the $pCO_2$ field with the smallest model-data misfit. Gregor et al. (2019) and Gregor and Gruber (2021) introduced machine-learning ensembles with six to sixteen different two-step clustering-regression models mapping surface $pCO_2$ and suggest that the use of their ensemble mean is better than each member estimate. In a broader context, Rödenbeck et al. (2015) presented an intercomparison of fourteen mapping methods targeting the identification of common or distinguishable features of different products in long-term mean, regional and temporal variations. Hauck et al. (2020) and Friedlingstein et al. (2020) also synthesized $pCO_2$ mapping products and take an ensemble of their observation-based estimates of air-sea $CO_2$ fluxes as a benchmark to compare with the one derived from ocean biogeochemical models. Despite positive conclusions overall, statistical data reconstructions are still subject to further improvements. In Rödenbeck et al. (2015), Hauck et al. (2020), Bushinsky et al. (2019), and Denvil-Sommer et al. (2021), the authors explain that substantial extensions of surface ocean observational network systems are essential to better determine $pCO_2$ and fluxes at finer scales and reduce mapping uncertainties. So far mapping uncertainties have been estimated by using misfits between the model outputs and SOCAT data (e.g., the root-mean-square deviation, RMSD). By construction, such uncertainty estimates are restricted to oceanic regions and periods when observations are available (Rödenbeck et al., 2015; Lebehot et al., 2019; Gregor et al., 2019) and the uncertainty quantification of an averaged $pCO_2$ or an integrated flux over space and time of interest is under low confidence due to sparse data density. Furthermore, most of the previous mapping methods target $pCO_2$ data and evaluate their estimates solely over the open ocean, with the coastal data excluded or not fully qualified. In Laruelle et al. (2014, 2017)*

*the authors present spatial distribution of air–sea flux density and estimates of total coastal C sink while a recent study (Landschützer et al., 2020) limits their estimation to monthly climatologies of $p\mathrm{CO}_2$ over the global ocean including the coastal regions.*

*In this work, we propose a new inference strategy for reconstructing the monthly $p\mathrm{CO}_2$ fields and the contemporary air–sea fluxes over the period 1985–2019 with a spatial resolution of $1° \times 1°$. It is based on a Monte Carlo approach, an ensemble of 100 neural network models mapping sub-samples drawn from the monthly gridded SOCATv2020 data and available data of predictors. This ensemble approach was developed at the Laboratoire des Sciences du Climat et de l'Environnement (LSCE) as both an extension and an improvement of the first version (LSCE-FFNN-v1, Denvil-Sommer et al., 2019). In the following sections, we first present the ensemble of neural networks designed with the aim of leaving aside the issue of discrete boundaries in the existing two-step clustering-regressions (see further discussion in Gregor and Gruber, 2021) and reducing the mapping uncertainties induced by the two-step reconstruction of the $p\mathrm{CO}_2$ fields (Denvil-Sommer et al., 2019) or by an ensemble-based reconstruction with a small ensemble size. In addition, each FFNN model follows a leave-$p$-out cross-validation approach, i.e., the exclusion of $p$ gridded SOCAT data of the reconstructed month itself in model training and validation. This allows to reduce model over-fitting and to leave much more independent data for model evaluation than the previous studies. Mean and standard deviation computed from the ensemble of 100 model outputs are defined as estimates of the mean state and uncertainty of the carbon fields. As one of the novel key findings of this study compared to the existing ones, we compute and analyze the estimates of $p\mathrm{CO}_2$ and air–sea fluxes, model errors, and model uncertainties for different time scales (e.g., monthly, yearly, and multi-decadal) and spatial scales (e.g., grid cells, sub-basins, and the global ocean). We then suggest the use of an indicator map built on the space-time varying uncertainty fields instead of model-data misfits for identifying regions that should be prioritized for future observational programs and model development in order to improve the data reconstruction. Last but not least, the model best estimates and uncertainty of $p\mathrm{CO}_2$ and air–sea fluxes are analysed seamlessly over the open ocean to the coastal zone. Potential drivers of the spatio-temporal distribution and the magnitude of open ocean and coastal $\mathrm{CO}_2$ fluxes are discussed with the aim to better identify underlying processes and to detect potential focus regions for further studies on the evolution of oceanic $\mathrm{CO}_2$ sources and sinks.*

**5   Minor suggestions**

- **L 40: $p\mathrm{CO}_2$ was not introduced as an abbreviation.**
- **L 59: Tr is not described.**
- **L 86: change to: p is the number of grid cells with observations.**
- **Figure 3: The yellow bars in panel c) are very difficult to read, especially the first one.**
- **Figure 4a/b: You show the number of observation (or grid cells with observations) per year. Please change that.**

We have taken them into account in this revision.

**6   Additionally change STD to s. Go through the manuscript and make sure, that you use consistent terminology.**

STD will be changed to $\sigma$ for a consistent use of its notation.

**7   L 140, add temporal offsets from the cell center. In many regions this will be the dominant one, especially during the productive season.**

There is no temporal offsets provided in the SOCAT database.

**8 L 182: Be aware that Laruelle et al. (2017) and Landschützer et al. (2020) are climatologies.**

Landschützer et al. (2020) reconstructed monthly open and coastal $pCO_2$ based on two approaches: one proposed in Landschützer et al. (2016) using SOCAT data at $1° \times 1°$ resolution for the open ocean over 1982-2016 and another proposed in Laruelle et al. (2017) using SOCAT coastal data at $0.25° \times 0.25°$ resolution over 1998-2015. The monthly reconstructions were evaluated with SOCAT data over the common period 1998-2015 and the long-term mean and seasonal climatologies of $pCO_2$ shown in Landschützer et al. (2020) were created from those fields.

In lines 179-182 in the manuscript, we have written: *For the 1998–2015 period, the CMEMS-LSCE-FFNN model scored an RMSD of 35.84 μatm, larger than the coastal reconstruction error of 26.8 μatm by Landschützer et al. (2020). The latter unified data for the same period from two conceptually equivalent reconstruction models, one covering the open ocean (Landschützer et al., 2016) and one targeting the coastal ocean (Laruelle et al., 2017).*

**9 L 307: Please round the uncertainties to 2 significant digits (or less if it seems unrealistically low) and the measured value to the same number of digits, for example $2.336 \pm 0.104$ to $2.34 \pm 0.10$. Please do so for all uncertainties in the manuscript.**

We have rounded estimates of air-sea fluxes and uncertainties to 3 digits since some of them become 0 with less than 3 digits; for instance, fluxes and uncertainty estimates over coastal regions (see in Table 2).

**10 L 328-330: Please correct this. Primary production and respiration have usually only a very small influence on alkalinity (if we neglect anerobic remineralization processes for the moment): primary production increases alkalinity, while remineralization processes reduce alkalinity**

This will be corrected.

**11 L 332: Another important influence factor in coastal regions is the inflow of terrestrial POC, e.g. in the southern North Sea, leading to the release of $CO_2$ to the atmosphere.**

We will consider to add this in the manuscript.

**12 L 376-377: Are these really the dominant factors? After you argumentation for why the open ocean region is neutral (vertical convection brings up old, DIC rich water which balances the influx during summer) I would expect the absence of this deep mixing in coastal, shallow regions to be one of the major reasons why the coastal regions are a larger sink than the open ocean.**

As shown in Bates (2006), Arrigo et al. (2010), and Ishii et al. (2014), surface DIC concentration is higher over the open, deep basins than the shallow coastal shelf seas of the subpolar Pacific, particularly induced by deep mixing during winter/spring. Also in this period, the coastal sector is covered by seasonal sea-ice resulting in a neutral region of air-sea fluxes while the open sea-ice free ocean (e.g.,the southern Bering Sea) acts as a strong source of $CO_2$. During spring/summer, high $CO_2$ uptake is found in coastal shelf seas influenced by river freshwater (e.g., Beaufort Sea, Arrigo et al., 2010) or by high biological production and sea-ice melt-water (e.g., Bering–Chukchi Shelves and the Gulf of Alaska, Yasunaka et al., 2016).

Thank you for correction. The text in lines 375-377 of the manuscript is revised with a broader context as follows.
Previous text:
*The enhanced uptake of $CO_2$ by the coastal ocean compared to the open ocean results from melt water discharge and high primary production over the shelves of the Chukchi and Bering Seas and the Gulf of Alaska in the spring/summer Yasunaka et al. (2016).*
Revised text:

[Figure]

**Figure RP3.** Yearly global integrated air–sea flux estimates derived from the CMEMS-LSCE-FFNN ensemble (mean ± uncertainty) for 1985–2019. Multivariate El Niño-Southern Oscillation Index (MEI; Wolter and Timlin, 1993, https://psl.noaa.gov/enso/mei/, last access: December 2020) is used to generally indicate a link between variations, e.g. Yearly uptake - Trend , in the CMEMS-LSCE-FFNN sink estimate and the ENSO climate mode (El Niño: MEI > 0.5, La Niña: MEI < -0.5, Neutral: otherwise).

*The annual uptake of $CO_2$ by the coastal shelf seas is much higher than that compared to the open, deep basins as a result of a lower surface DIC concentration induced by winter/spring mixing in the shallower areas and the restriction of seasonal sea-ice on air-sea $CO_2$ exchanges (Bates, 2006; Arrigo et al., 2010; Ishii et al., 2014). Thus, the coastal sector acts as a neutral region of $CO_2$ fluxes in winter (Fig. 8). During spring and summer, a substantial amount of $CO_2$ is also absorbed in the coastal shelf seas influenced by river freshwater (e.g., Beaufort Sea) or by high biological production and sea-ice melt-water (e.g., Bering–Chukchi shelves and the Gulf of Alaska) (Arrigo et al., 2010; Yasunaka et al., 2016).*

**13 To be honest, I can't really see from this figure that it covaries with the ENSO mode. As I see it the flux increases equally often during La Nina as during El Nino. It would be more interesting to see a comparison of the interannual variability with other air-sea flux products.**

The covariate between the ENSO events and the temporal variability of the global carbon sink has been covered by its increasing long-term trend in Figure 9 in the manuscript. We have added another curve whose values are ticked on the right y-axis of the same figure. This curve stands for the yearly flux variability, i.e., the yearly ocean uptake estimate after removing its long-term trend. See Figure RP3 as the revised version of Figure 9.

In-depth intercomparisons amongst different model-based and/or observation-based products are presented in previous works including Rödenbeck et al. (2015); Denvil-Sommer et al. (2019); Hauck et al. (2020). As far as we know, none of these studies shows a comparison of the covariate of the interannual variability of the flux products and the ENSO events. This point raised by the referee is interesting and will be considered in our future studies.

[revised manuscript text omitted]

---

## Author Response (AR1)

**Response to Referee's Comments**

Authors: This document is a point-by-point reply to all referees' comments on our manuscript entitled "*A seamless ensemble-based reconstruction of surface ocean $pCO_2$ and air–sea $CO_2$ fluxes over the global coastal and open oceans*". We split the document into Sections 1, 2, and 3. Throughout this document, the referees' comments are in bold and the revised text included in the manuscript is in italic. The manuscript has been revised corresponding to this point-by-point reply.

**1  #REFEREE 1: The authors use a neural network model to generate a $pCO_2$ product for the global ocean using the SOCAT data, and combine these $pCO_2$ estimates with a wind speed product to compute the $CO_2$ flux. The ensemble model results compare well overall to the observations, and the carbon flux estimates are in-line with the literature. My main comments concern how novel these results are compared to the extensive literature on the topic, and the interpretation of some of the model statistics.**

We would like to thank Referee 1 for constructive comments and suggestions on our study.

**1.1  There is a lot of previous literature using spatially and temporally sparse observations of surface $pCO_2$ to generate global data products and provide estimates of ocean carbon uptake, some of which use very similar methods to those in this present manuscript. The authors cite this previous literature, but there's very little discussion of it. Consequently, I found it difficult to interpret how the present authors' methods and results are novel and differed from these previous studies. The motivation appears to be in lines 41-46, however, I don't follow how the previous literature did not incorporate "space-time varying uncertainty estimates"? It would also appear that the incorporation of the coasts is relatively new, though the authors then cite a few recent studies and declare that it's a closed gap? I suggest that the introduction needs to contain a much clearer description of how the methods used here compare to previous studies, and what is new about this analysis.**

This study is made up of our efforts to reproduce and intensively analyse the spatially and temporally varying surface $pCO_2$ fields, the air–sea $CO_2$ fluxes, and their reconstruction uncertainties over the global ocean. We acknowledge previous studies pursuing the same target and are aware that the existing observation–based mapping methods (for instance proposed by Rödenbeck et al., 2013; Landschützer et al., 2016; Denvil-Sommer et al., 2019; Gregor et al., 2019; Watson et al., 2020) succeeded in obtaining a relatively low misfit between the reconstructed and gridded SOCAT data (see in Rödenbeck et al., 2015; Gregor et al., 2019; Friedlingstein et al., 2020). Although similar interpolation and machine learning approaches (e.g., clustering, classical regression, neural networks) and/or similar sets of predictor variables for $pCO_2$ have been considered in the preceding literature, model design and implementation are still different (e.g., proportion of SOCAT data used in model fitting and evaluation). The present manuscript reflects our vision on the following key features.

i. A design of an ensemble of numerous feed forward neural network (FFNN) models:

It is based on a Monte Carlo approach wherein each model is trained and validated on sub-samples randomly drawn from the monthly gridded SOCATv2020 data and available data of predictors. The ensemble size of 100 is considered in this study. Our proposed ensemble approach was developed at the Laboratoire des Sciences du Climat et de l'Environnement (LSCE) as both an extension and an improvement of the first version (LSCE-FFNN-v1, Denvil-Sommer et al., 2019). Quality assessments comparing these two model versions are documented in Chau et al. (2020). Besides, the proposed approach inherits strengths of the existing statistical models and further aims at reducing mapping uncertainties induced by, for instance, discrete boundaries in the two-step clustering-regression by Landschützer et al. (2016); Gregor et al. (2019) or the two-step FFNN-based reconstruction of $pCO_2$ climatologies and anomalies by Denvil-Sommer et al. (2019). As described in the Method section (2.2) in the manuscript, each FFNN model follows a leave-$p$-out cross-validation approach, i.e., the exclusion of $p$ gridded SOCAT data of the reconstructed month itself in model training and validation. This allows to reduce model over-fitting. In addition, it leaves more independent data for evaluation than previous approaches, results obtained by the proposed reconstruction are in line with the others though (see e.g., Friedlingstein et al., 2020, and references therein).

ii. Quantification and evaluation of model best estimates (ensemble means) and uncertainties (ensemble spreads):

There exists other ensemble-based methods, their concepts and principle objectives are nevertheless different. For example, Gregor et al. (2019) and Gregor and Gruber (2021) introduce machine-learning ensembles with a small ensemble size of different two-step clustering-regression models mapping surface $pCO_2$ and propose the ensemble mean as their model best estimate. In a broader context, Rödenbeck et al. (2015) suggest an intercomparison of multiple mapping methods targeting the identification of common or distinguishable features of different mapping results. Hauck et al. (2020) and Friedlingstein et al. (2020) synthesize $pCO_2$ mapping products and refer to an ensemble of their observation–based estimates of air-sea $CO_2$ fluxes as a benchmark to compare with the one derived from ocean biogeochemical models.

Up to recently, most of these studies have used misfits between the reconstructed and observation–based data (e.g., the root-mean-square deviation, RMSD) to evaluate product quality and infer uncertainty estimates of the reconstructed $pCO_2$. Reconstruction errors of $pCO_2$ are then propagated to get uncertainty estimates of the reconstructed $CO_2$ fluxes (see in Landschützer et al., 2014, for instance). By construction, such uncertainty estimates are restricted to oceanic regions and periods when observations are available (Lebehot et al., 2019; Hauck et al., 2020), and the uncertainty quantification of an averaged $pCO_2$ or an integrated flux is under low confidence due to sparse data density. An advantage of our approach is that an ensemble of 100 model outputs of $pCO_2$ and $CO_2$ fluxes is available at each $1° \times 1°$ ocean grid cell of the globe for each month in the period 1985–2019. The ensemble asset facilitates the quantification of model uncertainty of $pCO_2$ and $CO_2$ fluxes averaged or integrated over space and time of interest (see for instance Figures 5 and 9 in the manuscript). This is expected to provide more robust estimates than the ones based on reconstruction errors.

iii. Seamless analysis of the reconstructed data and uncertainty estimates over the open ocean and coastal zones:
An in-depth analysis has been made and presented for both the open and coastal regions divided by latitude bands. Interpretations of good or poor reconstructions of surface $pCO_2$ and air-sea $CO_2$ fluxes (e.g., data density and distribution, regional to local characteristics of $pCO_2$ and its potential drivers, model design and resolution) and changes in spatial and seasonal variations of $CO_2$ fluxes are given. To strengthen our interpretation, we have shown both the temporal and spatial distribution of the reconstructed $pCO_2$ and $CO_2$ fluxes fields, model-data misfits, model uncertainty, and linked these materials with their driving mechanisms suggested in previous literature. More importantly, we have made an intercomparison of model reconstruction ability between regions, identified oceanic sectors where the model does not fit the data well, and suggested further improvements on the data reconstruction based on the proposed space-time varying uncertainty fields.

With these points involved in the manuscript, we believe that our study is novel and statistics and keys findings therein would be useful contributions for the marine science community. However, we agree with the referee that the first version of the manuscript missed part of discussions on the comparison among the existing methods, and thus the novelty of this study was not bold. We consider this referee's feedback important and it has been taken into account in our revision. Precisely, we have reworked on the last two paragraphs in the Introduction section (Lines 37-55 of the first manuscript). The new paragraphs are produced below.

[revised manuscript text omitted]

**1.2   I think the methods section is missing a few key details that will help support this manuscript.**

- **First, it would be helpful for the authors to explain how to interpret and compare the RMSD and $r^2$ values for each region. The reason being, that these values are listed for each ocean region, but it's a little unclear what**
135         **differences in these values between regions is saying about the model estimate. For example, I was surprised by how low the RMSD value for the Southern Ocean is (slightly lower than the global mean), despite the somewhat limited observation–based data and well documented, substantial inter-annual variability. However, the Southern**

**Ocean does have a lower $r^2$ value, which the authors seem to attach a greater weight to in their interpretation. Second, I'm a little confused by equation (2). Why is the equation for the mean squared deviation (MSD) shown when it's the root mean squared deviation (RMSD) which is calculated throughout the manuscript? Also, the text refers back to this equation for the definition of the $\sigma$ misfit, but this definition is itself within the MSD equation and is not clearly labeled on its own.**

We have revised the manuscript and added in Section Methods details of the statistics used in this study to facilitate the readers' interpretation of our results (see the new Section 2.4 reproduced below). In general, RMSD measures the model skill in terms of mean distance between model estimates and evaluation data while $r^2$ measures the proportion of data variation predicted by the model. RMSDs between the model and SOCAT gridded data over the Southern Ocean (open: 19.18 $\mu$atm, coastal: 35.73 $\mu$atm) are slightly higher [lower] than the global errors (open: 17.87 $\mu$atm, coastal: 35.86 $\mu$atm) for the open ocean [coastal zone], but the regional $r^2$ values (open: 0.62, coastal: 0.65) are lower than the global ones (open: 0.78, coastal: 0.70). The global scores involve the ones of all the regions, where the poorest reconstruction were found over the Arctic, subpolar, and coastal regions. Compared to other metrics such as model bias and $r^2$, RMSD takes another role as an outlier detector of model-data misfits which gives larger weights to such high errors over these regions. Yet, data sampling is limited over the Southern Ocean similar to polar/subpolar and coastal regions. We have also learned that the interannual variability of $pCO_2$ over the Southern Ocean is moderate compared to that over the Equatorial Pacific and polar/subpolar regions (see also in Rödenbeck et al., 2015; Denvil-Sommer et al., 2019). However, air-sea fluxes vary greatly over the Southern Ocean (SO), we also show that the SO RMSD between our fluxes and SOCAT-based estimates are larger than those of certain regions (Table S2).

The statistics (e.g., Bias, RMSD, $r^2$, and number of data grided from SOCAT observations) listed in Table S2 and scattered in Figure 3c for different open and coastal regions provide a general comparison of the reconstruction skill of the CMEMS-LSCE-FFNN model among the oceanic basins. Nevertheless, examining merely these numbers would not give us a robust assessment of the full story behind. As one of the contributions of this study compared to the heretofore publications, an intensive analysis of the data reconstruction has been made and presented in Sections 3.1.2 and 3.1.3 for both the open ocean and the coastal zones. Interpretations of key factors driving a good or poor reconstruction of surface $pCO_2$ (e.g., data density and distribution, regional to local characteristics of $pCO_2$ and its potential drivers, model design and resolution) are given. To strengthen our interpretations, we have shown both the temporal and spatial distribution of SOCAT data, model-data errors, model uncertainty and scattered them with their driving mechanisms suggested in the literature. Based on these materials, we have made an intercomparison of model reconstruction ability between regions, identified oceanic sectors where the model does not fit the data, and importantly we have suggested improvements on the data reconstruction.

The precise definitions of $\sigma_{\mathrm{misfit}}$ and the root mean squared deviation (RMSD) are given in the revised manuscript. We have rewritten Section 2.4 (Statistics). The new Section 2.4 is as follows.

Lines 148-173 in the revised manuscript:
*The mean ($\mu$) and standard deviation ($\sigma$) of the 100-member ensembles of $pCO_2$ and $fgCO_2$ are respectively chosen as their best estimate and the associated uncertainty. Unless stated otherwise, a model best estimate and its uncertainty computed at each desired space-time resolution are denoted by $\mu_{\mathrm{ensemble}} \pm \sigma_{\mathrm{ensemble}}$, where*

$$\mu_{\mathrm{ensemble}} = \frac{\sum_{i=1}^{i=100} pCO_2^{\mathrm{Reconstruction}(i)}}{100}, \quad \sigma_{\mathrm{ensemble}} = \sqrt{\frac{\sum_{i=1}^{i=100}\left(pCO_2^{\mathrm{Reconstruction}(i)} - \mu_{\mathrm{ensemble}}\right)^2}{100}}, \quad (2)$$

*and $pCO_2^{\mathrm{Reconstruction}(i)}$ is one of the 100 members of the reconstructed $pCO_2$ fields. Similar definitions are applied for $fgCO_2$. The units of air-sea flux estimates is $\mathrm{molC\,m^{-2}yr^{-1}}$ for a flux density and converted to $\mathrm{PgC\,yr^{-1}}$ for an integral over a region or the global ocean.*

*Model robustness of the reconstructed $pCO_2$ fields is evaluated on the gridded SOCAT data and in situ observations (Sutton et al., 2019). The evaluation data is denoted as $pCO_2^{\text{Observation}}$ in the following formulas. Standard statistics include the coefficient of determination ($r^2$), misfit mean (model bias) and misfit standard deviation,*

$$\mu_{\text{misfit}} = \frac{\sum_{j=1}^{j=N} \mathrm{d}pCO_2^j}{N}, \quad \sigma_{\text{misfit}} = \sqrt{\frac{\sum_{j=1}^{j=N} \left(\mathrm{d}pCO_2^j - \mu_{\text{misfit}}\right)^2}{N}}, \tag{3}$$

185     *and the root-mean-square deviation (RMSD)*

$$\text{RMSD} = \sqrt{\frac{\sum_{j=1}^{j=N} \left(\mathrm{d}pCO_2^j\right)^2}{N}}, \tag{4}$$

*where $\mathrm{d}pCO_2^j = pCO_2^{\text{Reconstruction}}[j] - pCO_2^{\text{Observation}}[j]$, and $N$ is a number of evaluation data. All these scores are computed for different coastal and open regions from the scale of grid cells to the global scale.*

*Generally, RMSD measures the reconstruction skill in terms of mean distance between model estimates and evaluation*
190     *data while $r^2$ measures the proportion of data variation predicted by the model. Compared to other metrics such as mean absolute bias and $r^2$, RMSD takes another role, an outlier detector, which gives larger weights to high model–data misfits. Note that $r^2$, $\mu_{\text{misfit}}$, $\sigma_{\text{misfit}}$, and RMSD reflect the model performance with respect to evaluation data, while $\sigma_{\text{ensemble}}$ measures the stability of the model best estimate $\mu_{\text{ensemble}}$. Nevertheless, these different statistics should consistently reflect the skill of the model reconstruction, e.g., depending on the density and distribution of data sampling.*

195     *In the next section, both the temporal and spatial distributions of gridded SOCAT data and in situ observations, model–data errors, model best estimates and uncertainties are shown. An intensive analysis is presented for both the open ocean and the coastal zones. We then interpret key factors leading to a good or poor reconstruction of surface $pCO_2$ and $fgCO_2$, e.g., SOCAT data density and distribution, model design and resolution, regional to local characteristics of $pCO_2$ and $fgCO_2$, and their potential driving mechanisms.*

200     • **Lastly, I think the description of the wind speed product used should be included in the main text rather than the supplementary, considering that this will have a large impact on the overall flux numbers (which the authors do highlight in the results).**

The wind speed product is now added in the main text.

205     Lines 91-92 in the revised manuscript:
*$k$ is the gas transfer velocity computed as a function of the 10-meter ERA5 wind speed (Hersbach et al., 2020) following Wanninkhof (2014) and its coefficient is scaled to match a global mean transfer velocity of 16.5 $\text{cm h}^{-1}$ (Naegler, 2009).*

**1.3**    **I suggest re-working the 2nd paragraph of the abstract. This paragraph currently reads like a laundry list of different regions and where they fall in terms of largest total source/sink, largest flux density source/sink, along**
210     **with coastal and open ocean qualifiers. This many iterations of "X is the greatest . . . " makes it difficult to follow-along and is not particularly interesting (e.g. the equatorial Pacific as the strongest source of carbon to the atmosphere is not a surprising result). Instead, highlight some of the other key findings, like the increase in ocean carbon uptake over the 1985-2019 timeframe (right now the mean is just listed, but the change is highlighted in the conclusion).**

215 The second paragraph of the abstract has been modified. A new version of the full abstract is reproduced below.

Modified abstract:
*We have estimated global air–sea $CO_2$ fluxes ($fgCO_2$) from the open ocean to coastal seas. Fluxes and associated uncertainty are computed from an ensemble-based reconstruction of $CO_2$ sea surface partial pressure ($pCO_2$) maps trained with*

220   *gridded data from the Surface Ocean $CO_2$ Atlas v2020 database. The ensemble mean (which is the best estimate provided by the approach) fits independent data well and a broad agreement between the spatial distribution of model–data differences and the ensemble standard deviation (which is our model uncertainty estimate) is seen. Ensemble-based uncertainty estimates are denoted by $\pm 1\sigma$. The space-time varying uncertainty fields identify oceanic regions where improvements in data reconstruction and extensions of the observational network are needed. Poor reconstructions of $pCO_2$ are primarily found over the*

225   *coasts and/or in regions with sparse observations, while $fgCO_2$ estimates with largest uncertainty are observed over the open Southern Ocean ($44°S$ southward), the subpolar regions, the Indian gyre, and upwelling systems.*

    *Our estimate of the global net sink for the period 1985–2019 is $1.643 \pm 0.125 \, \mathrm{PgC\,yr^{-1}}$ including $0.150 \pm 0.010 \, \mathrm{PgC\,yr^{-1}}$ for the coastal net sink. Among the ocean basins, the subtropical Pacific ($18°N$–$49°N$) and the subpolar Atlantic ($49°N$–$76°N$) appear respectively to be the strongest $CO_2$ sinks for the open ocean and the coastal ocean. Based on mean flux density per*

230   *unit area, the most intense $CO_2$ drawdown is, however, observed over the Arctic ($76°N$ poleward) followed by the Subpolar Atlantic and Subtropical Pacific for both open ocean and coastal sectors. Reconstruction results also show significant changes in the global annual integral of all open- and coastal-ocean $CO_2$ fluxes with a growth rate of $+0.062 \pm 0.006 \, \mathrm{PgC\,yr^{-2}}$ and a temporal standard deviation of $0.526 \pm 0.022 \, \mathrm{PgC\,yr^{-1}}$ over the 35-year period. The link between its large interannual to multi-year variations and the El Niño-Southern Oscillation climate variability is reconfirmed.*

235

**OTHER COMMENTS**

**1.4   Lines 37-40: Should all the manuscripts be separated with a comma rather than a semicolon? And why is the Rödenbeck et al. (2015) manuscript specifically highlighted as "other mapping methods"?**

The comma is now used to separate the references if they are part of the sentence. Rödenbeck et al. (2015) is cited in the
240   manuscript as one of the studies which made an intercomparison between different observation–based mapping methods reconstructing ocean surface $pCO_2$ and quantifying $CO_2$ fluxes. However, we have changed the text in the Introduction (see our reply to Referee's comment **1.1**).

**1.5   Line 139: How is the variability in "analytical equipment" accounted for here?**

The word "analytical equipment" was not appropriate in the context of the sentence in Lines 139-141 of the first manuscript.
245   Thank you for pointing it out. We have rewritten this sentence as follows. Temporal sampling bias is also a source of uncertainty, it is now added in this sentence as suggested by Referee 3 (comment **3.7**).

Lines 197-200 in the revised manuscript:
*Variability in the sampling time and location of cruises and instruments induces temporal sampling bias (e.g., towards some*
250   *days in a month and/or the summer months at high latitudes) and latitude and longitude offsets from the cell center (e.g., with an average of $0.34° \pm 0.14°$ as reported in Sabine et al., 2013) which are not taken into account.*

**1.6   Figure 2: I suggest directly labeling each region in the figure with the abbreviated label (i.e. SpA for subpolar Atlantic) for clarity. Figure 5 and 8: The tick marks in the colorbar for these figures are relatively large and look like a negative sign, I'd suggest making them much smaller.**

255   The label of Figure 2 is now changed from the numbers to the abbreviated names of 11 regions. The size of tick marks in the colorbar of all the figures is also reduced.

**2   #REFEREE 2: At present, there are many data products in marine physics, such as temperature and salinity products, but there are few data products in marine chemistry. I support the publication of more marine chemistry data products.**

We thank Referee 2 for his/her interest in marine chemistry data products and comments/suggestions on our study.

**2.1   The author reconstructed surface ocean $p\text{CO}_2$ based on FFNN with region divided by latitudes and similar predictors with previous researches was used, which is not novel.**

With the proposed ensemble-based mapping method, statistics, and keys findings presented in the manuscript, we believe that our study is novel and would be a valuable contribution for the marine science community. However, we admit that the first version of the manuscript missed part of discussions on the comparison among the existing methods, and thus the novelty of this study was not easy to interpret. We consider this Referee's feedback important and have revised the manuscript in such a way that our three main contributions (see Lines 30-72 in this document and our reply to comment **2.5** below) are elaborated and highlighted. We added relevant information to the last paragraphs in Section Introduction (Lines 35-85 in the revised manuscript). Note that oceanic regions divided by latitude bands are only used for the analysis of our results, FFNN models themselves do not follow oceanic regions or biomes in clustering before training as opposed to Landschützer et al. (2016) and Gregor et al. (2019). The Referee's comment **"The author reconstructed surface ocean $p\text{CO}_2$ based on FFNN with region divided by latitudes"** seems to be misleading.

**2.2   The reconstruction of $p\text{CO}_2$ and sea-air $\text{CO}_2$ flux over global coastal oceans are interesting works but the author needs to do much more works on the validation of coastal results. Because a standard deviation of 41.79 $\mu$atm between $p\text{CO}_2$ results and SOCAT observations possibly leads to opposite results in the estimate of coastal $\text{CO}_2$ flux.**

After the introduction of our new ensemble-based approach, the current manuscript indeed presents numerous results and an intense analysis for evaluating our global reconstruction of monthly $p\text{CO}_2$ and fluxes from the open ocean to the coastal zone. The coastal-ocean reconstruction is compared with monthly gridded SOCAT data (not used in our model fitting) and with the open-ocean reconstruction. We compute and analyze the estimates of coastal $p\text{CO}_2$ and air–sea fluxes, their model errors, and model uncertainties for different time scales (e.g., monthly, yearly, and multi-decadal) and spatial scales (e.g., grid cells, sub-basins, and the global ocean) (see Figures 4-8 and Table 2 in the main text and more in the Supplementary). This is one of the novel contributions of this study which complement to the existing ones focusing on analyzing the spatial distribution and/or a monthly climatology of their coastal estimates (see our interpretation in the Introduction in the revised manuscript).

Referee 2 explains that **a standard deviation of 41.79 $\mu$atm between $p\text{CO}_2$ results and SOCAT observations possibly leads to opposite results in the estimate of coastal** $\text{CO}_2$ **flux**. This seems to be misleading. Indeed, we have written in the manuscript:
Lines 184-188 in the revised manuscript (Lines 126-130 in the first manuscript):
*The reconstructed $p\text{CO}_2$ field matches SOCAT data well: both are normally distributed with the same mean of 361.3 $\mu$atm (Fig. 3a) and a high agreement for all percentiles (Fig. 3b) is seen. The slight under- or overestimation at high and low percentiles implies that the model is slightly biased towards the mean value, as is expected when predictor variables do not fully explain predictand variables in the training dataset. This reduced variability is also reflected in the difference between the data standard deviation based on SOCAT $p\text{CO}_2$ (41.79 $\mu$atm) and the one based on CMEMS-LSCE-FFNN (36.30 $\mu$atm).*
In this context, 41.79 $\mu$atm is the standard deviation of SOCAT data itself, it is not the standard deviation of differences between the reconstructed and SOCAT data.

However, in this revision, we added a new subsection (3.1.3 Time series stations) including model evaluation on data sampled at in situ stations (Sutton et al., 2019). This would facilitate for the readers qualifying our product (see this new subsection in Lines 360-384 of the revised manuscript; text below). Consequently, we reorganise Section 3.1 (Evaluation) as follows:

- Section 3.1.1 Global ocean remains the same as in the first manuscript.

- Section 3.1.2 Ocean basins comprises the model evaluation for regions in the Arctic, Atlantic, Pacific, Indian Ocean, Southern Ocean.

- Section 3.1.3 Time series stations (new in this revision) includes the model evaluation on both open and coastal data sampled at in situ stations (Sutton et al., 2019).

As part of this new section, our coastal-ocean reconstruction is evaluated on data sampled at the time series stations (Figures S5). Despite less skill than the ocean-ocean reconstructions (Figures S6 and S7), our coastal-ocean reconstructions are rather compatible with observation–based $pCO_2$ data (Figure S8). In general, all reconstructed time series cover the full period 1985-2019 and would therefore be useful for estimating long-term means, trends, and variations of $CO_2$ surface partial pressure and ultimately the corresponding air-sea fluxes. observation–based data are still sparse and mostly distributed over the past two decades (see also the data density in Figures S1, S3, and S4 in the Supplementary), densifying observation networks is in priority to provide a better validation of both coastal- and open-ocean data reconstructions.

[Figure]

**Figure S5.** Location map of in situ measurements of ocean surface $pCO_2$ (Sutton et al., 2019).

Lines 360-384 in the revised manuscript (Section 3.1.3 Time series stations):

*CMEMS-FFNN-LSCE estimates of $pCO_2$ are now compared with moored $pCO_2$ time series provided by Sutton et al. (2019). This data product comprises $pCO_2$ measurements collected from a wide range of oceanic regions since 2004 (Figs. S5–S8). Most of the stations were established in the North Atlantic and the North and Equatorial Pacific, one site is in the IO and another in the SO. Approximately one third of Sutton et al. (2019) sites belong to the coastal seas and shelves (Fig. S8). Table S3 details the information of the moored $pCO_2$ time series.*

*Observation–based data used for model–data comparison (black points in Figs S6–S8) are monthly averages of $pCO_2$ measurements at each site. This interpolation results in monthly time series with a number of data $N$ between 9 (NH10) and 98 (WHOTS). The ensemble mean $\mu_{\mathrm{ensemble}}$ and ensemble spread $\sigma_{\mathrm{ensemble}}$ (Eq. 2) are computed from the CMEMS-LSCE-FFNN ensemble of model outputs at the four nearest model grid boxes of each location. Results confirm a reasonably good reconstruction of the proposed approach. The model best estimates (coloured thick lines) characterise $pCO_2$ trends and variations of in situ data well and the model ensembles almost catch the observation–based data in their 99% confidence interval (light shaded envelop). Over 90% of the time series stations, the model estimation obtains a moderate to high coefficient of determination $r^2$ with a linear model–data correlation $r$ larger than 0.5 (e.g., BTM: 0.98, CRESCENTREEF: 0.92, HOGREEF: 0.84, SOFS: 0.79, TAO110W: 0.75, WHOTS: 0.73). Mean bias $\mu_{\mathrm{misfit}}$ (Eq. 3) and RMSD (Eq. 4) are relatively low compared to mean $pCO_2$ values of the time series stations.*

*Half of the open-ocean reconstructions have model errors less than 20 $\mu atm$ and even less than 10 $\mu atm$ at KEO, PAPA, SOLS, STRATUS, and WHOTS (Figs S6 and S7). Despite less skill than the open-ocean reconstructions, the coastal-ocean reconstructions are quite compatible with the in situ data (Fig. S8). Most of RMSDs remain lower than 20% of the mean $pCO_2$ values of coastal time series (e.g., CCE2: 36.53 $\mu atm$, ICELAND: 12.26 $\mu atm$, M2: 36.58 $\mu atm$). For some other stations*

*in the US west coast and the oceanic regimes of coral reef, the estimates differ from the observation–based data in terms of*
335 *magnitude of $p$CO$_2$ (e.g., CRIMP2, LA PARGUERA) and/or of its seasonal cycle (e.g., CHABA, CHEECAROCKS, SEAK).*

*The reconstructed time series cover the full period 1985-2019 while observation–based data are still sparse and almost distributed over the past two decades (Figs. S6-S8). The CMEMS-LSCE-FFNN time series would be useful for estimating and assessing long-term means, trends, and variations of $CO_2$ surface partial pressure and the corresponding air-sea fluxes.*

**2.3 The CHL data used was only from 1992 to 2019, and was not available in the Arctic and the Southern Ocean in**
340 **winter, the details about how the reconstruction was carried out when CHL was not available should be declared clearly in the method section.**

This information is now added in the Method section (see the italic text below). To be precise, climatologies based on all available CHL data (1998-2019) were used as predictors for data unavailable before 1998. We set CHL approximately to 0 mg m$^{-3}$ over the Arctic and the Southern Ocean in winter when no data are available (e.g., Landschützer et al., 2016; Denvil-Sommer
345 et al., 2019; Gregor et al., 2019).

Lines 107-109 in the revised manuscript:
*CHL was set approximately to 0 mg m$^{-3}$ over the Arctic and the Southern Ocean winter when no data is available. In case of data unavailable before 1998, climatologies based on all available data were used as predictors.*

350 ### 2.4 The subskin temperature correction (Watson et al., 2020) should be considered in the estimate of sea-air $CO_2$ flux.

Watson et al. (2020) proposed a double correction to the SOCAT data and to the computation of the $CO_2$ flux in order to remove some aliasing caused by the temperature vertical gradient within the marine boundary layer. However, the Watson et al. (2020) adjustment, if applied here, would add roughly $0.9$ PgCyr$^{-1}$ to the global ocean sink estimate based on observations. The
355 adjusted ocean sink estimate would thus surpass the land sink and result in a large carbon budget imbalance. More evidence of their genericity is needed before applying skin and subskin corrections (Friedlingstein et al., 2020).

**2.5 The author should reconsider the topic of this manuscript. If the author want focus on the $CO_2$ flux of global open oceans, additional work was necessary rather than only discussing spatial distribution or interannual variability, because the reconstruction method in this manuscript and the results was not novel. If the author**
360 **want focus on the $CO_2$ flux of global coastal oceans, which was still a research gap, much more works are needed to make the result convince.**

As mentioned in our response to Referee's comments **1.1** and **2.1**, the manuscript presents three main contributions:

  i. Our data reconstruction is based on a new model design - an ensemble of 100 neural network models.

  ii. We quantify and evaluate model best estimates and uncertainties based on the ensemble asset. For the first time, the
365    space-time varying uncertainty estimates (see for instance the Introduction - Lines 35-85 - and Figures 5 and 9 in the revised manuscript) derived from the ensemble of model outputs are presented and analysed. We promote the use of the proposed uncertainty fields which would be more informative than the RMSD-based uncertainty fields (e.g., proposed in Landschützer et al., 2014) in identifying oceanic sectors where further improvements on the data reconstruction will be needed.

370  iii. More importantly, the manuscript presents a seamless analysis of the reconstructed data and uncertainty estimates over the open ocean and coastal zones. Furthermore, the open ocean estimates are considered as references for the coastal data assessment.

We believe that the ensemble-based approach and analysis therein are novel and the Introduction section has been changed to better highlight these contributions. Also, we have added a new section (3.1.3) presenting an evaluation of the reconstructed

375  data on independent in situ observations in the revised version of the manuscript. Precisely, we propose to add Figures S5-S8, Table S3, and an interpretation of these results (see also in our reply to Referee's comment **2.2**).

**2.6   Line 76: "An ensemble of 100 FFNNs was used to reconstruct monthly $p$CO$_2$ fields......", How are these 100 models built? Why did you do that? How are the results of 100 models selected? Line 84-85: "The random extraction and the FFNN training were repeated 100 times so that 100 versions of the monthly FFNNs have been obtained", Why is it the "100 times"? How is the "100 times" determined? Does it converge after 100 iterations?**

380

The description of the construction of the ensemble approach is already given in the manuscript as follows (Figure 1 is used to illustrate a neural network model mapping the target $p$CO$_2$ and predictor variables).

Lines 117-120 in the revised manuscript (Lines 81-84 in the first manuscript):
385  *To reconstruct the $p$CO$_2$ fields over the global ocean for each target month over the 1985–2019 period, all the available SO-CAT data and the co-located predictors have been collected for the month before and the month after the target month. We randomly extracted two thirds of each one of these datasets to make training datasets for the FFNNs, leaving the remaining third to be corresponding test datasets. The FFNNs were then trained for each target month.*

390  Our ensemble approach comprises multiple network models, each trained and validated on resampled data of SOCAT $p$CO$_2$ and predictors. We added Figure S2 in the Supplementary document and a new paragraph in (text below) in this revision of our manuscript, explaining the reasons we have selected 100 model runs for our study.

[Figure]

**Figure S2.** RMSD between a best estimate (ensemble mean) and SOCAT data of ocean surface $p$CO$_2$ with respect to the ensemble size in $\{5, 10, 20, 50, 75, 100\}$.

Lines 124-134 in the revised manuscript:
395  *The random extraction and the FFNN training were repeated 100 times so that 100 versions of the monthly FFNNs have been obtained. Note that our ensemble approach belongs to the classes of bootstraping and Monte Carlo methods in statistics. Theoretically, the number of samples or the ensemble size must be substantially large to get a convergence. However, it was demonstrated in the literature (e.g., Goodhue et al., 2012; Efron et al., 2015) that with the ensemble size of 50 the model estimation is likely stable and with the ensemble size over 100 the improvement in standard errors between model outputs*
400  *and evaluation data is negligible. Fig. S2 shows an illustration of the reconstruction skill with respect to the ensemble size $S$. For each ensemble of $S$ model outputs of $p$CO$_2$ ($S \in \{5, 10, 20, 50, 75, 100\}$), the root-mean-square deviation (RMSD) is computed between the ensemble mean (our best model estimate) and SOCAT data over the period 1985-2019. As seen in this figure, the reconstruction starts to stabilize with $S = 50$. In this study, we have exploited a large but realistic amount of computing resources to run an ensemble of $S = 100$ neural network models.*

**2.7  Table 2, What is the meaning of two numbers in the rightmost column of Table 2, for example: $0.07 \pm 0.04$, $0.30 \pm 0.13$**

The numbers without brackets (e.g., $0.07 \pm 0.04$, $0.30 \pm 0.13$) in the rightmost column of Table 2 in the manuscript refer to the estimates derived from observation–based methods. In the caption of Table 2, we wrote: *In column 'RECCAP1', values in parentheses are the 'best' estimates proposed by RECCAP1 studies, the others are the estimates computed with different methods using $p$CO$_2$ observations.* More information is now added to the caption of Table 2.

Modified caption of Table 2:
*Yearly mean of contemporary air–sea $CO_2$ fluxes ($PgC\,yr^{-1}$) integrated over the global ocean and 11 RECCAP1 regions. Mean estimate and uncertainty ($\mu_{\mathrm{ensemble}} \pm \sigma_{\mathrm{ensemble}}$) of the CMEMS-LSCE-FFNN approach is shown for the coast (C), the open ocean (O), and the total area (T). For a comparison, estimates derived from RECCAP1 (Canadell et al., 2011; Schuster et al., 2013; Ishii et al., 2014; Sarma et al., 2013; Lenton et al., 2013; Wanninkhof et al., 2013) are provided. In column 'RECCAP1', values in parentheses are the 'best' estimates proposed by RECCAP1 studies which were derived from averages or medians of estimates based on the $p$CO$_2$ climatology or $p$CO$_2$ diagnostic model, and/or the atmospheric and ocean inversions, and GOBM models. The 'RECCAP1' values out of parentheses are the estimates derived from different methods mapping observation–based data of $p$CO$_2$. With an exception for the global estimate\* (Wanninkhof et al., 2013), those of the RECCAP1 sub-basins are available only for the open ocean.*

**2.8  Line 444-446: "The global open ocean uptake obtained in this study of $1.344 \pm 0.111$ PgC yr$^{-1}$ lies between the observation based net sink estimate by Wanninkhof et al. (2013) ($1.18$ PgC yr$^{-1}$) and the global sum of regional best estimates given in Table 2 ($1.8$ PgC yr$^{-1}$)". In table 2, I can't find the value of $1.8$ PgC yr$^{-1}$**

It means that $1.8$ PgC yr$^{-1}$ is the sum of all the 'best' estimates (between brackets) given in Table 2.

**2.9  Line 462-463: The discrepancy is possibly due to an overestimation of Arctic $p$CO$_2$ by the CMEMS-LSCE-FFNN (see in Sect. 3.1.2) and to the lack of estimates over a large portion of the seasonally sea–ice covered regions. This sentence means that the data in the Arctic are not accurate at present. So the data in the Arctic is not suitable for use at present.**

Results shown in Sect. 3.1.2 in the manuscript confirm that the model reconstruction of $p$CO$_2$ over the Arctic does not fit SOCAT data well and is much more uncertain than for other oceanic regions. The factors behind the poor estimates of Arctic $p$CO$_2$ have been further discussed in the Discussion section (Lines 460-469 in the first manuscript, Lines 555-564 in the revised manuscript). Despite the need for further improvements, our analysis fairly documents the current status and discusses the way forward.

[Figure]

**Figure S6.** Time series of open ocean surface $pCO_2$ at different stations - part 1 (see station locations in Fig. S5 and Table S3). Evaluation data are monthly averages of measurements at each station (Sutton et al., 2019). The ensemble mean $\mu_{ensemble}$ and ensemble spread $\sigma_{ensemble}$ (Eq. 2) are computed from reconstructed data at the four nearest neighbors of that location. Number of grid boxes with observations $N$, model bias $\mu_{misfit}$ (Eq. 3), RMSD (Eq. 4), and model–data correlation $r^2$ have been computed on these monthly interpolated data. In each subplot, dots stand for observation–based data and the coloured line with shaded areas stand for the mean and uncertainty envelops computed from the CMEMS-LSCE-FFNN 100-member ensemble (dark: $68\%$ confidence interval, i.e. $\mu_{ensemble} \pm \sigma_{ensemble}$; light: $99\%$ confidence interval, i.e. $\mu_{ensemble} \pm 3\sigma_{ensemble}$).

[Figure]

**Figure S7.** Time series of open ocean surface $pCO_2$ at different stations - part 2 (see station locations in Fig. S5 and Table S3). Evaluation data are monthly averages of measurements at each station (Sutton et al., 2019). The ensemble mean $\mu_{\mathrm{ensemble}}$ and ensemble spread $\sigma_{\mathrm{ensemble}}$ (Eq. 2) are computed from reconstructed data at the four nearest neighbors of that location. Number of grid boxes with observations $N$, model bias $\mu_{\mathrm{misfit}}$ (Eq. 3), RMSD (Eq. 4), and model–data correlation $r^2$ have been computed on these monthly interpolated data. In each subplot, dots stand for observation–based data and the coloured line with shaded areas stand for the mean and uncertainty envelops computed from the CMEMS-LSCE-FFNN 100-member ensemble (dark: 68% confidence interval, i.e. $\mu_{\mathrm{ensemble}} \pm \sigma_{\mathrm{ensemble}}$; light: 99% confidence interval, i.e. $\mu_{\mathrm{ensemble}} \pm 3\sigma_{\mathrm{ensemble}}$).

[Figure]

**Figure S8.** Time series of coastal ocean surface $pCO_2$ at different stations (see station locations in Fig. S5). Evaluation data are monthly averages of measurements at each station (Sutton et al., 2019). The ensemble mean $\mu_{\mathrm{ensemble}}$ and ensemble spread $\sigma_{\mathrm{ensemble}}$ (Eq. 2) are computed from reconstructed data at the four nearest neighbors of that location. Number of grid boxes with observations $N$, model bias $\mu_{\mathrm{misfit}}$ (Eq. 3), RMSD (Eq. 4), and model–data correlation $r^2$ have been computed on these monthly interpolated data. In each subplot, dots stand for observation–based data and the coloured line with shaded areas stand for the mean and uncertainty envelops computed from the CMEMS-LSCE-FFNN 100-member ensemble (dark: $68\%$ confidence interval, i.e. $\mu_{\mathrm{ensemble}} \pm \sigma_{\mathrm{ensemble}}$; light: $99\%$ confidence interval, i.e. $\mu_{\mathrm{ensemble}} \pm 3\sigma_{\mathrm{ensemble}}$).

**3 #REFEREE 3 (Meike Becker): The authors present an estimate of global air-sea $CO_2$ fluxes based on interpolating gridded SOCAT $pCO_2$ data. They use an ensemble of 100 feed-forward neural network models (FFNN) and sea surface height, sea surface temperature, sea surface salinity, mixed layer depth, chlorophyll-a, atmospheric mole fractions, a $pCO_2$ climatology and position data as drivers. They present an uncertainty analysis based on their ensemble spread as a semi-independent parameter, which is better than many available air-sea flux products. However, there are a few things that should be improved.**

We would like to thank Meike Becker (Referee 3) for her positive feedback and suggestions.

**3.1 One point that needs improvement is the description of the driving data, where some important information is missing. The driving data that were used are not available for the full period for which the authors present flux maps. How did you deal with that? Did you use a climatology for CHL, SST, MLD etc. before the early/mid 90s? If so, what was this based on? This information is crucial for interpreting interannual variability prior to the mid-90s.**

The information of all data used in our reconstruction has been presented in Table S1 (Supplementary material). As shown in this table, monthly data of sea surface temperature (SST) and atmospheric mole fractions ($xCO_2$) which are possibly key drivers to trends and interannual variability of the $pCO_2$ field are available over the full period (1985-2019). It is also noted at the end of Table S1: ***For some data unavailable before 1998, climatologies based on all available data were used as predictors. Exceptionally, predictors for SSH before 1993 are climatologies plus a linear trend in order to retain the overall response to global warming. MLD before 1992 was taken as the average MLD between 1992 and 1997.** We, however, agree with the referee to make this information clearer to the readers. This information is now added to the main text.

Lines 107-111 in the revised manuscript:
*CHL was set approximately to $0 \, \mathrm{mg \, m^{-3}}$ over the Arctic and the Southern Ocean winter when no data is available. In case of data unavailable before 1998, climatologies based on all available data were used as predictors. Exceptionally, predictors for SSH before 1993 were climatologies plus a linear trend in order to retain the overall response to the global warming. MLD before 1992 was taken as the average MLD between 1992 and 1997.*

**3.2 Another thing that I want to point out, is the inconsistent and partly misleading use of the terms 'observations', 'sample' and 'data'. The authors base their product on a gridded version of the SOCAT data set (monthly, 1x1). In order to avoid confusion, the term 'observations' should be reserved for data that has been retrieved from field work, in the case the original $pCO_2$ measurements in the SOCAT database. The gridded version contains monthly, 1x1 averages of these $pCO_2$ measurements. When the authors write about 'X observations' in a certain region, they actually mean 'grid boxes with observations'. Please make sure that this becomes clearer. In line 195 for example, the authors write 50 to 220 samples per year'. Here the authors should specify that they mean 'grid boxes with data' as the reader easily can assume that there were only 50-220 $pCO_2$ observations every year.**

Thank you for highlighting this. We agree that the wording has to be carefully selected. We went through the manuscript and refer now to "observation–based data" rather than "observations" (see for instance our correction in Lines 71-82 in this document).

**3.3 I also want to comment on Figure S1. Here the authors show the coverage of the gridded SOCAT product and its variability where they mention '$pCO_2$ individuals'. I don't understand if this means the original SOCAT $pCO_2$ observations (i.e. a measure of how well the gid box mean represents the actual conditions), or the $pCO_2$ of the gridded version (showing the variability within the gridded product).**

In the (b) and (c) subplots of Figure S1 we show maximal variability of $pCO_2$ individuals within a grid cell, i.e.,

$$\max_t \{pCO_{2,tij}^{\mathrm{max}} - pCO_{2,tij}^{\mathrm{min}}\}$$

where $t$, $ij$ indicates time and space indices. $pCO_{2,tij}^{max}$ and $pCO_{2,tij}^{min}$ were converted from the corresponding values of $CO_2$ fugacity observations which are available in the monthly gridded SOCATv2020 database. We think that the term '$pCO_2$ individuals' is correct in this sense. We have made a consistent use of the terms of 'observations' and 'gridded data' throughout the main text, and reworded the caption of Figure S1 to avoid any ambiguity.

480

Modified caption of Figure S1:

*(a) Spatial distribution of monthly gridded SOCATv2020 data. (b,c) Maximal variability of $pCO_2$ individual data within a $1° \times 1°$-grid box (µatm), i.e. $\max_t\{pCO_{2,tij}^{max} - pCO_{2,tij}^{min}\}$, where $t$ and $ij$ indicate time and space indices. $pCO_{2,tij}^{max}$ and $pCO_{2,tij}^{min}$ were converted from the corresponding values of $CO_2$ fugacity observations in the monthly gridded SOCATv2020 database. Fig. S1c shows the distribution of the variability larger than the $80\%$-quantile.*

**3.4 I understand that the authors used the subocean divisions from RECCAP 1. This of course increases the comparability to the results of RECCAP 1, but also this makes the results difficult to interpret. Using a biome scheme such as used in RECCAP 2 (e.g. after Fay and McKinley (2014)) would have led to a clearer separation of regions with similar characteristics, and thus increased the interpretability. I also miss a discussion of how this**

490 **product performs in comparison to other global air-sea $CO_2$ flux products.**

We are on the same page with Referee 3 that using biomes proposed by Fay and McKinley (2014) would provide a better interpretation of small-scale characteristics of $pCO_2$ and air-sea $CO_2$ fluxes. However, geometries of the biomes (e.g., their boundaries) would complicate the evaluation of the CMEMS-LSCE-FFNN model estimates and uncertainty, e.g., the comparison between the model outputs and sparse SOCAT data, and the analysis of results obtained for the open ocean and the coastal

495 regions. Regarding the scope of this study, we have chosen to use the subocean divisions with latitude bands. Consequently, the CMEMS-LSCE-FFNN estimates of regional air-sea $CO_2$ fluxes have been compared to the ones presented in RECCAP1.

Each of the observation–based reconstruction methods for $pCO_2$ and $CO_2$ fluxes has both strengths and weaknesses. We have revised the Introduction (Lines 37-55 in the first manuscript) to better highlight the differences between the present

500 approach and previous ones, as well as to emphasize its novelty. See new paragraphs of the Introduction in Lines 35-85 in the revised manuscript. In-depth intercomparisons amongst different model-based and/or observation–based products are presented in previous works including Rödenbeck et al. (2015), Denvil-Sommer et al. (2019), and Hauck et al. (2020) are beyond the scope of this study.

**3.5 Minor suggestions**

505 • **L 40: $pCO_2$ was not introduced as an abbreviation.**

• **L 59: Tr is not described.**

• **L 86: change to: p is the number of grid cells with observations.**

• **Figure 3: The yellow bars in panel c) are very difficult to read, especially the first one.**

• **Figure 4a/b: You show the number of observation (or grid cells with observations) per year. Please change that.**

510 We took these suggestions into account in this revision. Particularly for $Tr$, it was defined in Eq (1) in the first manuscript with respect to $Tr = kL(1 - f_{ice})$. We deleted $Tr$ from Eq. (1) since it is not referred elsewhere throughout the text. In the revised manuscript, Eq. (1) is as follows

$$
\begin{aligned}
fgCO_2 &= kL(1 - f_{ice})\ \Delta pCO_2 \\
&= kL(1 - f_{ice})\left(pCO_2^{atm} - pCO_2\right).
\end{aligned} \tag{1}
$$

**3.6 Additionally change STD to s. Go through the manuscript and make sure, that you use consistent terminology.**

STD was changed to $\sigma$ for a consistent use of its notation.

**3.7 L 140, add temporal offsets from the cell center. In many regions this will be the dominant one, especially during the productive season.**

Temporal offsets of data sampling is now mentioned in the sentence in Lines 139-141 of the first manuscript. The revised text is below.

Lines 197-200 in the revised manuscript:
*Variability in the sampling time and location of cruises and instruments induces temporal sampling bias (e.g., towards some days in a month and/or the summer months at high latitudes) and latitude and longitude offsets from the cell center (e.g., with an average of $0.34° \pm 0.14°$ as reported in Sabine et al., 2013) which are not taken into account.*

**3.8 L 182: Be aware that Laruelle et al. (2017) and Landschützer et al. (2020) are climatologies.**

Landschützer et al. (2020) present a global ocean $pCO_2$ climatology product combining two individual data reconstructions: one for the open ocean proposed by Landschützer et al. (2016) using SOCAT data at a $1° \times 1°$ resolution over 1982-2016 and another for the coastal ocean proposed by Laruelle et al. (2017) using coastal SOCAT data at a $0.25° \times 0.25°$ resolution over 1998-2015. As stated in these two studies, their reconstructed data are available for each month. In Landschützer et al. (2020) (Table 1), the authors reported RMSDs (RMSEs in their study) of each of the open-ocean and coastal-ocean reconstructions computed with respect to monthly gridded SOCATv5 data over the common period 1998-2015. In Section 3.3, they then present a comparison between their seasonal climatology product and a seasonal climatology computed from SOCAT data but are aware that this assessment could not be robust due to temporal sampling bias of SOCAT observations.

In Lines 179-185 in the first manuscript, we cited 26.8 $\mu$atm in Landschützer et al. (2020) (Table 1) which is the RMSD of the coastal reconstruction by Laruelle et al. (2017). Even though the CMEMS-LSCE-FFNN RMSD (35.84 $\mu$atm) reported here were computed using Eq. (4) over the same period as in the previous study, RMSDs of the two approaches are quite different. Our study uses the MARCATS mask proposed by Laruelle et al. (2013) leading to a smaller coastal area and a different number of evaluation data than those in Laruelle et al. (2017) and Landschützer et al. (2020). These two studies use the coastal mask defined within $400$ km distance from the sea shore. In addition, the leave-$p$-out cross-validation used in CMEMS-LSCE-FFNN model fitting permits to leave much more independent data for model evaluation than the previous approaches (see Sections Introduction and Methods in the revised manuscript).

We rewrote Lines 179-185 in the first manuscript to better highlight differences in model errors between these coastal reconstructions.

Lines 238-246 in the revised manuscript:
*For the 1998–2015 period, the CMEMS-LSCE-FFNN approach scored an RMSD of 35.84 $\mu$atm while a recent coastal reconstruction by Landschützer et al. (2020) obtained an error of 26.8 $\mu$atm (see their Table 1). The latter presents a global ocean $pCO_2$ climatology product by unifying data over the same period from two conceptually equivalent reconstruction models: one covering the open ocean at a $1° \times 1°$ resolution (Landschützer et al., 2016) and one targeting the coastal ocean at a $0.25° \times 0.25°$ resolution (Laruelle et al., 2017). These heretofore reconstructions cover the coastal region with a broader definition ($400$ km distance from the sea shore) than the MARCATS mask used in this study leading to the differences in characteristics and numbers of evaluation data of $pCO_2$. In addition, the CMEMS-LSCE-FFNN model was designed with the leave-$p$-out cross-validation approach excluding much more independent data from monthly model fitting for model evaluation than in the previous models.*

**3.9 L 307: Please round the uncertainties to 2 significant digits (or less if it seems unrealistically low) and the measured value to the same number of digits, for example $2.336 \pm 0.104$ to $2.34 \pm 0.10$. Please do so for all uncertainties in the manuscript.**

We have rounded estimates of air-sea fluxes and uncertainties to 3 digits since some of them become 0 with less than 3 digits; for instance, fluxes and uncertainty estimates over coastal regions (see in Table 2).

**3.10 L 328-330: Please correct this. Primary production and respiration have usually only a very small influence on alkalinity (if we neglect anerobic remineralization processes for the moment): primary production increases alkalinity, while remineralization processes reduce alkalinity**

We have corrected it. Alkalinity was removed from the discussion in Lines 328-330 of the first manuscript. Below is the modified text.

Lines 419-422 in the revised manuscript:

*High wind speeds also strengthen vertical mixing, a process supplying dissolved inorganic carbon (DIC) and nutrients to the surface ocean. During the spring and summer months, a vigorous biological activity (Sigman and Hain, 2012) counteracts the warming induced decrease in $CO_2$ solubility and increase in $pCO_2$ by drawing down DIC (Feely et al., 2001).*

**3.11 L 332: Another important influence factor in coastal regions is the inflow of terrestrial POC, e.g. in the southern North Sea, leading to the release of $CO_2$ to the atmosphere.**

Thank you. The impact of the inflow of terrestrial nutrients on air-sea $CO_2$ exchanges over the coastal SpA is now discussed in our revised manuscript.

Lines 424-427 in the revised manuscript:

*This contrasts with other coastal regions (e.g., southern North Sea and Baltic Sea) where the respiration of terrestrial particulate organic carbon from river run-off contributes to making these areas a strong seasonal source of $CO_2$ (Borgesa and Gypensb, 2010; Becker et al., 2021).*

**3.12 L 376-377: Are these really the dominant factors? After you argumentation for why the open ocean region is neutral (vertical convection brings up old, DIC rich water which balances the influx during summer) I would expect the absence of this deep mixing in coastal, shallow regions to be one of the major reasons why the coastal regions are a larger sink than the open ocean.**

As shown in Bates (2006), Arrigo et al. (2010), and Ishii et al. (2014), surface DIC concentration is higher over the open, deep basins than the shallow coastal seas of the subpolar Pacific, particularly induced by deep mixing during winter/spring. Also in this period, the coastal sector is covered by seasonal sea-ice resulting in a neutral region of air-sea fluxes while the open sea-ice free ocean (e.g., the southern Bering Sea) acts as a strong source of $CO_2$. In spring/summer time, high $CO_2$ uptake is found in coastal shelf seas influenced by sea-ice retreat and high biological production, e.g. Chukchi and Gulf of Alaska (Yasunaka et al., 2016, 2018), and/or by dilution of sea waters from river freshwater with low salinity and DIC concentration, e.g., Beaufort Sea, Laptev, and East Siberia Seas (Arrigo et al., 2010).

Thank you for correction. We have rephrased our argument with a broader context as follows.

Lines 469-475 in the revised manuscript:

*As shown in Bates (2006), Arrigo et al. (2010), and Ishii et al. (2014), surface DIC concentration is higher over the open, deep basins than the shallow coastal seas of the SpP, particularly induced by deep mixing during winter/spring. Over the same period, seasonal sea-ice also restricts gas exchanging, the coastal sector thus acts as a neutral region of $CO_2$ fluxes (Fig. 8). During spring and summer, a substantial amount of $CO_2$ is also absorbed in the coastal shelf seas influenced by high biological production in large ice-free areas (e.g., Chukchi and Gulf of Alaska), and/or by dilution of sea waters from river freshwater*

*with low salinity and DIC concentration (e.g., Beaufort, Laptev, and East Siberia Seas) (Arrigo et al., 2010; Yasunaka et al., 2016, 2018).*

**3.13    To be honest, I can't really see from this figure that it covaries with the ENSO mode. As I see it the flux increases equally often during La Nina as during El Nino. It would be more interesting to see a comparison of the interannual variability with other air-sea flux products.**

The covariate between the ENSO events and the temporal variability of the global carbon sink has been covered by its increasing long-term trend in Figure 9 in the first manuscript. We have added another curve whose values are ticked on the right y-axis of the same figure. This curve stands for the yearly flux variability, i.e., the yearly ocean uptake estimate after removing its long-term trend. See the revised version of Figure 9 below.

In-depth intercomparisons amongst different model-based and/or observation–based products are presented in previous works including Rödenbeck et al. (2015); Denvil-Sommer et al. (2019); Hauck et al. (2020). This study further focuses on presenting the three main points: (1) an introduction of our novel ensemble-based approach for reconstructing the global monthly $pCO_2$ fields and air–sea fluxes from the open ocean to coastal regions, (2) quantification and evaluation of model best estimates and uncertainties based on the ensemble asset for different time scales (e.g., monthly, yearly, and multi-decadal) and spatial scales (e.g., grid cells, sub-basins, and the global ocean), (3) a seamless analysis of the reconstructed data and uncertainty estimates over the open ocean and coastal zones. These key points are now also elaborated in the Introduction (see Lines 35-85 in the revised manuscript). Regarding the scope of this study and intense analysis presented in the current manuscript, we suggest to retain Figure 9 as it is revised as follows. However, this point raised by the referee is interesting and will be considered in our future studies.

[Figure]

**Figure 9.** Yearly global integrated air–sea flux estimates derived from the CMEMS-LSCE-FFNN ensemble (mean ± uncertainty) for 1985–2019. Multivariate El Niño-Southern Oscillation Index (MEI; Wolter and Timlin, 1993, https://psl.noaa.gov/enso/mei/, last access: December 2020) is used to generally indicate a link between variations, e.g. Yearly uptake - Trend , in the CMEMS-LSCE-FFNN sink estimate and the ENSO climate mode (El Niño: MEI > 0.5, La Niña: MEI < -0.5, Neutral: otherwise).
* * *
**     In addition to the comments addressed above, we also account editorial suggestions from our colleague Nicolas Metzl. We would like to thank him for his comments and supports to improve our manuscript. Aside from some points matching with the ones suggested by the three referees, the following items make other changes in this revision from the first manuscript:

- The color in Figures 5 (a,b), 8, and S9 (a,b) is reset: blue for $CO_2$ sink and red for $CO_2$ sources.

625
- The study of Yasunaka et al. (2018) is cited when discussing $CO_2$ fluxes over some sectors in the Arctic and subpolar regions.

- "Fair data used statement for SOCAT" is added in Acknowledgements.